# Spatio-Spectral Graph Neural Networks

**Simon Geisler**[†]**, Arthur Kosmala**[†]**, Daniel Herbst, and Stephan Günnemann**
Department of Computer Science & Munich Data Science Institute
Technical University of Munich
{s.geisler, a.kosmala, d.herbst, s.guennemann}@tum.de

## Abstract

Spatial Message Passing Graph Neural Networks (MPGNNs) are widely used for learning on graph-structured data. However, key limitations of $\ell$-step MPGNNs are that their "receptive field" is typically limited to the $\ell$-hop neighborhood of a node and that information exchange between distant nodes is limited by over-squashing. Motivated by these limitations, we propose *Spatio-Spectral Graph Neural Networks (S²GNNs)* – a new modeling paradigm for Graph Neural Networks (GNNs) that synergistically combines spatially and spectrally parametrized graph filters. Parameterizing filters partially in the frequency domain enables global yet efficient information propagation. We show that S²GNNs vanquish over-squashing and yield strictly tighter approximation-theoretic error bounds than MPGNNs. Further, rethinking graph convolutions at a fundamental level unlocks new design spaces. For example, S²GNNs allow for free positional encodings that make them strictly more expressive than the 1-Weisfeiler-Leman (WL) test. Moreover, to obtain general-purpose S²GNNs, we propose spectrally parametrized filters for directed graphs. S²GNNs outperform spatial MPGNNs, graph transformers, and graph rewirings, e.g., on the peptide long-range benchmark tasks, and are competitive with state-of-the-art sequence modeling. On a 40 GB GPU, S²GNNs scale to millions of nodes.

## 1 Introduction

*Spatial* Message-Passing Graph Neural Networks (MPGNNs) ushered in various recent breakthroughs. For example, MPGNNs are able to predict the weather with unprecedented precision (Lam et al., 2023), can be composed as a foundation model for a rich set of tasks on knowledge graphs (Galkin et al., 2023), and are a key component in the discovery of millions of AI-generated crystal structures (Merchant et al., 2023). Despite this success, MPGNNs produce node-level signals solely considering *limited-size neighborhoods*, effectively bounding their expressivity. Even with a large number of message-passing steps, MPGNNs are limited in their capability of propagating information to distant nodes due to *over-squashing*. As evident by the success of global models like transformers (Vaswani et al., 2017), modeling long-range interactions can be pivotal and an important step towards foundation models that understand graphs.

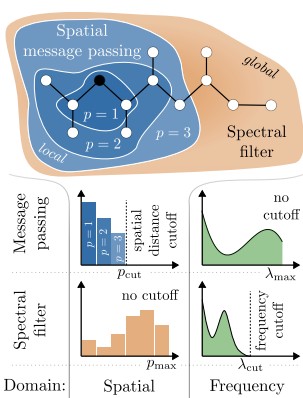

Figure 1: S²GNN principle.

**We propose *Spatio-Spectral Graph Neural Networks (S²GNNs)***, a new modeling paradigm for tackling the aforementioned limitations, that synergistically combine *message passing* with *spectral filters*, explicitly parametrized in the spectral domain. *Spectral filters* are virtually ignored by prior work but go beyond stacks of message-passing layers or polynomial parametrizations. Due to *message passing's* finite number

38th Conference on Neural Information Processing Systems (NeurIPS 2024). † equal contribution.

of propagation steps, it comes with a distance cutoff $p_{\text{cut}}$ (# hops, see Fig. 1). Conversely, *spectral filters* act globally ($p_{\max}$), even on a truncated frequency spectrum $\lambda_{\text{cut}}$. Truncating the frequency spectrum for *spectral filters* is required for efficiency, yet *message passing* has access to the entire spectrum (right plots in Fig. 1). The combination of *message passing and spectral filters* provably leverages the strengths of each parametrization. Utilizing this combination, S²GNNs generalize the concept of "virtual nodes" and distill many important properties of hierarchical message-passing schemes, graph-rewirings, and pooling into a single GNN (see Fig. 3). Outside of GNNs, a similar composition is at the core of some State Space Models (SSM) models (Poli et al., 2023), that deliver transformer-like properties with superior scalability on sequences – as do S²GNNs on graphs.

**Our analysis of S²GNNs** (§ 3.1) validates their capability for modeling long-range interactions. We prove in § 3.1.1 that combining spectral and spatial filters alleviates the over-squashing phenomenon (Alon & Yahav, 2020; Di Giovanni et al., 2023a,b), a necessity for effective information-exchange among distant nodes. Our approximation-theoretic analysis goes one step further and proves strictly tighter error bounds in terms of approximation of the target idealized GNN (§ 3.1.2).

**Design space of S²GNNs** (§ 3.2). Except for initial works like (Bruna et al., 2014) and in contrast to spatial MPGNNs, the design decisions for spectral filters are virtually unexplored – and so is their composition. The novel aspects of S²GNN's design space include the *spectral filter parametrization* (§ 3.2.1). We propose the first permutation-equivariance-preserving *neural network in the spectral domain* (§ 3.2.2) and generalize *spectral filters to directed graphs* (§ 3.2.3). The dual use of the partial eigendecomposition, required for spectral filters, allows us to propose "free-of-cost" *positional encodings* (§ 3.2.4), that are permutation-equivariant, stable, and increase expressivity strictly beyond the 1-Weisfeiler-Leman (WL) test.

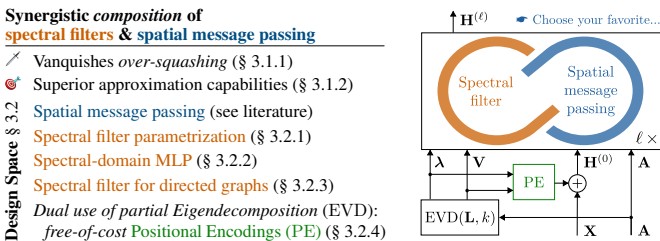

Figure 2: S²GNN framework with adjacency matrix $\boldsymbol{A}$, node features $\boldsymbol{X}$, and Laplacian $\boldsymbol{L}$ (function of $\boldsymbol{A}$).

**S²GNNs are effective and practical.** We empirically verify the shortcomings of MPGNNs and how S²GNNs overcome them (§ 4). E.g., we set a new state-of-the-art on peptides-func (Dwivedi et al., 2022) with ≈ 35% fewer parameters, outperforming MPGNNs and graph transformers. Although sequences are just a subdomain of (directed) graphs, we also study how S²GNNs compare to specialized sequence models like transformers (Vaswani et al., 2017) or Hyena (Poli et al., 2023). We find that S²GNNs are highly competitive even though they operate on a much more general domain (un-/directed graphs). Last, the runtime and space complexity of S²GNNs is equivalent to MPGNNs and, with vanilla full-graph training, S²GNNs can handle millions of nodes with a 40 GB GPU.

## 2 Background

We study graphs $\mathcal{G}(\boldsymbol{A}, \boldsymbol{X})$ with adjacency matrix $\boldsymbol{A} \in \{0,1\}^{n \times n}$ (or $\boldsymbol{A} \in \mathbb{R}_{\geq 0}^{n \times n}$ if weighted), node features $\boldsymbol{X} \in \mathbb{R}^{n \times d}$ and edge count $m$. $\boldsymbol{A}$ is symmetric for undirected graphs and, thus, has eigendecomposition $\boldsymbol{\lambda}, \boldsymbol{V} = \text{EVD}(\boldsymbol{A})$ with eigenvalues $\boldsymbol{\lambda} \in \mathbb{R}^n$ and eigenvectors $\boldsymbol{V} \in \mathbb{R}^{n \times n}$: $\boldsymbol{A} = \boldsymbol{V}\boldsymbol{\Lambda}\boldsymbol{V}^\top$ using $\boldsymbol{\Lambda} = \text{diag}(\boldsymbol{\lambda})$. Instead of $\boldsymbol{A}$, we decompose the *Laplacian* $\boldsymbol{L} := \boldsymbol{I} - \boldsymbol{D}^{-1/2}\boldsymbol{A}\boldsymbol{D}^{-1/2}$, with diagonal degree matrix $\boldsymbol{D} = \text{diag}(\boldsymbol{A1})$, since its ordered eigenvalues $0 = \lambda_1 \leq \lambda_2 \leq \cdots \leq \lambda_n \leq 2$ are similar to frequencies (e.g., low eigenvalues relate to low frequencies, see Fig. 4). Likewise, one could use, e.g., $\boldsymbol{L} = \boldsymbol{I} - \boldsymbol{D}^{-1}\boldsymbol{A}$ or more general variants (Yang et al., 2023); however, we focus our explanations on the most common choice $\boldsymbol{L} := \boldsymbol{I} - \boldsymbol{D}^{-1/2}\boldsymbol{A}\boldsymbol{D}^{-1/2}$. We choose the matrix of eigenvectors $\boldsymbol{V} \in \mathbb{R}^{n \times n}$ to be orthogonal $\boldsymbol{V}\boldsymbol{V}^\top = \boldsymbol{I}$. We refer to $\boldsymbol{V}$ as the Fourier basis of the graph, with Graph Fourier Transformation (GFT) $\hat{\boldsymbol{X}} = \boldsymbol{V}^\top \boldsymbol{X}$ and its inverse $\boldsymbol{X} = \boldsymbol{V}\hat{\boldsymbol{X}}$. To provide an overview, Table 5 lists the symbols used in this work.

**Spectral graph filters.** Many GNNs implement a graph convolution, where node signal $\boldsymbol{X} \in \mathbb{R}^{n \times d}$ is convolved $\boldsymbol{g} *_{\mathcal{G}} \boldsymbol{X}$ for every $d$ with a scalar filter $\boldsymbol{g} \in \mathbb{R}^n$. The graph convolution (Hammond et al., 2011) is defined in the spectral domain as $\boldsymbol{g} *_{\mathcal{G}} \boldsymbol{X} := \boldsymbol{V}([\boldsymbol{V}^\top \boldsymbol{g}] \odot [\boldsymbol{V}^\top \boldsymbol{X}])$, with element-wise product $\odot$ and broadcast of $\boldsymbol{V}^\top \boldsymbol{g}$ to match shapes. Instead of spatial $\boldsymbol{g}$, spectral graph filters parametrize $\hat{g} : [0, 2] \to \mathbb{R}$ explicitly and yield $\boldsymbol{V}^\top \boldsymbol{g} := \hat{g}(\boldsymbol{\lambda}) \in \mathbb{R}^n$ as a function of the eigenvalues.

**Message Passing Graph Neural Networks (MPGNNs)** circumvent the EVD via polynomial $\hat{g}(\boldsymbol{\lambda})_u = \sum_{j=0}^{p} \gamma_j \lambda_u^j$ since $\boldsymbol{V}[\sum_{j=0}^{p} \gamma_j \operatorname{diag}(\boldsymbol{\lambda})^j] \boldsymbol{V}^\top \boldsymbol{X} = \sum_{j=0}^{p} \gamma_j \boldsymbol{L}^j \boldsymbol{X}$. In practice, many MPGNNs use $p = 1$: $\boldsymbol{H}^{(l)} = (\gamma_0 \boldsymbol{I} + \gamma_1 \boldsymbol{L}) \boldsymbol{H}^{(l-1)}$ with $\boldsymbol{H}^{(0)} = \boldsymbol{X}$, and stack $1 \le l \le \ell$ layers interleaved with node-wise transformations and activations $\sigma$. We refer to Balcilar et al. (2021b) for similar interpretations of MPGNNs like GAT (Veličković et al., 2018) or GIN (Xu et al., 2019).

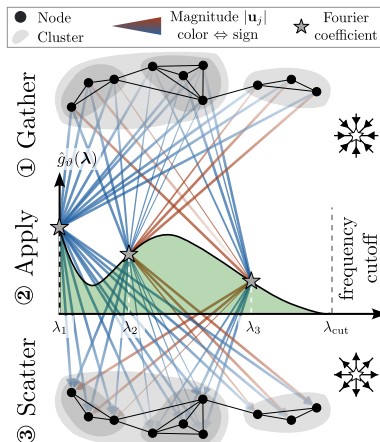

## 3 Method

*$S^2$GNNs* symbiotically pair spatial $\operatorname{Spatial}(\boldsymbol{H}^{(l-1)}; \boldsymbol{A})$ MPGNNs and $\operatorname{Spectral}(\boldsymbol{H}^{(l-1)}; \boldsymbol{V}, \boldsymbol{\lambda})$ filters, using a partial eigendecomposition. Even though the spectral filter operates on a truncated eigendecomposition (**spectrally bounded**), it is **spatially unbounded**. Conversely, spatial MPGNNs are **spatially bounded** yet **spectrally unbounded** (see Fig. 1).

A spectrally bounded filter is sensible for modeling global pair-wise interactions, considering its **message-passing interpretation** of Fig. 3. Conceptually, a spectral filter consists of three steps: ① Gather: The multiplication of the node signal with the eigenvectors $\boldsymbol{v}_u^\top \boldsymbol{X}$ (GFT) is a weighted and signed aggregation over all nodes; ② Apply: the "Fourier co-

Figure 3: **Message-passing interpretation** of $\boldsymbol{V}(\hat{g}_\vartheta(\boldsymbol{\lambda}) \odot [\boldsymbol{V}^\top \boldsymbol{X}])$ (spectral filter): via the Fourier coefficients they may **exchange information globally** and allow **intra- and inter-cluster message passing**. Edge width/color denotes the magnitude/sign of $\boldsymbol{V}$.

efficients" are weighted; and ③ Scatter broadcasts the signal $\boldsymbol{v}_u \hat{\boldsymbol{X}}$ back to the nodes (inverse GFT). The first eigenvector (here for $\boldsymbol{L} = \boldsymbol{D} - \boldsymbol{A}$) acts like a "virtual node" (Gilmer et al., 2017) (see also § E). That is, it calculates the average embedding and then distributes this information, potentially interlayered with neural networks. Importantly, the other eigenvectors effectively allow messages to be passed within or between clusters. As we show for exemplary graphs in Fig. 4, low frequencies/eigenvalues capture coarse structures, while high(er) frequencies/eigenvalues capture details. For example, the second eigenvector in Fig. 4b contrasts the inner with the outer rectangle, while the third eigenspace models both symmetries up/down and left/right. In conclusion, $S^2$GNNs augment spatial message-passing with a *graph-adaptive hierarchy* (spectral filter). Thus, **$S^2$GNNs distill many important properties of hierarchical message-passing schemes** (Bodnar et al., 2021)**, graph-rewirings** (Di Giovanni et al., 2023a)**, pooling** (Lee et al., 2019) etc. See § J.1 for more examples.

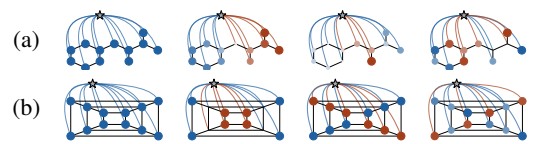

Figure 4: Exemplary (lowest) eigenspaces.

**$S^2$GNN's composition.** We study (1) an *additive combination* for its simpler approximation-theoretic interpretation (§ 3.1.2), or (2) an *arbitrary sequence of filters* due to its flexibility. In both cases, residual connections may be desirable (see § J.2).

$$\boldsymbol{H}^{(l)} = \operatorname{Spectral}^{(l)}(\boldsymbol{H}^{(l-1)}; \boldsymbol{V}, \boldsymbol{\lambda}) + \operatorname{Spatial}^{(l)}(\boldsymbol{H}^{(l-1)}; \boldsymbol{A}) \tag{1}$$

$$\boldsymbol{H}^{(\ell)} = (h^{(\ell)} \circ h^{(\ell-1)} \circ \cdots \circ h^{(1)})(\boldsymbol{H}^{(0)}) \quad \text{with } h^{(j)} \in \{\operatorname{Spectral}, \operatorname{Spatial}\} \tag{2}$$

**Spectral Filter.** The building block that turns a spatial MPGNN into an $S^2$GNN is the spectral filter:

$$\operatorname{Spectral}^{(l)}(\boldsymbol{H}^{(l-1)}; \boldsymbol{V}, \boldsymbol{\lambda}) = \boldsymbol{V}\left(\hat{g}_\vartheta^{(l)}(\boldsymbol{\lambda}) \odot \left[\boldsymbol{V}^\top f_\theta^{(l)}(\boldsymbol{H}^{(l-1)})\right]\right) \tag{3}$$

with a point-wise transformation $f_\theta^{(l)} : \mathbb{R}^{n \times d^{(l-1)}} \to \mathbb{R}^{n \times d^{(l)}}$, a learnable spectral filter $\hat{g}_\vartheta^{(l)}(\boldsymbol{\lambda}) \in \mathbb{R}^{k \times d^{(l)}}$ parameterized element-wise as $\hat{g}_\vartheta^{(l)}(\boldsymbol{\lambda})_{u,v} := \hat{g}_v^{(l)}(\lambda_u; \vartheta_v)$ (see § 3.2.1), and truncated $\boldsymbol{V} \in \mathbb{R}^{n \times k}, \boldsymbol{\lambda} \in \mathbb{R}^k$. Due to the combination of message passing with spectral filters, $S^2$GNNs' hypothesis class goes beyond (finite-order) polynomials of the Laplacian $\boldsymbol{L}$ (or stacks message passing layers), unlocking a larger class of filters. In Algo. 1, we provide pseudo code for $S^2$GNNs (Eq. 1).

**Truncated spectrum.** We omit extra notation for the truncated eigendecompositon $\operatorname{EVD}(\boldsymbol{L}, k)$ since it is equivalent to define $\hat{g}(\lambda_j) = 0$ for $j > k$. However, truncating after the $k$-th eigenvector requires care with the last eigenspace to maintain permutation equivariance. Due to the

ambiguity of eigenvectors in the presence of repeated eigenvalues, we must ensure that we only include eigenspaces in their entirety. That is, we only include $\{\lambda_j \,|\, j \leq k \wedge \lambda_j \neq \lambda_{k+1}\}$. Thus, $\text{Spectral}(\boldsymbol{H}^{(l-1)}; \text{EVD}(\boldsymbol{L}, k))$ is permutation equivariant nonetheless. We defer all proofs to § H.

**Theorem 1.** *$\text{Spectral}(\boldsymbol{H}^{(l-1)}; \text{EVD}(\boldsymbol{L}, k))$ of Eq. 3 is equivariant to all $n \times n$ permutation matrices $\boldsymbol{P} \in \mathcal{P}$: $\text{Spectral}(\boldsymbol{P}\boldsymbol{H}^{(l-1)}; \text{EVD}(\boldsymbol{P}\boldsymbol{L}\boldsymbol{P}^\top, k)) = \boldsymbol{P}\,\text{Spectral}(\boldsymbol{H}^{(l-1)}; \text{EVD}(\boldsymbol{L}, k))$.*

**Complementary high-resolution filters.** Our *Spectral filters are highly discriminative between the frequencies* and, e.g., can readily access a single eigenspace. Yet, for efficiency, we limit the spectral filter to a specific frequency band. *Due to the combination with message passing, this choice of band does not decide on, say, low-pass behavior; it solely determines where to increase the spectral selectivity.* While S²GNNs with subsequent guarantees adapt to domain-specific choices for the spectral filter's frequency band, a sensible default is to focus on the low frequencies. The two main reasons for this are (see § J.3 for an extensive list): (1) Low frequencies model the smoothest global signals w.r.t. the graph structure (see Fig. 3 & 4). (2) Under a relative perturbation model (perturbation budget proportional to degree), stability implies $C$-integral-Lipschitzness ($\exists C > 0 \colon |\lambda d\hat{g}/d\lambda| \leq C$), i.e., the filter can vary strongly around zero but must level out for large $\lambda$ (see Gama et al. (2020)).

## 3.1 Theoretical Analysis

We show that how S²GNNs alleviate oversquashing in § 3.1.1. Next, § 3.1.2 makes the approximation-theoretic advantages precise.

### 3.1.1 S²GNNs Vanquish Over-Squashing

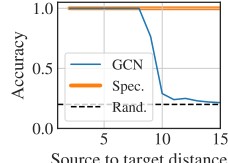

Figure 5: Spectral filters do not exhibit over-squashing on "Clique Path" graphs (Di Giovanni et al., 2023a).

Alon & Yahav (2020) pointed out that MPGNNs must pass information through bottlenecks that connect different communities using fixed-size embedding vectors. Topping et al. (2022) and Di Giovanni et al. (2023a) formalize this via an $L^1$-norm Jacobian sensitivity analysis: $\|\partial \mathbf{h}_v^{(\ell)}/\partial \mathbf{h}_u^{(0)}\|_{L^1}$ models the output's $\mathbf{h}_v^{(\ell)}$ change if altering input $\mathbf{h}_u^{(0)}$. MPGNNs' Jacobian sensitivity typically decays $\mathcal{O}\left(\exp\left(-r\right)\right)$ with node distance $r$ if the number of walks between the two nodes is small. See § F for results of Di Giovanni et al. (2023a).

S²GNNs are not prone to such an exponential sensitivity decay due to their global message scheme. We formalize this in Theorem 2, refer to Fig. 4 for intuition and Fig. 5 for empirical verification. All theoretical guarantees hold if a $\theta$ exists such that $f_\theta = \boldsymbol{I}$.

**Theorem 2.** *An $\ell$-layer S²GNN can be parametrized s.t. output $\mathbf{h}_v^{(\ell)}$ has a uniformly lower-bounded Jacobian sensitivity on a connected graph: $\|\partial \mathbf{h}_v^{(\ell)}/\partial \mathbf{h}_u^{(0)}\|_{L^1} \geq C_\vartheta d/m$ with rows $\boldsymbol{h}_u^{(0)}$, $\boldsymbol{h}_v^{(\ell)}$ of $\boldsymbol{H}^{(0)}$, $\boldsymbol{H}^{(\ell)}$ for nodes $u$, $v \in \mathcal{G}$, a parameter-dependent $C_\vartheta$, network width $d$ and edge count $m$.*

In contrast to Di Giovanni et al. (2023a), we prove a lower bound for S²GNNs, guaranteeing a minimum "influence" for any $u$ on $v$. This is true since S²GNNs contain a virtual node as a special case with $\hat{g}_\vartheta^{(l)}(\boldsymbol{\lambda}) = \mathbb{1}_{\{0\}}$, with $\mathbb{1}_\mathcal{S}$ denoting the indicator function of a set $\mathcal{S}$ (see also § E). However, we find that a virtual node is insufficient for some long-range tasks, including our long-range clustering (LR-CLUSTER) of Fig. 10b. Hence, the exponential sensitivity decay of spatial MPGNNs only shows their inadequacy in long-range settings. Proving its absence is not sufficient to quantify long-range modeling capabilities, noting that the lower bound is not tight for S²GNNs on many graphs. We close this gap with our subsequent analysis rooted in polynomial approximation theory.

### 3.1.2 Approximation Theory: Superior Error Bounds Despite Spectral Cutoff

To demonstrate how S²GNNs can express a more general hypothesis class than MPGNNs, we study how well an "idealized" GNN (IGNN) can be approximated. Each IGNN layer $l$ can express convolution operators $g^{(l)}$ of *any* spectral form $\hat{g}^{(l)} \colon [0, 2] \to \mathbb{R}$. We approximate IGNNs with S²GNNs from Eq. 1, with a spectral filter as in Eq. 3 and a spatial part parametrized by a polynomial. While we assume here that the S²GNN spectral filter is bandlimited to and a universal approximator on the interval $[0, \lambda_{\max}]$, the findings generalize to, e.g., a high-pass interval. In the main body, we focus on the key insights for architectures without nonlinear activations. Wang & Zhang (2022) prove that even linear IGNNs can produce any one-dimensional output under certain regularity assumptions on the graph and input signal. Thus, we solely need to consider a single layer. In § H.4, we cover the generic setting including nonlinearities, where multiple layers are helpful.

**Locality relates to spectral smoothness.** The locality of the true/ideal filter $g$ is related to the smoothness of its Fourier transform $\hat{g}$. For instance, if $g$ is a low-order polynomial of $\boldsymbol{L}$, it is localized to a few-hop neighborhood, and $\hat{g}$ is regularized to vary slowly (Fig. 6a w/o discontinuity). The other extreme is a discontinuous spectral filter $\hat{g}$, such as the entirely non-local virtual node filter, $\hat{g} = \mathbb{1}_{\{0\}}$ (discontinuity in Fig. 6a, details in § E). This viewpoint of

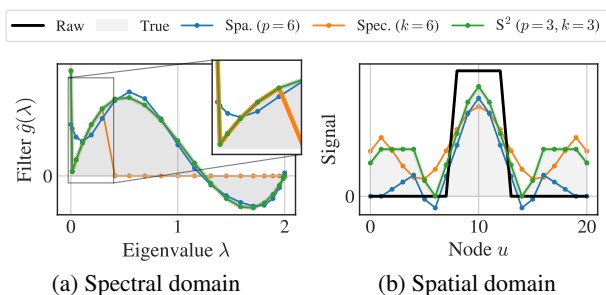

(a) Spectral domain      (b) Spatial domain

Figure 6: $S^2$ filter perfectly approximates true filter (a) with a discontinuity at $\lambda = 0$, while polynomial ("Spa.") and spectral ("Spec.") alone do not. (b) shows responses on a path graph.

spectral smoothness illuminates the limitations of finite-hop message passing from an angle that complements spatial analyses in the over-squashing picture. It informs a lower bound on the error, which shows that spatial message passing, i.e, order-$p$ polynomial graph filters $g_{\gamma_p}$ with $p+1$ coefficients $\gamma_p \in \mathbb{R}^{p+1}$, can converge exceedingly slowly – slower than any inverse root (!) of $p$ – to a discontinuous ground truth in the Frobenius-induced operator norm:

**Theorem 3.** *Let $\hat{g}$ be a discontinuous spectral filter. For any approximating sequence $\left(g_{\gamma_p}\right)_{p \in \mathbb{N}}$ of polynomial filters, an adversarial sequence $(\mathcal{G}_p)_{p \in \mathbb{N}}$ of input graphs exists such that*

$$\nexists \alpha \in \mathbb{R}_{>0}: \quad \sup_{0 \neq \boldsymbol{X} \in \mathbb{R}^{|\mathcal{G}_p| \times d}} \frac{\|(g_{\gamma_p} - g) *_{\mathcal{G}_p} \boldsymbol{X}\|_{\mathrm{F}}}{\|\boldsymbol{X}\|_{\mathrm{F}}} = \mathcal{O}\left(p^{-\alpha}\right)$$

**Superior $S^2$GNN error bound.** A spatio-spectral convolution wins over a purely spatial filter when the sharpest irregularities of the ground truth $\hat{g}$ are within reach of its expressive spectral part. The spatial part, which can "focus" on learning the remaining, smoother part outside of this window, now needs much fewer hops to give a faithful approximation. We illustrate this principle in Fig. 6 where we approximate an additive combination of an order-three polynomial filter with discontinuous low-pass. Only the $S^2$ filter is faithfully approximating this filter. Formally, we find:

**Theorem 4.** *Assume $\hat{g}\big|_{[\lambda_{cut},2]}$ is $r$-times continuously differentiable on $[\lambda_{cut}, 2]$, and a bound $K_r(\hat{g}, \lambda_{cut}) \geq 0$ such that $\left|\frac{d^r}{d\lambda^r}\hat{g}(\lambda)\right| \leq K_r(\hat{g}, \lambda_{cut}) \,\forall \lambda \in [\lambda_{cut}, 2]$. An approximating $S^2$GNN sequence with parameters $\left(\vartheta_p^*, \gamma_p^*\right)_{p \in \mathbb{N}}$ exists such that, for arbitrary graph sequences $(\mathcal{G}_p)_{p \in \mathbb{N}}$,*

$$\sup_{0 \neq \boldsymbol{X} \in \mathbb{R}^{|\mathcal{G}_p| \times d}} \frac{\|(g_{\gamma_p^*} + g_{\vartheta_p^*} - g) *_{\mathcal{G}_p} \boldsymbol{X}\|_F}{\|\boldsymbol{X}\|_F} = \mathcal{O}\left(K_r(\hat{g}, \lambda_{cut})p^{-r}\right)$$

*with a scaling constant that depends only on $r$, not on $\hat{g}$ or $(\mathcal{G}_p)_{p \in \mathbb{N}}$.*

The above bound extends to purely spatial convolutions in terms of $K_r(\hat{g}, 0)$ if $\hat{g}$ is $r$-times continuously differentiable on the full interval $[0, 2]$. The $S^2$GNN bound of Theorem 4 is then still strictly tighter if $K_r(\hat{g}, \lambda_{\text{cut}}) < K_r(\hat{g}, 0)$. In particular, taking the limit $K_1(\hat{g}, 0) \to \infty$ towards discontinuity makes the purely spatial upper bound arbitrarily loose, whereas a benign filter might still admit a small $K_1(\hat{g}, \lambda_{\text{cut}})$ for some $\lambda_{\text{cut}} > 0$. Theorem 3 suggests that this is not an artifact of a loose upper bound but that there is an inherent difficulty in approximating unsmooth filters with polynomials.

We conclude the analysis by instantiating the bounds: assuming $\hat{g}$ is $C$-integral-Lipschitz for stability reasons (see Gama et al. (2020) and the paragraph before § 3.1.1) yields $K_1(\hat{g}, \lambda_{\text{cut}}) = C/\lambda_{\text{cut}}$, whereas for the electrostatics example $\hat{g}_\sigma$ in § G, we find upper bounds $K_r(\hat{g}_\sigma, \lambda_{\text{cut}}) = r!/\lambda_{\text{cut}}^{(r+1)}$. In both cases, the pure spatial bound diverges as smoothness around 0 remains unconstrained.

## 3.2 Design Space

As shown in Fig. 2, we identify three major, yet unexplored, directions in $S^2$GNNs' design space. In § 3.2.1, we discuss how we parametrize the spectral filter. In § 3.2.2, we propose the first neural network for the spectral domain. That is, we allow transformations and non-linearities in the "Fourier" domain. In § 3.2.3, we are the first to instantiate spectral filters for directed graphs. Additionally,

due to the availability of the partial eigendecomposition, positional encodings may dual use them to improve epxressivity at negligible cost. In § 3.2.4, we propose the first permutation equivariant, stable and efficient positional encodings that provably admit an expressivity beyond 1-WL. § J provides further details and considerations, like some remarks on batching (§ J.7). For the (sub-) design space of spatial message passing (You et al., 2020), we refer to its rich literature.

### 3.2.1 Parametrizing Spectral Filters

For spectral filter function $\hat{g}_\vartheta(\boldsymbol{\lambda})$ of Eq. 3, we learn a channel-wise linear combination of translated Gaussian basis functions (see "Gaussian smearing" used by Schütt et al. (2017)), as depicted in Fig. 7. This choice (1) may represent any possible $\hat{g}_\vartheta(\boldsymbol{\lambda})$ with sufficient resolution (assumption in § 3.1.2); (2) avoids overfitting towards numerical inaccuracies of the eigenvalue calculation; (3) limits the discrimination of almost repeated eigenvalues and, in turn, should yield stability (similar to § 3.2.4). Strategies to cope with a variable $\lambda_{\text{cut}}$ and $k$ (e.g., using attention similar to SpecFormer (Bo et al., 2023a)) did usually not yield superior experimental results.

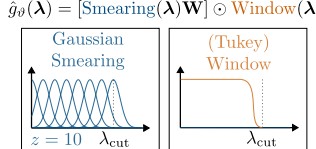

$$\hat{g}_\vartheta(\boldsymbol{\lambda}) = [\text{Smearing}(\boldsymbol{\lambda})\mathbf{W}] \odot \text{Window}(\boldsymbol{\lambda})$$

Figure 7: $\hat{g}_\vartheta(\boldsymbol{\lambda})$ with $\text{Smearing}(\boldsymbol{\lambda}) : [0,2]^k \to \mathbb{R}^{k\times z}$, linear map $\boldsymbol{W} \in \mathbb{R}^{z\times d}$ ($\vartheta = \{\boldsymbol{W}\}$), and fixed window function $\text{Window}(\boldsymbol{\lambda})$.

**Window.** We multiply the learned combinations of Gaussians by an envelope function (we choose a Tukey window) that decays smoothly to zero around cutoff $\lambda_{\text{cut}}$. This counteracts the so-called "Gibbs phenomenon" (aka "ringing"): as visualized for a path graph/sequence of 100 nodes in Fig. 8, trying to approximate a spatially-discontinuous target signal using an ideal low-pass range of frequency components results in an overshooting oscillatory behavior near the spatial discontinuity. Dampening the frequencies near $\lambda_{\text{cut}}$ by a smooth envelope/window function alleviates this behavior. We note that the learned filter may, in principle, overrule the windowing at the cost of a higher weight decay penalty. See Algo. 2 for $\hat{g}_\vartheta(\boldsymbol{\lambda})$'s algorithmic description.

**Depth-wise separable convolution** (Sifre, 2014; Howard et al., 2017): Applying different filters for each dimension is computationally convenient for spectral filters. While "full" convolutions are also possible, we find that such a construction is more prone to over-fitting. In practice, we even use parameter sharing and apply fewer filters than dimensions to counteract over-fitting. We argue that sharing filters among dimensions is similar to the heads in a transformer (Vaswani et al., 2017).

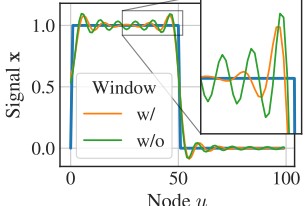

Figure 8: Ringing of ideal low pass filter on path graph.

**Feature transformations** $f_\theta^{(l)}$**.** As sketched in Fig. 3 & 4, all nodes participate in the global data transfer. While this global message-passing scheme is *graph-adaptive*, it does not adjust to the inputs. For adaptivity, we typically consider non-linear feature transformations $f_\theta^{(l-1)}(\boldsymbol{H}^{(l-1)})$, like gating mechanism $f_\theta^{(l-1)}(\boldsymbol{H}^{(l-1)}) = \boldsymbol{H}^{(l-1)} \odot \sigma'(\boldsymbol{H}^{(l-1)}\boldsymbol{W}_G^{(l)} + \vec{1}\boldsymbol{b}^\top))$ with element-wise multiplication $\odot$, SiLU function $\sigma'$, learnable weight $\boldsymbol{W}$, and bias $\boldsymbol{b}$. A linear transformation $f_\theta^{(l)}(\boldsymbol{H}^{(l-1)}) = \boldsymbol{H}^{(l-1)}\boldsymbol{W}^{(l)}$ is another interesting case since we may first apply the GFT and then the transformation: $(\boldsymbol{V}^\top\boldsymbol{H}^{(l-1)})\boldsymbol{W}^{(l)}$. Next, we extend this linear transformation to a neural network in the spectral domain by adding multiple transformations and nonlinearities.

### 3.2.2 Neural Network for the Spectral Domain

Applying a neural network $s_\zeta$ in the spectral domain is highly desirable due to its negligible computational cost if $k \ll n$. Moreover, $s_\zeta$ allows the spectral filter to become data-dependent and may mix between channels. Data-dependent filtering is one of the properties that is hypothesized to make transformers powerful Fu et al. (2023). We propose the first neural network for the spectral domain of graph filters $s_\zeta^{(l)} : \mathbb{R}^{k\times d^{(l)}} \to \mathbb{R}^{k\times d^{(l)}}$ that is designed to preserve permutation equivariance.

$$\boldsymbol{H}^{(l)} = \text{Spectral}^{(l)}(\boldsymbol{H}^{(l-1)}; \boldsymbol{V}, \boldsymbol{\lambda}) = \boldsymbol{V}s_\zeta^{(l)}\left(\hat{g}_\vartheta^{(l)}(\boldsymbol{\lambda}) \odot \left[\boldsymbol{V}^\top f_\theta^{(l)}(\boldsymbol{H}^{(l-1)})\right]\right) \qquad (4)$$

We achieve permutation equivariance via sign equivariance $s_\zeta(\boldsymbol{S} \odot \boldsymbol{X}) = \boldsymbol{S} \odot s_\zeta(\boldsymbol{X}), \forall \boldsymbol{S} \in \{-1, 1\}^{k\times d^{(l)}}$, combined with a permutation equivariance $s_\zeta(\boldsymbol{P}\boldsymbol{X}) = \boldsymbol{P}s_\zeta(\boldsymbol{X}), \boldsymbol{P} \in \mathcal{P}_k$, where $\mathcal{P}_k$ is the set of all $k \times k$ permutation matrices. Specifically, we stack linear mappings $\boldsymbol{W}_s \in \mathbb{R}^{d^{(l)}\times d^{(l)}}$ (*without bias*) with a gated nonlinearity $\phi(\hat{\boldsymbol{H}}) = \hat{\boldsymbol{H}} \odot \sigma(\vec{1}[\boldsymbol{m}^\top\boldsymbol{W}_a + \boldsymbol{b}_a^\top])$ with sigmoid $\sigma$, column-wise norm $m_j = \|\hat{\boldsymbol{H}}_{:,j}\|$, and learnable $\boldsymbol{W}_a \in \mathbb{R}^{d^{(l)}\times d^{(l)}}$ as well as $\boldsymbol{b}_a \in \mathbb{R}^{d^{(l)}}$.

### 3.2.3 Directed Graphs

Directed graphs are an important topic that did not discuss so far. For S²GNNs to generalize the capabilities of non-local sequence models like transformers (Vaswani et al., 2017) or SSMs (Poli et al., 2023; Gu & Dao, 2023) it is vital to support direction, e.g., for distinguishing source/beginning and sink/end. However, all discussion before assumed the existence of the eigenvalue *decomposition* of $\boldsymbol{L}$. This was the case for symmetric $\boldsymbol{L}$; however, for directed graphs, $\boldsymbol{L}$ may be asymmetric.

To guarantee $\boldsymbol{L}$ is diagonalizable with real eigenvalues, we use the Magnetic Laplacian (Forman, 1993; Shubin, 1994; De Verdière, 2013) which is Hermitian and models direction in the complex domain: $\boldsymbol{L}_q = \boldsymbol{I} - (\boldsymbol{D}_s^{-1/2}\boldsymbol{A}_s\boldsymbol{D}_s^{-1/2}) \odot \exp[i2\pi q(\boldsymbol{A} - \boldsymbol{A}^\top)]$ with symmetrized adjacency/degrees $\boldsymbol{A}_s/\boldsymbol{D}_s$, potential $q \in [0, 2\pi]$, element-wise exponential $\exp$, and imaginary unit $i^2 = -1$. While other parametrizations of a Hermitian matrix are also possible, with $\boldsymbol{A} \in \{0,1\}^{n \times n}$ and appropriate choice of $q$, $\boldsymbol{L}_q : \{0,1\}^{n \times n} \to \mathbb{C}^{n \times n}$ is *injective*. In other words, every possible asymmetric $\boldsymbol{A}$ maps to exactly one $\boldsymbol{L}_q$ and, thus, this representation is lossless. Moreover, for sufficiently small potential $q$, the order of eigenvalues is well-behaved (Furutani et al., 2020). In contrast to Koke & Cremers (2024), a Hermitian parametrization of spectral filters does not require a dedicated propagation for forward and backward information flow. For simplicity we choose $q < 1/n_{\max}$ with maximal number of nodes $n_{\max}$ (with binary $\boldsymbol{A}$). This choice ensures that the first eigenvector suffices to obtain, e.g., the topological sorts of a Directed Acyclic Graph (DAG). Due to the real eigenvalues of a Hermitian matrix, the presented content generalizes with minor adjustments. Most notably, we use a feature transformation $f_\theta^{(l)} : \mathbb{R}^{n \times d^{(l-1)}} \to \mathbb{C}^{n \times d^{(l)}}$ and map back into the real domain after the spectral convolution. We give more implementation details in § J.6 and provide additional background on directed graphs in § C.

### 3.2.4 Efficient Yet Stable and Expressive Positional Encodings

The availability of the partial eigendecomposition allows for their *dual use* for positional encodings at negligible cost. Motivated by this, we propose the first efficient ($\mathcal{O}(km)$) and (fully) permutation equivariant spectral Positional Encodings PE that provably increase the expressivity strictly beyond the 1-Weisfeiler-Leman (1-WL) test (Xu et al., 2019; Morris et al., 2019). In contrast to the Laplacian encodings of Dwivedi & Bresson (2021), our PE do not require augmenting eigenvectors w.r.t. their sign and maintain permutation equivariance also in the presence of repeated eigenvalues. In comparison to Huang et al. (2024), our PE come with drastically lower computational cost and have no learnable parameters. Due to the absence of learnable parameters, we need to calculate our PE only once.

We construct our $k$-dimensional positional encodings $\text{PE}(\boldsymbol{V}, \boldsymbol{\lambda}) \in \mathbb{R}^{n \times k}$ as

$$\text{PE}(\boldsymbol{V}, \boldsymbol{\lambda}) = \|_{j=1}^k [(\boldsymbol{V}\hat{h}_j(\boldsymbol{\lambda})\boldsymbol{V}^\top) \odot \boldsymbol{A}] \cdot \vec{1} \tag{5}$$

with concatenation $\|$ and binary adjacency $\boldsymbol{A} \in \{0,1\}^{n \times n}$. We use a Radial Basis Function (RBF) filter with normalization around each eigenvalue $\hat{h}_j(\boldsymbol{\lambda}) = \text{softmax}((\lambda_j - \boldsymbol{\lambda}) \odot (\lambda_j - \boldsymbol{\lambda})/\sigma^2)$ with small width $\sigma \in \mathbb{R}_{>0}$. This parametrization is not only permutation equivariant but also stable according to the subsequent definition via the Hölder continuity. Note that $C$ depends on the eigengap between $1/(\lambda_{k+1} - \lambda_k)$ at the frequency cutoff (for exact constant $C$ see proof in § H.5).

**Definition 1** (Stable PE). *(Huang et al., 2024) A PE method* $\text{PE} : \mathbb{R}^{n \times k} \times \mathbb{R}^k \to \mathbb{R}^{n \times k}$ *is called stable, if there exist constants* $c, C > 0$*, such that for any Laplacian* $\boldsymbol{L}, \boldsymbol{L}'$*, and* $\boldsymbol{P}_* = \arg\min_{\boldsymbol{P}} \|\boldsymbol{L} - \boldsymbol{P}\boldsymbol{L}'\boldsymbol{P}^\top\|_{\text{F}}$

$$\|\text{PE}(\text{EVD}(\boldsymbol{L})) - \boldsymbol{P}_* \,\text{PE}\,(\text{EVD}\,(\boldsymbol{L}'))\|_{\text{F}} \leq C \cdot \|\boldsymbol{L} - \boldsymbol{P}_*\boldsymbol{L}'\boldsymbol{P}_*^\top\|_{\text{F}}^c. \tag{6}$$

**Theorem 5.** *The Positional Encodings* PE *in Eq. 5 are stable according to Definition 1.*

Next to their stability, our PE can discriminate certain degree-regular graphs (e.g., Fig. 9). Since degree-regular graphs cannot be distinguished by 1-WL, our PE makes the equipped GNN (as expressive as 1-WL) strictly more expressive than 1-WL. See § I for continued expressivity analyses.

**Theorem 6.** *S²GNNs are strictly more expressive than 1-WL with the* PE *of Eq. 5.*

(a)

(b)

(c)
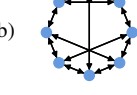
(d)

(e)
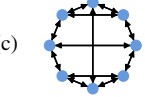

Figure 9: PE discriminates the depicted degree-regular graphs, except for (a) vs. (c).

# 4 Empirical Results

With state-of-the-art performance on the peptides-func task of the long-range benchmark (Dwivedi et al., 2022), plus strong results on further benchmarks, we demonstrate that $S^2GCN$, *a GCN paired with spectral filters*, is highly capable of **modeling long-range interactions (§ 4.1)**. We assess $S^2$GNNs' **long sequence performance (§ 4.2)** (mechanistic in-context learning) and show that $S^2$GCN, a graph machine learning method, can achieve competitive results to state-of-the-art sequence models, including H3, Hyena, and transformers. We exemplify $S^2$GNNs' practicality and competitiveness at scale on **large-scale benchmarks (§ 4.3)** like TPUGraphs (Phothilimthana et al., 2023), PCQM4Mv2 (Hu et al., 2021), and Open Graph Benchmark (OGB) Products (Hu et al., 2020). Further, in § M.8, we report state-of-the-art performance on the heterophilic arXiv-year (Lim et al., 2021) and, in § M.4, we study combinations of spatial and spectral filters beyond Eq. 1 & 2.

**Setup.** We pair different MPGNNs with spectral filters and name the composition S²<base>. For example, a $S^2$GNN with GAT as base will be called $S^2$GAT. We typically perform 3 to 10 random reruns and report the mean $\pm$ standard deviation. The experiments of § 4.1 require <11 GB (e.g. Nvidia GTX 1080Ti); for the experiments in § 4.2 & 4.3 we use a 40 GB A100. We usually optimize weights with AdamW (Loshchilov & Hutter, 2019) and cosine annealing scheduler (Loshchilov & Hutter, 2017). We use early stopping based on the validation loss/score. See § M for more details and https://www.cs.cit.tum.de/daml/s2gnn for code as well as supplementary material.

## 4.1 Long-Range Interactions

**Finding (I): S²GCN outperforms state-of-the-art graph transformers, MPGNNs, and graph rewirings** on the peptides-func long-range benchmarks (Dwivedi et al., 2022) by a substantial margin. Simultaneously, we remain approximately 35% below the 500k parameter threshold and. On peptides-struct we are only outperformed by NBA-GIN (Park et al., 2023). We extend the best configuration for a GCN of Tönshoff et al. (2023) (see GCN in Table 1), lower the number of message passing steps from six to three, and interleave spatial and spectral filters (Eq. 2) with $\lambda_{cut} = 0.7$.

**Dataset contribution: Clustering**, given a single seed node per cluster, measures the ability (1) to spread information within the cluster and (2) to discriminate between the clusters. We complement the semi-supervised task CLUSTER from Dwivedi et al. (2023) with **(our)** LR-CLUSTER dataset, a scaled-up version with long-range interactions (1). We closely follow Dwivedi et al. (2023), but instead of using graphs sampled from Stochastic Block Models (SBMs), we sample coordinates from a Gaussian Mixture Model (GMM) and then connect nearby nodes. CLUSTER has 117 nodes on average, while ours has 896. LR-CLUSTER has an average diameter of $\approx 33$ and often contain hub nodes that cause over-squashing. For full details on the dataset construction, see § M.6.

Table 1: Long-range benchmark. Our $S^2$GNN uses $\approx 35\%$ fewer parameters than the other models. AP is Peptides-func's and MAE peptides-struct's target metric. The best/second best is bold/underlined.

| | Model | peptides-func ($\uparrow$) | peptides-struct ($\downarrow$) |
|---|---|---|---|
| **Transformer** | TIGT (Choi et al., 2024) | $0.6679 \pm 0.0074$ | $0.2485 \pm 0.0015$ |
| | MGT+WPE (Ngo et al., 2023) | $0.6817 \pm 0.0064$ | $0.2453 \pm 0.0025$ |
| | G.MLPMixer (He et al., 2023) | $0.6921 \pm 0.0054$ | $0.2475 \pm 0.0015$ |
| | Graph ViT (He et al., 2023) | $0.6942 \pm 0.0075$ | $0.2449 \pm 0.0016$ |
| | GRIT (Ma et al., 2023) | $0.6988 \pm 0.0082$ | $0.2460 \pm 0.0012$ |
| | GPS+HDSE (Luo et al., 2024) | $0.7156 \pm 0.0058$ | $0.2457 \pm 0.0013$ |
| **Rewiring:** DRew-GCN (Gutteridge et al., 2023) | | $0.7150 \pm 0.0044$ | $0.2536 \pm 0.0015$ |
| **State Space Models:** Graph Mamba (Behrouz & Hashemi, 2024) | | $0.7071 \pm 0.0083$ | $0.2473 \pm 0.0025$ |
| | GRED (Behrouz & Hashemi, 2024) | $0.7133 \pm 0.0011$ | $0.2455 \pm 0.0013$ |
| **GNN** | PathNN (Michel et al., 2023) | $0.6816 \pm 0.0026$ | $0.2545 \pm 0.0032$ |
| | CIN++ (Giusti et al., 2023) | $0.6569 \pm 0.0117$ | $0.2523 \pm 0.0013$ |
| | NBA-GIN (Park et al., 2023) | $0.7071 \pm 0.0067$ | $\mathbf{0.2424 \pm 0.0010}$ |
| | GCN (Tönshoff et al., 2023) | $0.6860 \pm 0.0050$ | $0.2460 \pm 0.0007$ |
| | $S^2GCN$ **(ours)** | $\underline{0.7275 \pm 0.0066}$ | $0.2467 \pm 0.0019$ |
| | + PE **(ours)** | $\mathbf{0.7311 \pm 0.0066}$ | $\underline{0.2447 \pm 0.0032}$ |

**Dataset contribution: Distance regression** is a task with long-range interactions used in prior work (Geisler et al., 2023; Lim et al., 2023). Here, the regression targets are the shortest path distances to the only root node (in-degree 0). We generate random trees/DAGs with $\approx 750$ # of nodes on average (details are in § M.7). The target distances often exceed 30 hops. We evaluate on similarly sized graphs as in the training data, i.e., in-distribution (**ID**) samples, and out-of-distribution (**OOD**) samples that consist of slightly larger graphs. Details on the dataset construction are in § M.7.

**Finding (II): spatial MPGNNs are less effective as S²GNNs**, for long-range interactions. This is evident for peptides Table 1, clustering Fig. 10, distance regression Fig. 11, and over-squashing Fig. 12. Specifically, if the task requires long-range interactions beyond the receptive field of MPGNNs, they return crude estimates. E.g., in Fig. 11, the MPGNN predicts (approx.) constantly 20 for all distances beyond its receptive field – roughly the mean in the training data. Moreover,

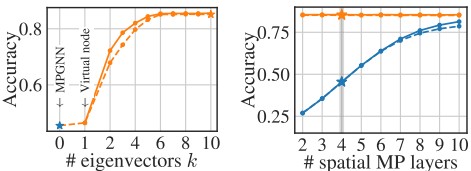
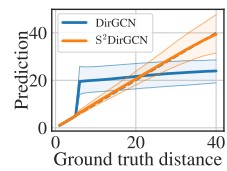
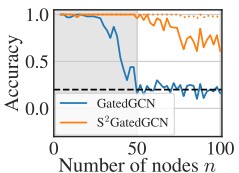

(a) 4+1 layer MP + Spec. (b) w/ one vs. w/o Spec.

Figure 10: Results on `LR-CLUSTER`. Solid lines are w/, dashed lines are w/o our PE (§ 3.2.4).

Figure 11: 90% pred. intervals on OOD DAGs.

Figure 12: Over-sq.: 25-layer GatedGCN vs. 1-layer spec. ID is grey.

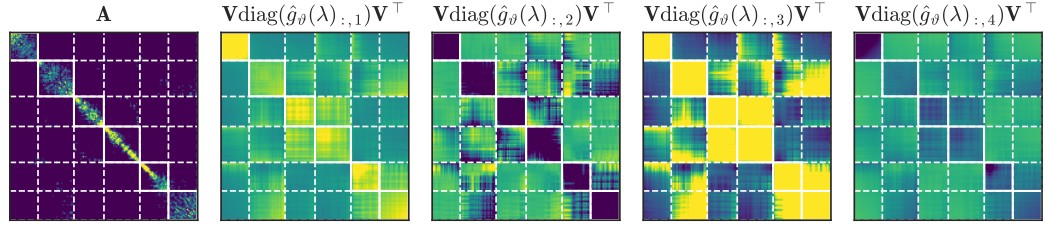

Figure 13: 4 filters on `LR-CLUSTER`. Large/small entries are yellow/blue, white lines mark clusters.

$S^2$GNNs may converge faster (see Fig. 25 in § M.6.2) and are more parameter-efficient, as we show on PCQM4Mv2 (Hu et al., 2021) in § M.9.

**Finding (III): virtual nodes are insufficient.** We frequently find that including more than a single eigenvector ($k > 1$) yields substantial gains. We make this explicit in Fig. 10a, where we append a single spectral layer and sweep over the number of eigenvectors $k$. We complement these findings with an ablation for the frequency cutoff $\lambda_{\text{cut}}$ on peptides-func in § M.5.

**Finding (IV): our Positional Encodings** PE **consistently help**, when concatenated to the node features. While this finding is true throughout our evaluation, the differences are more pronounced in certain situations. For example, on `LR-CLUSTER` in Fig. 10, the PE help with spectral filter and a small $k$ or without spectral filter and many message passing steps.

**Finding (V): spectral filters align with clusters**, as we illustrate in Fig. 13 for four arbitrary spectral filters learned on `LR-CLUSTER`. We observe that (a) the spectral filters reflect the true clustering structure, (b) some filters are smooth while others contain details, and (c) they model coarser or finer cluster structures (e.g., first vs. third filter).

### 4.2 Sequence Modelling: Mechanistic In-Context Learning

Following the evaluation of Hyena (Poli et al., 2023) and H3 (Fu et al., 2023), we benchmark $S^2$GCN with sequence models on the *associative recall* in-context learning task, stemming from mechanistic interpretability (Elhage et al., 2021; Power et al., 2022; Zhang et al., 2023; Olsson et al., 2022). In associative recall, the model is asked to retrieve the value for a key given in a sequence. For example, in the sequence `a,0,e,b,z,9,h,2,=>,z`, the target is the value for key `z`, which is 9 since it follows `z` in its prior occurrences. We create a sequence/path graph with a node for each "token" (separated by "," in the example above) and label the target node with its value. We assess the performance of $S^2$GCN on graphs that vary in size by almost two orders of magnitude and follow Poli et al. (2023) with a vocabulary of 30 tokens. Moreover, we finetune our $S^2$GCN on up to 30k nodes.

Table 2: 30k token associative recall.

| Model | Accuracy ($\uparrow$) |
|---|---|
| Transformer (Vaswani et al., 2017) | *OOM* |
| w/ FlashAttention (Dao et al., 2022) | 0.324 |
| H3 (Fu et al., 2023) | 0.084 |
| Hyena (Poli et al., 2023) | **1.000** |
| $S^2$GCN (**ours**) | $0.97 \pm 0.05$ |

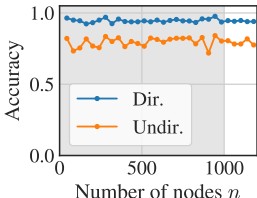

Figure 14: $S^2$GCN solves associative recall for sequences varying in size by two orders of magnitude. Grey area marks **ID**.

**Finding (VI): our spectral filter for directed are effective** and may improve generalization, as we find in Fig. 14 (and Table 13 of § M.7).

**Finding (VII): $S^2$GCN a state-of-the-art sequence model**, as it performs on par with Hyena and, here, outperforms transformers (Table 2).

### 4.3 Large-Scale Benchmarks

**Finding (VIII): $S^2$GNNs is practical and scalable.** We demonstrate this on the OGB Products graph (2.5 mio. nodes, Table 3) and the (directed) 10 million graphs dataset TPUGraphs (average number of nodes $\approx 10{,}000$, Table 4). In both cases, we find full-graph training (without segment training (Cao et al., 2023)) using 3 (Dir-) GCN layers interlayered with spectral filters, a reasonable configuration on a 40 GB A100. However, for OGB Products, we find that batching is superior, presumably because the training nodes are drawn from a "small" region of the graph (see § K).

**The cost of partial** EVD for each dataset (excluding TPUGraphs and distance regression) is between 1 to 30 minutes on CPUs. We report the detailed costs of EVD and experiments in § M.3.

Table 3: OGB Products.

| Split | Model | Accuracy (↑) | F1 (↑) |
|---|---|---|---|
| Train | GAT | 0.866±0.001 | 0.381±0.001 |
| | S²GAT | **0.902±0.000** | **0.472±0.006** |
| Val | GAT | 0.907±0.001 | 0.508±0.002 |
| | S²GAT | **0.913±0.002** | **0.582±0.014** |
| Test | GAT | 0.798±0.003 | 0.347±0.004 |
| | S²GAT | **0.811±0.007** | **0.381±0.009** |

Table 4: Graph ranking on TPUGraphs "layout".

| Model | Kendall tau (↑) |
|---|---|
| GCN | 60.25 |
| S²GCN | **63.62** |

## 5 Related Work

**Combining spatial and spectral filters** has recently attracted attention outside of the graph domain in models like Hyena (Poli et al., 2023), Spectral State Space Models (Agarwal et al., 2024), etc. with different flavors of parametrizing the global/FFT convolution. Nevertheless, the properties of spatial and spectral filter parametrization (e.g., local vs. global) are well-established in classical signal processing. A combination of spectral and spatial filters was applied to (periodic) molecular point clouds (Kosmala et al., 2023). For GNNs, Stachenfeld et al. (2020) compose a spatial and spectral message passing but do not handle the ambiguity of the eigendecomposition and, thus, do not maintain permutation equivariance. Moreover, Beaini et al. (2021) use the EVD for localized anisotropic graph filters; Liao et al. (2019) propose an approach that combines spatial and spectral convolution via the Lanczos algorithm; and Huang et al. (2022) augment message passing with power iterations. Behrouz & Hashemi (2024) apply a Mamba-like state space model to graphs via arbitrarily ordering the nodes and, thus, sacrifice permutation equivariance.

**Long-range interactions on graphs.** Works that model long-range interactions can be categorized into: (a) MPGNNs on rewired graphs (Gasteiger et al., 2019a,b; Gutteridge et al., 2023); (b) higher-order GNNs (Fey et al., 2020; Wollschläger et al., 2024) that, e.g., may pass information to distant nodes through hierarchical message passing schemes; and (c) message passing adaptations to facilitate long-range interactions. For example, Park et al. (2023) propose "non-backtracking" message passing, Errica et al. (2024) adaptively choose the numbers of message passing steps, and Ding et al. (2024) use linear RNNs to aggregate over each node's neighborhoods. While approaches (a-c) can increase the receptive field of GNNs, they are typically still spatially bounded. In contrast, (d) alternative architectures, like graph transformers (Ma et al., 2023; Dwivedi & Bresson, 2021; Kreuzer et al., 2021; Rampášek et al., 2022; Geisler et al., 2023; Deng et al., 2024) with global attention, may model all possible $n \times n$ interactions. We provide notes on the limitations of graph transformers with absolute positional encodings in § D, which highlights the importance of capturing the relative relationships between nodes, as $S^2$GNNs do. Moreover, in a recent/contemporary non-attention model for all pair-wise interactions, Batatia et al. (2024) use a resolvent parametrization of matrix functions relying on the LDL factorization of a matrix, but do not characterize their approximation-theoretic properties, over-squashing, expressivity on graphs, nor how to deal with directed graphs.

In § B, we discuss additional related work w.r.t. expressivity and directed graphs.

## 6 Discussion

We propose $S^2$GNNs, adept at efficiently modeling complex long-range interactions via the synergistic composition of spatially and spectrally parametrized filters (§ 3). We show that $S^2$GNNs share many properties with graph rewirings, pooling, and hierarchical message passing schemes (Fig. 3 & 4). $S^2$GNNs outperform the aforementioned techniques with a substantial margin on the peptides long-range benchmark (§ 4.1), and we show that $S^2$GNNs are also strong sequence models, performing on par or outperforming state-of-the-art like Hyena or H3 in our evaluation (§ 4.2). Even though we find global graph models, like $S^2$GNNs, more prone to overfitting (see § K/L for further limitations/impact), moving to global models aligns with the trend for other deep learning domains.

## Acknowledgments and Disclosure of Funding

We want to express our gratitude to Nicholas Gao for his feedback and the discussions about modeling choices. Moreover, we thank Leo Schwinn and Tim Beyer for their helpful and on-point feedback and suggestions.

This research was supported by the Helmholtz Association under the joint research school "Munich School for Data Science - MUDS", as well as by the Munich Data Science Institute (MDSI) via the Linde/MDSI Doctoral Fellowship program and the MDSI Seed Fund.

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

# Appendix

## A Notation

Table 5: List of most important symbols used in this work (with the most general domain).

| | |
|---|---|
| $b$ | Scalar |
| $\boldsymbol{b}$ | (column) Vector |
| $\boldsymbol{B}$ | Matrix |
| $\mathbb{B}$ | Set |
| $\boldsymbol{B}^\top$ | Transpose of matrix $\boldsymbol{B}$ |
| $\boldsymbol{B}^{\mathrm{H}}$ | Conjugate transpose of matrix $\boldsymbol{B}$ |
| $i, i^2 = -1$ | Complex number |
| $\odot$ | Element-wise multiplication |
| $\circ$ | Function composition, i.e., $f_2 \circ f_1 = f_1(f_2(\cdot))$ |
| $\|\cdot\|_F$ | Frobenius norm |
| $\mathcal{O}$ | "Big O" notation for asymptotic growth of function |
| $\boldsymbol{P} \in \mathbb{P}$ | Permutation matrix |
| $*_{\mathcal{G}}$ | Graph convolution (see § 2) |
| $(\mathcal{G}_p)_{p \in \mathbb{N}}$ | Graph sequence (cf. Theorem 3) |
| $\mathcal{G}(\boldsymbol{A}, \boldsymbol{X})$ | Graph |
| $\boldsymbol{A} \in \mathbb{R}_{\geq 0}^{n \times n}$ | Adjacency matrix |
| $\boldsymbol{X} \in \mathbb{R}^{n \times d}$ | Node features |
| $n$ | Number of nodes |
| $m$ | Number of edges |
| $d$ | Number of attributes |
| $\boldsymbol{L}_q \in \mathbb{C}^{n \times n}$ | Graph/combinatorial/random walk (Magnetic) Laplacian |
| $q \in \mathbb{R}_{\geq 0}$ | Potential (see Eq. 7) |
| $\boldsymbol{L}_{q=0} = \boldsymbol{L} \in \mathbb{R}^{n \times n}$ | Real-valued Laplacian ($q = 0$) |
| $\boldsymbol{\lambda}, \boldsymbol{V} = \mathrm{EVD}(\boldsymbol{L})$ | Eigendecomposition s.t. $\boldsymbol{L} = \boldsymbol{V} \operatorname{diag}(\boldsymbol{\lambda}) \boldsymbol{L}^{\mathrm{H}}$ |
| $\boldsymbol{\lambda}, \boldsymbol{V} = \mathrm{EVD}(\boldsymbol{L}, k)$ | Partial eigendecomposition containing $\lambda_1 \leq \lambda_2 \leq \cdots \leq \lambda_k \quad (\leq \lambda_{k+1})$ |
| $k$ | Number of eigenvectors |
| $p$ | Polynomial order |
| $p_{\mathrm{cut}}$ | Receptive field size (number hops) of polynomial filter |
| $p_{\max}$ | Maximal diameter of graph |
| $\lambda_{\mathrm{cut}}$ | Eigenvalue threshold considered in spectral filter |
| $\lambda_{\max}$ | Maximal eigenvalue of graph |
| $\ell$ | Number of layers |
| $K_r(\hat{g}, \lambda_{\mathrm{cut}})$ | Bound on $r$-th derivative of spectral filter on interval $[\lambda_{\mathrm{cut}}, 2]$ |
| $\mathrm{Spectral}^{(l)}(\boldsymbol{H}^{(l-1)}; \boldsymbol{V}, \boldsymbol{\lambda})$ | Spectral(ly parametrized) filter |
| $\mathrm{Spatial}^{(l)}(\boldsymbol{H}^{(l-1)}; \boldsymbol{A})$ | Spatial message passing |
| $g$ | Filter in graph convolution |
| $g_{\boldsymbol{\gamma}_p}$ | Order-$p$ polynomial filter in graph convolution, with coefficients $\boldsymbol{\gamma}_p$ |
| $\hat{g}$ | Filter in graph convolution, in spectral domain |
| $\hat{g}(\boldsymbol{\lambda})$ | as function evaluated at eigenvalues |
| $\hat{g}_\vartheta(\boldsymbol{\lambda})$ | as parametrized function evaluated at eigenvalues |
| $\hat{g}_\vartheta^{(l)}(\boldsymbol{\lambda})$ | as parametrized function at layer $l$, evaluated at eigenvalues |
| $\mathrm{PE}(\boldsymbol{V}, \boldsymbol{\lambda})$ | Positional encodings |
| $f_\theta$ | Feature transformation in spatial domain |
| $s_\zeta$ | Feature transformation in spectral domain |

## B Related Work for Expressivity and Directed Graphs

**Expressivity.** Laplacian eigenvectors have been used previously to construct positional encodings that improve the expressivity of GNNs or Transformers (Lim et al., 2023; Wang et al., 2022; Geisler et al., 2023; Huang et al., 2024). Our positional encodings are similar to the preprocessing of Balcilar et al. (2021a), where the authors design an edge-level encoding/mask to surpass 1-WL. The hierarchy of Weisfeiler-Leman (WL) tests is a common way to categorize the expressivity of GNNs (Grohe

et al., 2021). Xu et al. (2019) showed that most MPGNNs are bound by or as strong as the 1-WL test. Lim et al. (2023) point out that spectral GNNs suffer from similar limitations as MPGNNs w.r.t. their expressivity. Generally, the development of expressive GNNs is an active research direction, and we refer to Li & Leskovec (2022) for a broad overview.

**Directed graphs.** Rossi et al. (2023) also extend the WL test to directed graphs and propose an MPGNN for directed graphs. How to model direction in graphs is also still an open question and various approaches were proposed (Battaglia et al., 2018; Tong et al., 2020; Zhang et al., 2021; Rossi et al., 2023; Koke & Cremers, 2024). We utilize a Hermitian Laplacian for direction awareness, namely the Magnetic Laplacian, which was also used by Zhang et al. (2021) for an MPGNN and Geisler et al. (2023) for positional encodings.

## C  Background for Directed Graphs

**Undirected vs. directed graphs.** For spatial filtering, it is straightforward to plausibly extend the message passing (e.g. Battaglia et al. (2018); Rossi et al. (2023)). However, the spectral motivation and spectral filter on directed graphs require more care. The eigendecomposition is guaranteed to exist for real symmetric matrices. Real symmetric matrices are always diagonalizable, and the eigenvectors will then span a complete orthogonal basis to represent all possible signals $\boldsymbol{X} \in \mathbb{R}^{n \times d}$. Note that some non-symmetric square matrices are also diagonalizable and, thus, also have an eigendecomposition, albeit the eigenvectors may not be orthogonal. Thus, further consideration is required to generalize the graph Laplacian to general directed graphs.

**Magnetic Laplacian.** For the spectral filter on directed graphs, we build upon a direction-aware generalization, called Magnetic Laplacian (Forman, 1993; Shubin, 1994; De Verdière, 2013; Furutani et al., 2020; Geisler et al., 2023)

$$\boldsymbol{L}_q = \boldsymbol{I} - (\boldsymbol{D}_s^{-1/2} \boldsymbol{A}_s \boldsymbol{D}_s^{-1/2}) \odot \exp[i2\pi q(\boldsymbol{A} - \boldsymbol{A}^\top)] \tag{7}$$

where $\boldsymbol{A}_s = \boldsymbol{A} \vee \boldsymbol{A}^\top$ is the symmetrized graph with diagonal degree matrix $\boldsymbol{D}_s$. $\odot$ denotes the element-wise product, $\exp$ the element-wise exponential, $i = \sqrt{-1}$ an imaginary number, and $q$ the potential (hyperparameter). By construction $\boldsymbol{L}_q$ is a Hermitian matrix $\boldsymbol{L}_q = \boldsymbol{L}_q^{\mathrm{H}}$ where the conjugate transpose is equal to $\boldsymbol{L}_q$ itself. Importantly, Hermitian matrices naturally generalize real symmetric matrices and have a well-defined eigendecomposition $\boldsymbol{L}_q = \boldsymbol{V} \boldsymbol{\Lambda} \boldsymbol{V}^{\mathrm{H}}$ with real eigenvalues $\boldsymbol{\Lambda}$ and unitary eigenvectors $\boldsymbol{V} \boldsymbol{V}^{\mathrm{H}} = I$. For appropriate choices of the potential $q$, the order of eigenvalues is well-behaved (Furutani et al., 2020). Recently Geisler et al. (2023) demonstrated the efficacy of these eigenvectors for positional encodings for transformers. Moreover, the Magnetic Laplacian was used for a spectrally designed spatial MPGNN (Zhang et al., 2021), extending Defferrard et al. (2017). Due to the real eigenvalues, one could, in principle, also apply a monomial basis (Chien et al., 2021), or different polynomial bases stemming from approximation theory (He et al., 2021; Wang & Zhang, 2022; He et al., 2022; Guo & Wei, 2023).

To see why Eq. 7 describes an injection for appropriate choices of $q$, consider that the sparsity pattern of $\boldsymbol{L}_q$ matches $\boldsymbol{A}$ up to the main diagonal. If $\boldsymbol{A}$ contains a self-loop the main diagonal will have a 0 instead of 1 entry at the self-loop location. $\boldsymbol{A} - \boldsymbol{A}^\top$ can be directly inferred from the phase $\exp[i2\pi q(\boldsymbol{A} - \boldsymbol{A}^\top)]$, assuming that $q < 1/(2 \max_{u,v} A_{u,v})$. Thus, it is solely left to obtain $\boldsymbol{A}_s$ from $\boldsymbol{I} - \boldsymbol{D}_s^{-1/2} \boldsymbol{A}_s \boldsymbol{D}_s^{-1/2}$, which is trivial for a binary adjacency but more involved for real-valued weights. Determining if and when $\boldsymbol{L}_q$ is injective for real-valued $\boldsymbol{A}$ is left for future work.

**Properties of the eigendecomposition.** The eigendecomposition is not unique, and thus, one should consider the result of the eigensolver arbitrary in that regard. One ambiguity becomes apparent from the definition of an eigenvalue itself $\boldsymbol{L}\boldsymbol{v} = \lambda \boldsymbol{v}$ since one can multiply both sides of the equation with a scalar $c \in \mathbb{C} \setminus \{0\}$: $\boldsymbol{L}(c\boldsymbol{v}) = \lambda(c\boldsymbol{v})$. We already implicitly normalized the magnitude of the eigenvectors $\boldsymbol{V}$ by choosing them to be orthogonal ($\boldsymbol{V}\boldsymbol{V}^\top = \boldsymbol{I}$) or unitary ($\boldsymbol{V}\boldsymbol{V}^{\mathrm{H}} = \boldsymbol{I}$). Thus, after this normalization, $c$ only represents an arbitrary sign for real-valued eigenvectors or a rotation on the unit circle in the complex case. Another reason for ambiguity occurs in the case of repeated / multiple eigenvalues (e.g., $\lambda_u = \lambda_v$ for $u \neq v$). In this case, the eigensolver may return an arbitrary set of orthogonal eigenvectors chosen from the corresponding eigenspace.

## D  Limitations of Graph Transformers Using Absolute Positional Encodings

Here, we consider a vanilla graph transformer $f(\boldsymbol{X})$ that solely becomes structure-aware due to the addition (or concatenation) of positional encodings: $f(\boldsymbol{X} + \mathrm{PE}(\boldsymbol{A}))$. The main point we are going to demonstrate is that a vanilla transformer with such absolute positional encodings $\mathrm{PE}(\boldsymbol{A}) \in \mathbb{R}^{n \times d}$ will be limited in its expressivity if the positional encodings are permutation equivariant $\boldsymbol{P} \, \mathrm{PE}(\boldsymbol{A}) = \mathrm{PE}(\boldsymbol{PAP}^\top)$ w.r.t. any $n \times n$ permutation matrix $\boldsymbol{P} \in \mathcal{P}$.

The limitation particularly arises in the presence of automorphisms $\boldsymbol{P}_a \boldsymbol{A} \boldsymbol{P}_a^\top = \boldsymbol{A}$ with specifically chosen permutation $\boldsymbol{P}_a$. To be more specific, assume that nodes $u$ and $v$ are automorphic to each other. That is, there exists a $\boldsymbol{P}_a$ that will swap the order of $u$ and $v$ (among other nodes) s.t. $\boldsymbol{P}_a \boldsymbol{A} \boldsymbol{P}_a^\top = \boldsymbol{A}$. By permutation equivariance, we know $\boldsymbol{P}_a \, \mathrm{PE}(\boldsymbol{A}) = \mathrm{PE}(\boldsymbol{P}_a \boldsymbol{A} \boldsymbol{P}_a^\top) = \mathrm{PE}(\boldsymbol{A})$ and, hence, $\mathrm{PE}(\boldsymbol{A})_u = \mathrm{PE}(\boldsymbol{A})_v$.

We have just shown that automorphic nodes will have the same positional encodings PE if the positional encodings are permutation equivariant. This implies that permutation equivariant positional encodings PE are not even able to capture simple neighboring relationships. For example, consider an undirected sequence/path graph o-o-o-o-o with five nodes. Here, the two end nodes, which we also all first and last node, are automorphic. So are the second and second-last nodes. Assuming the second and second last nodes have different node features (e.g., A-B-C-D-A), that breaks the symmetry, it is still not possible for a transformer with absolute positional encodings to tell the first and last node apart. In other words, in the example, the transformer cannot tell apart the end node with neighboring feature B from the end node with neighboring feature D. This shows a severe limitation of architectures without additional components capturing the relative distances (e.g., as $S^2$GNNs can). This concern is not as critical for architectures where the positional encodings are not entirely permutation equivariant (Dwivedi & Bresson, 2021; Kreuzer et al., 2021), with relative positional encodings (Ma et al., 2023), and might also be of lesser concern for directed graphs (Geisler et al., 2023).

## E  $S^2$GNN Generalizes a Virtual Node

Adding a fully connected virtual node (Gilmer et al., 2017) is among the simplest ways to add the ability for long-range information exchange. An equivalent method was proposed as a simple over-squashing remedy in the seminal work by Alon & Yahav (2020). A single Spectral layer amounts to a type of virtual nodes in the special case of $f_\theta = \boldsymbol{I}$ and

$$\hat{g}^{(l)}(\lambda) = \begin{cases} 1 \text{ for } \lambda = 0, \\ 0 \text{ for } \lambda > 0, \end{cases} \tag{8}$$

Assuming a simply-connected graph $\mathcal{G}$, the unique normalized zero-eigenvector $\boldsymbol{v}$ of the symmetrically-normalized graph Laplacian $\boldsymbol{L} = \boldsymbol{I} - \boldsymbol{D}^{-1/2} \boldsymbol{A} \boldsymbol{D}^{-1/2}$ has components $\boldsymbol{v}_u = \sqrt{\frac{d_u}{2|E|}}$, where $d_u$ denotes the degree of node $u \in \mathcal{G}$, and $|E|$ the number of edges in the graph. At node $u \in \mathcal{G}$, we therefore find

$$\mathrm{Spectral}_u^{(l)}(\boldsymbol{H}^{(l-1)}; \boldsymbol{V}, \boldsymbol{\lambda}) = \frac{\sqrt{d_u}}{2|E|} \sum_{v \in \mathcal{G}} \sqrt{d_v} \boldsymbol{h}_v^{(l-1)} \tag{9}$$

with $\boldsymbol{h}_v^{(l-1)}$ denoting the row of $\boldsymbol{H}^{(l-1)}$ corresponding to node $v \in \mathcal{G}$. In other words, filtering out the zero-frequency component of the signal means scattering a global, degree-weighted embedding average to all nodes of the graph. For the unnormalized graph Laplacian, Eq. 9 instead becomes an unweighted average, which is consistent with the usual definition of a virtual node. We refer to Fig. 3 & 4 for additional intuition.

## F  Existing Results on Over-Squashing

We restate two key results from Di Giovanni et al. (2023a) using our notation. They imply the existance of a regime in which 1-hop MPNN architectures suffer from exponentially decaying Jacobian sensitivity. Meanwhile, $S^2$GNNs can easily learn a signal of constantly lower-bounded sensitivity, as shown by invoking its trivial subcase of a virtual node in Theorem 2.

**Theorem 7** (Adapted from Di Giovanni et al. (2023a)). *In an $\ell$-layer spatial MPGNN with message-passing matrix $\boldsymbol{S} = c_r \boldsymbol{I} + c_a \boldsymbol{A}$ ($c_r, c_a \in \mathbb{R}^+$) and a Lipschitz nonlinearity $\sigma$,*

$$\boldsymbol{H}^{(l)} = \text{Spatial}^{(l)}(\boldsymbol{H}^{(l-1)}; \boldsymbol{A}) = \sigma\left(\boldsymbol{S}\boldsymbol{H}^{(l-1)}\boldsymbol{W}^{(l-1)}\right), \, 1 \leq l \leq \ell \quad (10)$$

*the Jacobian sensitivity satisfies the following upper bound:*

$$\left\|\frac{\partial \mathbf{h}_v^{(\ell)}}{\partial \mathbf{h}_u^{(0)}}\right\|_{L^1} \leq (c_\sigma w d)^\ell \left(\boldsymbol{S}^\ell\right)_{vu}, \quad (11)$$

*with $\mathbf{h}_u^{(0)}$, $\mathbf{h}_v^{(\ell)}$ denoting the rows of $\boldsymbol{H}^{(0)}$, $\boldsymbol{H}^{(\ell)}$ corresponding to the nodes $v, u \in \mathcal{G}$, $c_\sigma$ the Lipschitz constant of the nonlinearity, $w$ the maximum entry value over all weight matrices $\boldsymbol{W}^{(l)}$, and $d$ the network width.*

The dependence of the upper bound on the matrix power $\left(\boldsymbol{S}^\ell\right)_{vu}$ – not generally present for S²GNN by Theorem 2 – leads to a topology-dependence which becomes explicit in the following theorem. It concerns the typical shallow-diameter regime, in which the number $\ell$ of MPGNN layers is comparable to the graph diameter.

**Theorem 8** (Adapted from Di Giovanni et al. (2023a)). *Given an MPNN as in Eq. 10, with $c_a \leq 1$, let $v, u \in \mathcal{G}$ be at distance $r$. Let $c_\sigma$ be the Lipschitz constant of $\sigma$, $w$ the maximal entry-value overall weight matrices, $d_{\min}$ the minimal degree of $\mathcal{G}$, and $\gamma_\ell(v, u)$ the number of walks from $v$ to $u$ of maximal length $\ell$. For any $0 \leq k < r$, there exists $C_k > 0$ **independent** of $r$ and of the graph, such that*

$$\left\|\frac{\partial \mathbf{h}_v^{(r+k)}}{\partial \mathbf{h}_u^{(0)}}\right\|_{L^1} \leq C_k \gamma_{r+k}(v, u) \left(\frac{2c_\sigma w d}{d_{\min}}\right)^r.$$

For 1-hop MPGNNs with $2c_\sigma w d < d_{\min}$, we therefore identify an exponential decay of sensitivity with node distance $r$ in the weak-connectivity limit for which $\gamma_{r+k}(v, u)$ increases sub-exponentially with $r$. As Di Giovanni et al. (2023a) point out, sharper bounds can be derived under graph-specific information about $(\boldsymbol{S}^r)_{vu}$.

## G    Construction of an explicit ground truth filter

We express the electric potential along a periodic sequence of screened 1D charges as a convolution of a corresponding graph signal with a consistently defined graph filter. This closed-form example underscores our default use of a low-pass window for the spectral part of S²GNNs by showing how a continuous problem with a convolutional structure and quickly flattening spectral response (typical for pair interactions in physics and chemistry) discretizes into a graph problem with similar features.

The approach exploits the surjective mapping of Fourier modes on $[0, n]$ onto the Laplacian eigenvectors of a cycle graph $C_n$. We consider two corresponding representations of the same problem:

$$\rho(x) = \sum_{l=0}^{n-1} q_l \Delta_n(x - l), \; \Delta_m(x) = \sum_{m \in \mathbb{Z}} \delta(x - mn), \; V(x) = (\phi_\sigma *_{\mathbb{R}} \rho)(x), \quad (12)$$

$$\phi_\sigma(x) = \left(x \text{erf}\left(\frac{x}{\sqrt{2}\sigma}\right) + \sigma\sqrt{\frac{2}{\pi}} \exp\left(-\frac{x^2}{2\sigma^2}\right)\right) - |x|, \quad \sigma > 0$$

$$\left[V(l) = (\phi_\sigma *_{\mathbb{R}} \rho)(l) \stackrel{!}{=} (g_\sigma *_{\mathcal{G}} \boldsymbol{q})_l, \, 0 \leq k \leq n-1, \, \forall \boldsymbol{q} \in \mathbb{R}^n\right] \qquad \Updownarrow \quad (13)$$

$$\mathcal{G} = \mathcal{C}_n, \quad \boldsymbol{q} = (q_1, \ldots, q_n)^\top \quad (14)$$

- A continuous representation (Eq. 12) in terms of a 1D distribution $\rho$ of $n$ point charges $q_1, \ldots, q_n$ and their periodically repeating image charges, written as a sum of Dirac combs at equidistant offsets $l$ with $0 \leq l \leq n-1$, interacting via the potential profile $\phi_\sigma$ obtained from solving in Gauss' law of electrostatics for a 1D point charge screened by a Gaussian cloud of opposite background charge with standard deviation $\sigma$. The screening ensures

convergence to a finite potential and its exact form is insignificant (we choose the Gaussian-type screening due to its analytical tractability). Note that $\phi_\sigma(x) \simeq \text{const.} - |x|$ for $x \to 0$ (the unscreened 1D potential in the direction normal to an infinitely wide capacitor plate), while the screening guarantees an exponential dropoff to zero as $x \to \infty$,

- A graph representation (Eq. 14) by placing the $n$ charges $q_1, \ldots, q_n$ onto a cycle graph $\mathcal{C}_n$.

We derive the graph filter $g_\sigma$ from a consistency condition (Eq. 13) between both representations: the graph convolution $(g_\sigma *_{\mathcal{G}} \boldsymbol{q})$ has to yield the electric potential $V$ sampled at the charge loci if we want $g_\sigma$ to act like the continuous convolution kernel $\phi_\sigma$ in the discrete graph picture.

The Fourier transform of $\phi_\sigma$ (in the convention without integral prefactor and with a $2\pi$ frequency prefactor) reads $\hat{\phi}_\sigma(\kappa) = \frac{1}{\pi\kappa^2}\left(1 - \exp\left(-\frac{1}{2}\sigma^2\kappa^2\right)\right)$. For the density, the Poisson sum formula gives $\hat{\rho}(\kappa) = \sum_{k=0}^{n-1} \frac{1}{\sqrt{n}}\hat{q}_k \Delta_1\left(\kappa - \frac{k}{n}\right)$ with $\hat{q}_k = \frac{1}{\sqrt{n}}\sum_{j=0}^{n-1} q_i \exp\left(-i2\pi\frac{k}{n}j\right)$. The coefficients $\hat{q}_k$ are precisely the components of the graph Fourier transform of $\boldsymbol{q}$ (physically, they amount for the structure factor). By the convolution theorem, $\hat{V}(\kappa) = \hat{\phi}_\sigma(\kappa)\hat{\rho}(\kappa)$. By noting that all integer-shifted frequencies in the Dirac combs $\Delta_1\left(\cdot - \frac{k}{n}\right)$ (or all Brillouin zones, in physics terminology) yield the same phase $\exp\left(i2\pi\frac{k}{n}l\right)$ if we only sample $V(x)$ at the charge loci $x = l$, $0 \le l \le n - 1$, we can write $V(l) = \frac{1}{2\pi}\sum_{k=0}^{n-1}\hat{q}_k\left(\sum_{m\in\mathbb{Z}}\hat{\phi}_\sigma\left(\frac{k}{n} + m\right)\right)\frac{1}{\sqrt{n}}\exp\left(i2\pi\frac{k}{n}l\right)$. Through pattern-matching with the consistency condition of Eq. 13, we can therefore identify that the graph filter is a sum over Brillouin zones, $(\hat{g}_\sigma)(\lambda_k) = \frac{1}{2\pi}\sum_{m\in\mathbb{Z}}\hat{\phi}_\sigma\left(\frac{k}{n} + m\right)$, where $\lambda_k$ denotes the eigenvalues of the normalized $\mathcal{C}_n$ graph Laplacian, $\lambda_k = 1 - \cos\left(\frac{2\pi k}{n}\right)$. To fulfill this relation for all $n$, $k$ we set

$$\hat{g}_\sigma(\lambda) = \frac{1}{2\pi}\sum_{m\in\mathbb{Z}}\hat{\phi}_\sigma\left(\frac{1}{2\pi}\arccos(1 - \lambda) + m\right)$$

We claim now (and prove in a later paragraph) that for $\lambda > \lambda_0 > 0$ and a sufficiently large choice $\sigma > \sigma(r, \lambda_0)$, the absolute $r$-th derivative satisfies the upper bound $\left|\frac{d^r}{d\lambda^r}\hat{g}_\sigma(\lambda)\right| \le \left|\frac{d^r}{d\lambda^r}\hat{g}_\infty(\lambda)\right|$, where we can think of $\hat{g}_\infty$ as the limit of taking $\sigma \to \infty$ (i.e., a constant background charge):

$$\hat{g}_\infty(\lambda) = \frac{1}{2\pi}\sum_{m\in\mathbb{Z}}\hat{\phi}_\infty\left(\frac{1}{2\pi}\arccos(1 - \lambda) + m\right), \quad \hat{\phi}_\infty(\kappa) = \frac{1}{\pi\kappa^2}$$

The merit of this is that unlike the screened $\hat{g}_\sigma(\lambda)$, $\hat{g}_\infty(\lambda)$ can be solved analytically to find closed-form bounds on the absolute derivatives $\left|\frac{d^r}{d\lambda^r}\hat{g}_\sigma(\lambda_0)\right|$. By invoking the sum expansion form of the trigamma function $\Psi_1(z) = \sum_{m=0}^{\infty}\frac{1}{(z+m)^2}$, the reflection identity $\psi_1(1 - z) + \psi_1(z) = \frac{\pi^2}{\sin^2\pi z}$, and the half-angle formula $\sin^2\left(\frac{x}{2}\right) = \frac{1-\cos(x)}{2}$, we find

$$\hat{g}_\infty(\lambda) = \frac{1}{2\pi^2}\left(\Psi_1\left(\frac{1}{2\pi}\arccos(1 - \lambda)\right) + \Psi_1\left(1 - \frac{1}{2\pi}\arccos(1 - \lambda)\right)\right)$$
$$= \frac{1}{2\sin^2\left(\frac{1}{2}\arccos(1 - \lambda)\right)} = \frac{1}{\lambda},$$

a remarkably simple result. We can now readily evaluate $\left|\frac{d^r}{d\lambda^r}\hat{g}_\infty(\lambda)\right| = \frac{r!}{\lambda^{r+1}}$, but it remains to prove that this upper-bounds $\left|\frac{d^r}{d\lambda^r}\hat{g}_\sigma(\lambda)\right|$ for any $\lambda > \lambda_0 > 0$ and sufficiently large $\sigma > \sigma(r, \lambda_0)$. For compactness, define the expressions $z(\lambda) := \frac{1}{2\pi}\arccos(1 - \lambda) \in \left[0, \frac{1}{2}\right]$ (strictly increasing in

$\lambda$), $y_\sigma(z) := \exp\left(-\frac{1}{2}\sigma^2 z^2\right)$, and $\tilde{z} = 1 - z \geq z$. Consider the series of "term-by-term" derivatives

$$\frac{d}{dz}\hat{g}_\sigma(\lambda(z)) = -\frac{1}{\pi^2}\sum_{m=0}^{\infty}\left(\frac{1}{(z+n)^3}(1 - y_\sigma(z+m)) - \frac{1}{(\tilde{z}+n)^3}(1 - y_\sigma(\tilde{z}+m))\right)$$

$$+ \sum_{m=0}^{\infty}\mathcal{O}\left(y_\sigma(m)\right)$$

$$\frac{d^2}{dz^2}\hat{g}_\sigma(\lambda(z)) = \frac{3}{\pi^2}\sum_{m=0}^{\infty}\left(\frac{1}{(z+n)^4}(1 - y_\sigma(z+m)) + \frac{1}{(\tilde{z}+n)^4}(1 - y_\sigma(\tilde{z}+m))\right)$$

$$+ \sum_{m=0}^{\infty}\mathcal{O}\left(y_\sigma(m)\right)$$

$$\vdots$$

They converge uniformly on $\left[0, \frac{1}{2}\right]$ as they clearly are Cauchy sequences under uniform bound (moreover, well-definedness in $z = 0$ follows by applying l'Hospital's rule – physically, this is the merit provided by including Gaussian screening in our model). Therefore, they indeed converge to the respective derivatives $\frac{d^r}{dz^r}\hat{g}_\sigma(\lambda(z))$ (justifying the above notation). The same holds for the corresponding series for $\frac{d^r}{dz^r}\hat{g}_\infty(\lambda(z))$: they are not defined in $z = 0$, but otherwise still converge as they match the known series expansion of the polygamma function. Given $\lambda_0 > 0$ and thus $z(\lambda_0) > 0$, taking $\sigma$ larger than some $\sigma(r, \lambda_0)$ guarantees that $\frac{d^r}{dz^r}\hat{g}_\sigma(\lambda(z))$ and $\frac{d^r}{dz^r}\hat{g}_\infty(\lambda(z))$ are of the same sign for $\lambda > \lambda_0$ ($z(\lambda) > z(\lambda_0)$). This holds for all orders $r \in \mathbb{N}$ since we see by induction that the product rule always yields one term analogous to the first respective terms above, and otherwise only terms of $\mathcal{O}\left(y_\sigma(m)\right)$. Then, observing that $0 \leq 1 - y_\sigma(x) < 1 \ \forall \ x \geq 0$ and $\tilde{z} \geq z$ implies $|\frac{d^r}{dz^r}\hat{g}_\sigma(\lambda_0(z_0))| \leq |\frac{d^r}{dz^r}\hat{g}_\infty(\lambda_0(z_0))|$. The same must hold for the $\lambda$-derivatives by the chain rule.

One interesting question is whether $\hat{g}_\sigma$ is also $C$-integral-Lipschitz for some constant $C > 0$. We discuss this stability-informed criterion (Gama et al., 2020) in the main body as a domain-agnostic prior assumption about the "ideal" graph filter if no other ground truth knowledge informing additional smoothness bounds (such as here) is available. While the above bound is too loose to certify this directly ($|\frac{d}{d\lambda}\hat{g}_\sigma(\lambda)| \leq C\lambda^{-1}$ would be needed), integral-Lipschitzness under some constant follows from the fact that $|\frac{d}{d\lambda}\hat{g}_\sigma(\lambda)|$ is bounded on $[0, 2]$: by the uniform convergence of the term-by-term derivative series, it is continuous. Well-definedness of the product $\frac{d}{dz}\hat{g}_\sigma\frac{dz}{d\lambda}$ has to be checked in $\lambda = 0$, where it follows by continuous extension using l'Hospital's rule. As a continuous function defined on a compact interval, $|\frac{d}{d\lambda}\hat{g}_\sigma|$ assumes a maximum.

## H Proofs

### H.1 Proof of Theorem 1

We next prove the permutation equivariance of the spectral filter in Eq. 3:

**Theorem 1.** Spectral($\boldsymbol{H}^{(l-1)}$; EVD($\boldsymbol{L}, k$)) *of Eq. 3 is equivariant to all $n \times n$ permutation matrices $\boldsymbol{P} \in \mathcal{P}$:* Spectral($\boldsymbol{P}\boldsymbol{H}^{(l-1)}$; EVD($\boldsymbol{P}\boldsymbol{L}\boldsymbol{P}^\top, k$)) $= \boldsymbol{P}$ Spectral($\boldsymbol{H}^{(l-1)}$; EVD($\boldsymbol{L}, k$)).

for the general case of parametrizing a Hermitian "Laplacian" $\boldsymbol{L} \in \mathbb{C}^{n \times n}, \boldsymbol{L}^{\mathrm{H}} = \boldsymbol{L}$. Note that this proof does not rely in any means on the specifics of $\boldsymbol{L}$, solely that the eigendecomposition exists $\boldsymbol{L} = \boldsymbol{V}\boldsymbol{\Lambda}\boldsymbol{V}^{\mathrm{H}}$ with unitary eigenvectors $\boldsymbol{V}\boldsymbol{V}^{\mathrm{H}} = \boldsymbol{I}$. For practical reasons, it is suitable to define $\boldsymbol{L}(\boldsymbol{A})$ as a function of $\boldsymbol{A}$. A similar proof for real-valued eigenvectors is given by (Lim et al., 2023). The specific spectral filter we consider is

$$\mathrm{Spectral}(\boldsymbol{H}^{(l-1)}; \boldsymbol{V}, \boldsymbol{\lambda}) = h\left[\boldsymbol{V}\left(\hat{g}(\boldsymbol{\lambda}) \odot \left[\boldsymbol{V}^{\mathrm{H}}f(\boldsymbol{H}^{(l-1)})\right]\right)\right] \tag{15}$$

with arbitrary $f : \mathbb{C}^{d_1} \to \mathbb{C}^{d_2}$, applied row-wise to $\boldsymbol{H}^{(l-1)} \in \mathbb{C}^{n \times d_1}$. Analogously, $h : \mathbb{C}^{d_2} \to \mathbb{C}^{d_3}$ is applied row-wise. We choose complex functions to emphasize generality, although we restrict

Spectral to real in- and outputs in all experiments. The graph filter is defined as element-wise function $\hat{g}_{u,v}(\boldsymbol{\lambda}) := \hat{g}_v(\lambda_u, \{\lambda_1, \lambda_2, \ldots, \lambda_k\})$ that depends on the specific eigenvalue $\lambda$ and potentially the set of eigenvalues $\{\lambda_1, \lambda_2, \ldots, \lambda_k\}$ (or its vector representation $\boldsymbol{\lambda}$) of the partial eigendecomposition.

We need to make sure that the partial decomposition includes all eigenvalues of the same magnitude, i.e., $\lambda_u \neq \lambda_{u'}, \forall u \in \{1, 2, \ldots, k\}, u' \in \{k+1, k+2, \ldots, n\}$. In practice, this is achieved by choosing large enough $k$ to accommodate all eigenvalues $\lambda_{\text{cut}} < \lambda_{k+1}$, or by dropping trailing eigenvalues where $\lambda_j = \lambda_{k+1}$ for $j \in \{1, 2, \ldots, k\}$. Generally, it is also not important that we consider the $k$ *smallest* eigenvalues in the spectral filter. We only need to ensure that the spectral filter is either calculated on all or no eigenvalues/-vectors of an eigenspace.

*Proof.* Assuming functions $\phi(\boldsymbol{X})$ and $\psi(\boldsymbol{X})$ are permutation equivariant, then $\phi(\psi(\boldsymbol{X}))$ is permutation equivariant $\phi(\psi(\boldsymbol{PX})) = \phi(\boldsymbol{P}\psi(\boldsymbol{X})) = \boldsymbol{P}\phi(\psi(\boldsymbol{X}))$ for any $n \times n$ permutation $\boldsymbol{P} \in \mathcal{P}$. Thus, it suffices to prove permutation equivariance for $h, f, \boldsymbol{V}(\hat{g}(\boldsymbol{\lambda}) \odot [\boldsymbol{V}^{\text{H}}\boldsymbol{X}])$ independently, where $\boldsymbol{X} \in \mathbb{C}^{n \times d_2}$.

Regardless of the complex image and domain of $h$ and $f$, they are permutation equivariant since they are applied row-wise

$$f(\boldsymbol{X}) = [f(\boldsymbol{X}_1) \quad f(\boldsymbol{X}_2) \quad \ldots \quad f(\boldsymbol{X}_n)]^{\text{H}}$$

and reordering the rows in $\boldsymbol{X} \in \mathbb{C}^{n \times d_1}$ also reorders the outputs: $f(\boldsymbol{PX}) = \boldsymbol{P}f(\boldsymbol{X})$.

For finalizing the proof of permutation equivariance, we first rearrange $\boldsymbol{Y} = \boldsymbol{V}(\hat{g}(\boldsymbol{\lambda}) \odot [\boldsymbol{V}^{\text{H}}\boldsymbol{X}]) = \sum_{u=1}^{k} \boldsymbol{v}_u(\hat{g}_{u,:}(\lambda_u) \odot [\boldsymbol{v}_u^{\text{H}}\boldsymbol{X}])$ and $\boldsymbol{Y}_{:,v} = \sum_{u=1}^{k} \hat{g}_{u,v}(\lambda_u)\boldsymbol{v}_u\boldsymbol{v}_u^{\text{H}}\boldsymbol{X}_{:,v}$.

This construction (a) is invariant to the ambiguity that every eigenvector $\boldsymbol{v}_u$ can be arbitrarily rotated $c_u\boldsymbol{v}_u$ by $\{c_u \in \mathbb{C} \,|\, |c_u| = 1\}$. That is, $(c_u\boldsymbol{v}_u)(c_u\boldsymbol{v}_u)^{\text{H}} = c_u\bar{c}_u\boldsymbol{v}_u\boldsymbol{v}_u^{\text{H}} = \boldsymbol{v}_u\boldsymbol{v}_u^{\text{H}}$.

Moreover, (b) in the case of $j$ repeated eigenvalues $\{s+1, s+2, \ldots, s+j\}$ where $\lambda_{s+1} = \lambda_{s+2} = \cdots = \lambda_{s+j}$, we can choose a set of orthogonal eigenvectors arbitrarily rotated/reflected from the $j$-dimensional eigenspace (basis symmetry). The given set of eigenvectors can be arbitrarily transformed $\boldsymbol{V}_{:,s+1:s+j}\boldsymbol{\Gamma}_j$ by a matrix chosen from the unitary group $\boldsymbol{\Gamma}_j \in U(j)$. Since

$$\sum_{u=s}^{s+j} \hat{g}_{u,v}(\lambda_u)\boldsymbol{v}_u\boldsymbol{v}_u^{\text{H}}\boldsymbol{X}_{:,v} = \hat{g}_{s,v}(\lambda_s)\left[\sum_{u=s}^{s+j} \boldsymbol{v}_u\boldsymbol{v}_u^{\text{H}}\right]\boldsymbol{X}_{:,v} = \hat{g}_{s,v}(\lambda_s)\left[\boldsymbol{V}_{:,s+1:s+j}\boldsymbol{V}_{:,s+1:s+j}^{\text{H}}\right]\boldsymbol{X}_{:,v}$$

we simply need to show that the expression is invariant to a transformation by $\boldsymbol{\Gamma}_j$:

$$\boldsymbol{V}_{:,s+1:s+j}\boldsymbol{\Gamma}_j(\boldsymbol{V}_{:,s+1:s+j}\boldsymbol{\Gamma}_j)^{\text{H}} = \boldsymbol{V}_{:,s+1:s+j}\boldsymbol{\Gamma}_j\boldsymbol{\Gamma}_j^{\text{H}}\boldsymbol{V}_{:,s+1:s+j}^{\text{H}} = \boldsymbol{V}_{:,s+1:s+j}\boldsymbol{V}_{:,s+1:s+j}^{\text{H}}$$

To see why $\boldsymbol{\Gamma}_j \in U(j)$ is a sufficient choice in the light of repeated/multiple eigenvalues, consider the defintion of eigenvalues/vectors

$$\boldsymbol{L}\boldsymbol{V}_{:,s+1:s+j} = \boldsymbol{L}\begin{bmatrix} | & | & & | \\ \boldsymbol{v}_{s+1} & \boldsymbol{v}_{s+2} & \ldots & \boldsymbol{v}_{s+j} \\ | & | & & | \end{bmatrix}$$

$$= \begin{bmatrix} | & | & & | \\ \boldsymbol{v}_{s+1} & \boldsymbol{v}_{s+2} & \ldots & \boldsymbol{v}_{s+j} \\ | & | & & | \end{bmatrix}\begin{bmatrix} \lambda_{s+1} & 0 & \ldots & 0 \\ 0 & \lambda_{s+2} & \ldots & 0 \\ \vdots & \vdots & \ddots & \vdots \\ 0 & 0 & \ldots & \lambda_{s+j} \end{bmatrix}$$

$$= \lambda_{s+1}\begin{bmatrix} | & | & & | \\ \boldsymbol{v}_{s+1} & \boldsymbol{v}_{s+2} & \ldots & \boldsymbol{v}_{s+j} \\ | & | & & | \end{bmatrix}$$

$$= \lambda_{s+1}\boldsymbol{V}_{:,s+1:s+j}$$

we can now multiply both sides from the right with an arbitrary matrix $\boldsymbol{B} \in \mathbb{C}^{j \times j}$. To preserve the unitary property $\boldsymbol{V}_{:,s+1:s+j}\boldsymbol{V}_{:,s+1:s+j}^{\text{H}} = \boldsymbol{I}$, we require $(\boldsymbol{V}_{:,s+1:s+j}\boldsymbol{B})(\boldsymbol{V}_{:,s+1:s+j}\boldsymbol{B})^{\text{H}} = \boldsymbol{I}$. Thus, the eigenvectors can be arbitrarily transformed by $\boldsymbol{\Gamma}_j \in U(j)$ instead of $\boldsymbol{B} \in \mathbb{C}^{j \times j}$.

This concludes the proof. $\square$

## H.2 Proof of Theorem 2

We restate Theorem 2 in more detail and also considering graphs that contain multiple connected components. The unchanged bottom line is that S²GNNs can express signals lower-bounded by a constant that is unaffected by local properties of the graph topology, instead of suffering from exponential sensitivity decay like spatial MPGNNs.

**Theorem** (Theorem 2, formal). *Consider an $\ell$-layer $S^2$GNN of the form Eq. 1. Let $(\tilde{\vartheta}, \vartheta, \theta)$ be parameters of the spatial GNN, spectral filters $\hat{g}^{(l)}_{\vartheta}$, and feature transformation $f_\theta$. Assume the existence of parameters $\tilde{\vartheta}$ such that $\mathrm{Spatial}^{(l)}(\boldsymbol{H}^{(l-1)}; \boldsymbol{A}, \tilde{\vartheta}) = 0 \; \forall 1 \leq l \leq \ell$ and $\theta$ such that $f_\theta = \boldsymbol{I}$. Then, a filter choice $\vartheta$ exists such that the $\ell$-layer $S^2$GNN of the additive form Eq. 1 can express a signal $\mathbf{h}^{(\ell)}_v(\boldsymbol{H}^{(0)}; \tilde{\vartheta}, \vartheta, \theta)$ with uniformly lower-bounded Jacobian sensitivity,*

$$\left\| \frac{\partial \mathbf{h}^{(\ell)}_v(\boldsymbol{H}^{(0)}; \tilde{\vartheta}, \vartheta, \theta)}{\partial \mathbf{h}^{(0)}_u} \right\|_{L^1} \geq \begin{cases} \frac{d K^{\ell}_{\vartheta}}{2|E_C|} & \text{if } u, v \text{ connected,} \\ 0 & \text{otherwise,} \end{cases} \tag{16}$$

*with $\boldsymbol{h}^{(0)}_u$, $\boldsymbol{h}^{(\ell)}_v$ denoting the rows of $\boldsymbol{H}^{(0)}$, $\boldsymbol{H}^{(\ell)}$ corresponding to the nodes $u \neq v \in \mathcal{G}$, connected component $\mathcal{C} \subset \mathcal{G}$ containing $|E_C|$ edges, network width $d$ and parameter-dependent constant $K_\vartheta$.*

*Proof.* Choose $\tilde{\vartheta}$ such that $\mathrm{Spatial}^{(l)}(\boldsymbol{H}^{(l-1)}; \boldsymbol{A}) = 0 \; \forall 1 \leq l \leq \ell$ (typically by setting all weights and biases to zero), $\theta$ such that $f_\theta = \boldsymbol{I}$, and set $\vartheta$ such that

$$\hat{g}^{(l)}_k(\lambda; \vartheta) = K_\vartheta \begin{cases} 1 \text{ for } \lambda = 0, \\ 0 \text{ for } \lambda > 0, \end{cases} \quad \forall 1 \leq l \leq \ell, \; 1 \leq k \leq d \tag{17}$$

for some $K_\vartheta > 0$. This choice of filter parameters $\vartheta$ lets $\mathrm{Spectral}$ act like a type of virtual node across all hidden dimensions $k$: In the standard orthonormal basis of the 0-eigenspace given by

$$\left( \boldsymbol{v}^{(\mathcal{C})} \right)_u = \sqrt{\frac{d_u}{2|E_\mathcal{C}|}} \begin{cases} 1 \text{ for } u \in \mathcal{C}, \\ 0 \text{ else,} \end{cases} \tag{18}$$

where $\mathcal{C}$ enumerates all connected components, and $d_u$ denotes the degree of node $u$, we find

$$\begin{aligned} \mathbf{h}^{(\ell)}_v(\boldsymbol{H}^{(0)}; \tilde{\vartheta}, \vartheta, \theta) &= \left( \mathrm{Spectral}^{(\ell)} \circ \cdots \circ \mathrm{Spectral}^{(0)} \right)_v (\boldsymbol{H}^{(0)}; \boldsymbol{V}, \boldsymbol{\lambda}) \\ &= \frac{K^{\ell}_{\vartheta} \sqrt{d_v}}{2|E_{\mathcal{C}^{(v)}}|} \sum_{u \in \mathcal{C}^{(v)}} \sqrt{d_u} \mathbf{h}^{(0)}_u, \end{aligned} \tag{19}$$

with $\mathcal{C}^{(v)}$ denoting the connected component containing $v$. Particularly, note that applying the spectral layer more than once does not affect the result since the projector onto an eigenvector is idempotent (up to $K_\vartheta$). The result must also hold in any other orthonormal basis of the 0-eigenspace due to the invariance of $\mathrm{Spectral}$ under orthogonal eigenbasis transformations. Differentiating with respect to $\mathbf{h}^{(0)}_u$, taking the $L^1$ norm and using $\sqrt{d_u d_v} \geq 1$ shows the statement. $\qquad\square$

## H.3 Proof of Theorem 3

**Theorem 3.** *Let $\hat{g}$ be a discontinuous spectral filter. For any approximating sequence $(g_{\gamma_p})_{p \in \mathbb{N}}$ of polynomial filters, an adversarial sequence $(\mathcal{G}_p)_{p \in \mathbb{N}}$ of input graphs exists such that*

$$\nexists \alpha \in \mathbb{R}_{>0} : \sup_{0 \neq \boldsymbol{X} \in \mathbb{R}^{|\mathcal{G}_p| \times d}} \frac{\|(g_{\gamma_p} - g) *_{\mathcal{G}_p} \boldsymbol{X}\|_{\mathrm{F}}}{\|\boldsymbol{X}\|_{\mathrm{F}}} = \mathcal{O}\left(p^{-\alpha}\right)$$

The proof makes use of a result by S. Bernstein (Natanson, 1964):

**Theorem 9** (Bernstein). *Let $f \colon [0, 2\pi] \to \mathbb{C}$ be a $2\pi$-periodic function. Then $f$ is $\alpha$-Hölder continuous for some $\alpha \in (0, 1)$ if, for every $p \in \mathbb{N}$, there exists a degree-$p$ trigonometric polynomial $T_p(x) = a_0 + \sum_{j=1}^{p} a_j \cos(jx) + \sum_{j=1}^{p} b_j \sin(jx)$ with coefficients $a_j, b_j \in \mathbb{C}$, such that*

$$\sup_{0 \leq x \leq 2\pi} |f(x) - T_p(x)| \leq \frac{C(f)}{p^\alpha}$$

*where C(f) is a positive number depending on $f$.*

*Proof.* Given a discontinuous filter $\hat{g}\colon [0,2] \to \mathbb{R}$, construct the function $f\colon [0,2\pi] \to \mathbb{C}$ fulfilling the prerequisites of Theorem 9 by pre-composing $f := \hat{g} \circ (\cos(\cdot) + 1)$. We proceed via contradiction. Suppose that there is an $\alpha \in (0,1)$ and a sequence of degree-$p$ polynomial filters, $\hat{g}_{\boldsymbol{\gamma}_p}(\lambda) = \sum_{j=0}^{p} \gamma_j \lambda^j$, $\boldsymbol{\gamma} = (\gamma_0, \ldots, \gamma_p)^\top \in \mathbb{R}^{p+1}$, such that $\|\hat{g}_{\boldsymbol{\gamma}_p} - \hat{g}\|_\infty = \mathcal{O}(p^{-\alpha})$. Then, the sequence of trigonometric polynomials $T_p := \hat{g}_{\boldsymbol{\gamma}_p} \circ (\cos(\cdot) + 1)$ fulfills the condition of Theorem 9. This would imply that $f = \hat{g} \circ (\cos(\cdot) + 1)$ is $\alpha$-Hölder continuous, meaning that a constant $K > 0$ exists such that

$$|\hat{g}(\cos(x) + 1) - \hat{g}(\cos(y) + 1)| \leq K|x - y|^\alpha \ \forall x, y \in [0, 2\pi]$$

Considering $\lambda_0 \in [0,2]$, $\lambda \to \lambda_0$ and $x = \arccos(\lambda_0 - 1)$, $y = \arccos(\lambda - 1)$ (using the arccos branch in which both $\lambda_0$, $\lambda$ eventually end up) shows a contradiction to the assumed discontinuity of $\hat{g}$. Therefore, no polynomial filter sequence $\left(\hat{g}_{\boldsymbol{\gamma}_p}\right)_{p\in\mathbb{N}}$ together with an $\alpha \in (0,1)$ exist such that $\|\hat{g}_{\boldsymbol{\gamma}_p} - \hat{g}\|_\infty = \mathcal{O}(p^{-\alpha})$. In particular, for any sequence $\left(\hat{g}_{\boldsymbol{\gamma}_p}\right)_{p\in\mathbb{N}}$, a sequence of adversarial values $(\lambda_p)_{p\in\mathbb{N}}$, $\lambda_p \in [0,2]$ exists such that

$$\nexists \alpha \in (0,1)\colon \ |\hat{g}_{\boldsymbol{\gamma}_p}(\lambda_p) - \hat{g}(\lambda_p)| = \mathcal{O}(p^{-\alpha})$$

The proof is finished if we can find a sequence of graphs $(\mathcal{G}_p)$ such that the symmetrically-normalized graph Laplacian $\boldsymbol{L}_p$ of $\mathcal{G}_p$ contains $\lambda_p$ as an eigenvalue. In this case, we can construct adversarial input signals $\boldsymbol{X}_p$ on the graphs $\mathcal{G}_p$ by setting the first embedding channel to an eigenvector corresponding to $\lambda_p$, and the remaining channels to zero, such that $\left(g_{\boldsymbol{\gamma}_p} - g\right) *_{\mathcal{G}_p} \boldsymbol{X}_p = |\hat{g}_{\boldsymbol{\gamma}_p}(\lambda_p) - \hat{g}(\lambda_p)|\boldsymbol{X}_p$. In particular, it then holds that

$$\nexists \alpha \in \mathbb{R}^+\colon \ \sup_{0 \neq \boldsymbol{X} \in \mathbb{R}^{|\mathcal{G}_p| \times d}} \frac{\|(g_{\boldsymbol{\gamma}_p} - g) *_{\mathcal{G}_p} \boldsymbol{X}\|_{\mathrm{F}}}{\|\boldsymbol{X}\|_{\mathrm{F}}} = \mathcal{O}\left(p^{-\alpha}\right)$$

If we assume only simple graphs, such a construction is unfortunately not possible since the set of all simple graphs and therefore the set of all realizable eigenvalues is countable, whereas the adversarial values $\lambda_p$ could lie anywhere in the uncountable set $[0,2]$. We can, however realize arbitrary eigenvalues by using weighted graphs with three nodes. Consider a cyclic graph structure and tune the weight of edge $(1,2)$ to $\sin^2(\theta_p)$ and the weight of edges $(2,3)$ and $(3,1)$ to $\cos^2(\theta_p)$ with $\theta_p \in \left[0, \frac{\pi}{2}\right]$. The symmetrically-normalized graph Laplacian,

$$\boldsymbol{L}_p = \begin{pmatrix} 1 & -\cos^2(\theta_p) & -\sin^2(\theta_p) \\ -\cos^2(\theta_p) & 1 & -\sin^2(\theta_p) \\ -\sin^2(\theta_p) & -\sin^2(\theta_p) & 1 \end{pmatrix},$$

has eigenvalues $\lambda_p^{(1)} = 1$, $\lambda_p^{(2)} = \sin^2(\theta_p)$, $\lambda_p^{(3)} = 2 - \sin^2(\theta_p)$. $\lambda_p^{(2)}$ can assume all values $\lambda_p \in [0,1]$, whereas $\lambda_p^{(3)}$ can assume all values $\lambda_p \in [1,2]$. This finishes the proof. $\qquad\square$

*Remark.* If one wishes to restrict the set of possible adversarial graph sequences $(\mathcal{G}_p)_{p\in\mathbb{N}}$ to include only simple graphs, a version of Theorem 3 still holds where we restrict the assumption to filters $\hat{g}$ which are piecewise-continuous with discontinuities on a finite set of points $\mathcal{D} \subset \mathcal{S}$, where $\mathcal{S} \subset [0,2]$ denotes the countable set of eigenvalues realizable by simple graphs. This still covers a large class of filters to which order-$p$ polynomial filters can provably converge slower than any inverse root of $p$ in the operator norm, and includes the virtual node filter (discontinuous only in $\lambda = 0$) presented as an example in the main body. The proof is fully analogous up to the point of constructing $\lambda_p$. If $\lambda_p \in \mathcal{D}$, we can find a graph that realizes it exactly. Now assume $\lambda_p \notin \mathcal{D}$. We note that the set $\mathcal{S}$ is dense in $[0,2]$ (clear from considering, e.g., the cyclic graphs $\mathcal{C}_n$ with symmetrically-normalized Laplacian eigenvalues $\lambda_k = 1 - \cos\left(\frac{2\pi k}{n}\right)$). Since we assume that $\hat{g}$ and therefore also $|\hat{g}_{\boldsymbol{\gamma}_p} - \hat{g}|$ is piecewise-continuous anywhere but on $\mathcal{D} \subset \mathcal{S}$ and $\mathcal{D}$ is finite, we can find an open neighborhood $\mathcal{N}(\lambda_p)$ for any $\lambda_p \notin \mathcal{D}$ on which $\hat{g}$ is continuous. Using that $\mathcal{S}$ is dense in $[0,2]$, we find a graph sequence $\left(\tilde{\mathcal{G}}_p^{(l)}\right)_{l\in\mathbb{N}}$ with eigenvalues $\tilde{\lambda}_p^{(l)} \in \mathcal{N}(\lambda_p) \ \forall l \in \mathbb{N}$, $\left(\tilde{\lambda}_p^{(l)}\right)_{l\in\mathbb{N}} \to \lambda_p$ for which $\|\hat{g}_{\boldsymbol{\gamma}_p}(\tilde{\lambda}_p^{(l)}) - \hat{g}(\tilde{\lambda}_p^{(l)})\| \to \|\hat{g}_{\boldsymbol{\gamma}_p}(\lambda_p) - \hat{g}(\lambda_p)\|$. Therefore, by the same reasoning as in the proof of Theorem 3, we find that there can be no $\alpha \in (0,1)$ for which $\sup_{0 \neq \boldsymbol{X} \in \mathbb{R}^{|\mathcal{G}_p| \times d}} \frac{\|(g_{\boldsymbol{\gamma}_p} - g) *_{\mathcal{G}_p} \boldsymbol{X}\|_{\mathrm{F}}}{\|\boldsymbol{X}\|_{\mathrm{F}}}$ is of $\mathcal{O}\left(p^{-\alpha}\right)$.

## H.4 Proof of Theorem 4

We first introduce the setting and notation to state Theorem 4 in its general version. We study how well S$^2$GNNs can approximate "idealized" GNNs (*IGNNs*) containing $L$ graph convolution layers $1 \leq l \leq L$, each of which can express a convolution operator $g$ with *any* spectral representation $\hat{g}^{(l)} \colon [0,2] \to \mathbb{R}^{d^{(l)}}$. An IGNN layer therefore has the structure

$$\boldsymbol{\mathcal{H}}^{(l)} = \sigma\left(g^{(l)} *_{\mathcal{G}} \left[\boldsymbol{\mathcal{H}}^{(l-1)} \boldsymbol{W}^{(l)}\right]\right) = \sigma\left(\boldsymbol{V}\hat{g}^{(l)}(\boldsymbol{\lambda}) \odot [\boldsymbol{V}^\top \boldsymbol{\mathcal{H}}^{(l-1)} \boldsymbol{W}^{(l)}]\right) \tag{20}$$

with $\boldsymbol{\mathcal{H}}^{(l)} \in \mathbb{R}^{n \times D^{(l)}}$, $\boldsymbol{W}^{(l)} \in \mathbb{R}^{D^{(l)} \times D^{(l-1)}}$ and $\boldsymbol{V} \in \mathbb{R}^{n \times n}$.

We compare this to S$^2$GNNs with $\ell = (m+1)L$ layers for $m \geq 1$, in the additive form of Eq. 1,

$$\boldsymbol{H}^{(l)} = \text{Spectral}^{(l)}(\boldsymbol{H}^{(l-1)}; \boldsymbol{V}, \boldsymbol{\lambda}) + \text{Spatial}^{(l)}(\boldsymbol{H}^{(l-1)}; \boldsymbol{A}) \tag{21}$$

Each layer $1 \leq l \leq \ell$ parametrizes a spatio-spectral convolution. The spectral part satisfies Eq. 3,

$$\text{Spectral}^{(l)}(\boldsymbol{H}^{(l-1)}; \boldsymbol{V}, \boldsymbol{\lambda}) = \boldsymbol{V}\left(\hat{g}_{\vartheta}^{(l)}(\boldsymbol{\lambda}) \odot \left[\boldsymbol{V}^\top \boldsymbol{H}^{(l-1)} \boldsymbol{W}_{\text{spec}}^{(l)}\right]\right) \tag{22}$$

with embeddings $\boldsymbol{H}^{(l)} \in \mathbb{R}^{n \times d^{(l)}}$, linear feature transforms $f_\theta^{(l)} := \boldsymbol{W}_{\text{spec}}^{(l)} \in \mathbb{R}^{d^{(l)} \times d^{(l-1)}}$ and a spectral filter $\hat{g}_{\vartheta}^{(l)} \colon [0,2] \to \mathbb{R}$ that is fully supported and a universal approximator on $[0, \lambda_{\text{cut}}]$. Note we assume here that in every layer, there is only one spectral filter which gets reshaped as to act on every hidden component, whereas in practice, we relax this assumption to different filters per component, which can only be more expressive. The spatial part is a polynomial filter of the form

$$\text{Spatial}^{(l)}(\boldsymbol{H}^{(l-1)}; \boldsymbol{A}) = \sigma\left(\left[\sum_{j=0}^{p} \gamma_j^{(l)} \boldsymbol{L}^j\right] \boldsymbol{H}^{(l-1)} \boldsymbol{W}_{\text{spat}}^{(l)}\right)$$

$$= \sigma\left(\boldsymbol{V}\left(\hat{g}_{\boldsymbol{\gamma}}^{(l)}(\boldsymbol{\lambda}) \odot \left[\boldsymbol{V}^\top \boldsymbol{H}^{(l-1)} \boldsymbol{W}_{\text{spat}}^{(l)}\right]\right)\right)$$

with $\boldsymbol{W}_{\text{spat}}^{(l)} \in \mathbb{R}^{d^{(l)} \times d^{(l-1)}}$, polynomial order $p$ (fixed across layers), and a spectral representation $\hat{g}_{\boldsymbol{\gamma}}^{(l)}(\lambda) = \sum_{j=0}^{p} \gamma_j^{(l)} \lambda^j$ with coefficients $\boldsymbol{\gamma}^{(l)} = (\gamma_0^{(l)}, \dots, \gamma_p^{(l)})^\top \in \mathbb{R}^{p+1}$. We note that theorem 4 extends immediately to the case of directed graphs if the spatial part is instead a polynomial of the magnetic Laplacian (see section 3.2.3) over complex-valued embeddings like in Zhang et al. (2021).

Note that the layer-wise hidden dimensions $D^{(l)}$ vs. $d^{(l)}$ of the IGNN vs. S$^2$GNN do not have to agree except at the input layer, $d^{(0)} = D^{(0)}$ (of course, both networks receive the same input $\boldsymbol{\mathcal{H}}^{(0)} = \boldsymbol{H}^{(0)} = \boldsymbol{X}$), and at the output layer, $d^{(\ell)} = D^{(L)}$. We now state the general version of Theorem 4.

**Theorem** (Theorem 4, general)**.** *Assume an $L$-layer IGNN with filters $\hat{g}^{(l)}$ such that $\hat{g}^{(l)}\big|_{[\lambda_{cut},2]} \in C^r[\lambda_{cut}, 2]$ and $\left\|\frac{d^r}{d\lambda^r} \hat{g}^{(l)}\big|_{[\lambda_{cut},2]}\right\|_\infty \leq K_r^{max}(\lambda_{cut})$ for all $1 \leq l \leq L$. Let $\|\hat{g}^{(l)}\|_\infty \leq \|\hat{g}\|_\infty^{max}$ and $\|\boldsymbol{W}^{(l)}\|_2 \leq \|\boldsymbol{W}\|_2^{max}$ for all $1 \leq l \leq L$. Assume that $\sigma = [\,\cdot\,]_\geq$ is the ReLu function. Then,*

*(1) For a fixed polynomial order $p \geq 2$, an approximating sequence $\left(S^2GNN_m\right)_{m \in \mathbb{N}}$ of $[(m+1)L]$-layer S$^2$GNNs exists such that, for arbitrary graph sequences $(\mathcal{G}_m)_{m \in \mathbb{N}}$,*

$$\sup_{0 \neq \boldsymbol{X} \in \mathbb{R}^{|\mathcal{G}_p| \times d}} \frac{\left\|\left[(S^2GNN_m)_{\mathcal{G}_m} - (IGNN)_{\mathcal{G}_m}\right](\boldsymbol{X})\right\|_F}{\|\boldsymbol{X}\|_F}$$

$$= \mathcal{O}\left(C_L(\|\hat{g}\|_\infty^{max}, \|\boldsymbol{W}\|_2^{max}) \, K_r^{max}(\lambda_{cut}) \, (pm)^{-r}\right),$$

$$C_L(\|\hat{g}\|_\infty^{max}, \|\boldsymbol{W}\|_2^{max}) = \|\boldsymbol{W}\|_2^{max} \prod_{l=1}^{L-1} \left[\|\hat{g}\|_\infty^{max} \|\boldsymbol{W}\|_2^{max} + (\|\hat{g}\|_\infty^{max} \|\boldsymbol{W}\|_2^{max})^l\right]$$

*with a leading-order scaling constant that depends only on $r$. Here, $(\,\cdot\,)_{\mathcal{G}_m}$ denotes the instantiation of all model filters on the eigenvalues of an input graph $\mathcal{G}_m$, which maps both models onto a $\mathcal{G}_m$-dependent function $\mathbb{R}^{D^{(0)}} \to \mathbb{R}^{D^{(L)}}$.*

*(2) For fixed $m \geq 1$, an approximating sequence $\left(S^2GNN_p\right)_{p \in \mathbb{N}}$ of $[(m+1)L]$-layer S$^2$GNNs with increasing layer-wise polynomial order $p$ exists such that, for all $(\mathcal{G}_p)_{p \in \mathbb{N}}$, the same bound holds.*

*Proof.* We first prove the following lemma, narrowing down the previous theorem to a single layer.

**Lemma 1.** *Let $IGNN^{(l)}$ denote a single IGNN layer as in Eq. 20, with a filter $\hat{g}^{(l)}$ such that $\hat{g}^{(l)}\big|_{[\lambda_{cut},2]}$ is r-times continuously differentiable on $[\lambda_{cut},2]$ and satisfies a bound $K_r\left(\hat{g}^{(l)},\lambda_{cut}\right) \geq 0$, $\left|\frac{d^r}{d\lambda^r}\hat{g}^{(l)}(\lambda)\right| \leq K_r\left(\hat{g}^{(l)},\lambda_{cut}\right) \; \forall \lambda \in [\lambda_{cut},2]$. Let $\sigma = [\,\cdot\,]_{\geq}$ be the ReLu function, and let $\|\boldsymbol{W}^{(l)}\|_2$ denote the spectral norm of $\boldsymbol{W}^{(l)}$. Then,*

*(1) For fixed polynomial order $p \geq 2$, an approximating sequence $\left(S^2GNN_m^{(l)}\right)_{m\in\mathbb{N}}$ of $(m+1)$-layer $S^2GNNs$ exists such that, for arbitrary graph sequences $(\mathcal{G}_m)_{m\in\mathbb{N}}$,*

$$\sup_{0\neq\boldsymbol{X}\in\mathbb{R}^{|\mathcal{G}_p|\times d}} \frac{\left\|\left[(S^2GNN_m^{(l)})_{\mathcal{G}_m} - (IGNN^{(l)})_{\mathcal{G}_m}\right](\boldsymbol{X})\right\|_F}{\|\boldsymbol{X}\|_F} = \mathcal{O}\left([\|\boldsymbol{W}^{(l)}\|_2 K_r(\hat{g},\lambda_{cut})](pm)^{-r}\right)$$

*with a scaling constant that depends only on $r$. Here, $(\,\cdot\,)_{\mathcal{G}_m}$ denotes the instantiation of all model filters on the eigenvalues of an input graph $\mathcal{G}_m$, which maps both models onto a $\mathcal{G}_m$-dependent function $\mathbb{R}^{D^{(l-1)}} \to \mathbb{R}^{D^{(l)}}$.*

*(2) For fixed $m \geq 1$, an approximating sequence $\left(S^2GNN_p^{(l)}\right)_{p\in\mathbb{N}}$ of $(m+1)$-layer $S^2GNNs$ with increasing layer-wise polynomial order $p$ exists such that, for all $(\mathcal{G}_p)_{p\in\mathbb{N}}$, the same bound holds.*

*Remark.* The proof of the simplified Theorem 4 used in the main body is analogous to the proof of Lemma 1 just without the nonlinearity, which has the following consequences:

- The final layer $m+1$ which we only need to apply one last nonlinearity to the output (since the spectral part of all layers, including the previous layer $m$, has none) becomes obsolete, so the final layer instead becomes $m$,

- The two limits (1) and (2) are equivalent by the reduction to an $mp$-order polynomial filter,

- We do not need the dimension-doubling "trick" outlined below to get rid of the nonlinearity in the proof and instead set all feature transform matrices in layers 1 through $m-1$ to the identity and the final ones to $\boldsymbol{W}_{\text{spec}}^{*(m)} = \boldsymbol{W}^{(l)}$, $\boldsymbol{W}_{\text{spat}}^{*(m)} = \boldsymbol{W}^{(l)}$.

*Proof of Lemma 1.* We first note that $m$ S$^2$GNN spatial parts, each of order $p$, would act like an $(mp)$-order polynomial filter (factorized into $m$ order-$p$ polynomials), were it not for the nonlinearities in between. However, using the fact that $\sigma$ is the ReLu function, we can choose intermediate hidden dimensions twice the size of the input dimension and then use the linear transforms to store a positive and a negative copy of the embeddings, add them back together after applying each ReLu, just to split the result back into a positive and negative copy for the next layer. This essentially gets us rid of $\sigma$. Throughout the proof, we use a star superscript to denote the specific parameters that will ultimately satisfy our bound, whereas we put no star above parameters that are yet to be fixed in a later part of the proof.

For $m \geq 2$, the trick discussed above works if we set

$$\boldsymbol{W}_{\text{spec}}^{*(1)} = \frac{1}{2}\begin{pmatrix} \boldsymbol{I} & -\boldsymbol{I} \end{pmatrix} \in \mathbb{R}^{D^{(l-1)}\times 2D^{(l-1)}},$$

$$\boldsymbol{W}_{\text{spec}}^{*(2)},\ldots,\boldsymbol{W}_{\text{spec}}^{*(m-1)} = \frac{1}{2}\begin{pmatrix} \boldsymbol{I} \\ -\boldsymbol{I} \end{pmatrix}\begin{pmatrix} \boldsymbol{I} & -\boldsymbol{I} \end{pmatrix} \in \mathbb{R}^{2D^{(l-1)}\times 2D^{(l-1)}},$$

$$\boldsymbol{W}_{\text{spec}}^{*(m+1)} = \boldsymbol{W}^{(l)}\begin{pmatrix} \boldsymbol{I} \\ -\boldsymbol{I} \end{pmatrix} \in \mathbb{R}^{2D^{(l-1)}\times D^{(l)}},$$

$$\boldsymbol{W}_{\text{spat}}^{*(1)} = \begin{pmatrix} \boldsymbol{I} & -\boldsymbol{I} \end{pmatrix} \in \mathbb{R}^{2D^{(l-1)}\times D^{(l-1)}},$$

$$\boldsymbol{W}_{\text{spat}}^{*(2)},\ldots,\boldsymbol{W}_{\text{spat}}^{*(m-1)} = \begin{pmatrix} \boldsymbol{I} \\ -\boldsymbol{I} \end{pmatrix}\begin{pmatrix} \boldsymbol{I} & -\boldsymbol{I} \end{pmatrix} \in \mathbb{R}^{2D^{(l-1)}\times 2D^{(l-1)}},$$

$$\boldsymbol{W}_{\text{spat}}^{*(m+1)} = \boldsymbol{W}^{(l)}\begin{pmatrix} \boldsymbol{I} \\ -\boldsymbol{I} \end{pmatrix} \in \mathbb{R}^{2D^{(l-1)}\times D^{(l)}}.$$

In the case $m = 1$, pick the matrices $\boldsymbol{W}_{\text{spec}}^{*(1)}, \boldsymbol{W}_{\text{spat}}^{*(1)}$ from above for the first, and the matrices $\boldsymbol{W}_{\text{spec}}^{*(m+1)}, \boldsymbol{W}_{\text{spat}}^{*(m+1)}$ from above for the second layer.

Set $\hat{g}_{\boldsymbol{\gamma}}^{*(m+1)}(\lambda) = 1$ and $\hat{g}_{\vartheta}^{*(m+1)}(\lambda) = 0$. Given these choices and a graph $\mathcal{G}$ with eigenvalues $\boldsymbol{\lambda}$,

$$(\text{S}^2\text{GNN}^{(l)})_{\mathcal{G}}(\boldsymbol{X}) = \sigma\Big(\boldsymbol{V}\Big(\hat{g}_{\text{spsp}}(\boldsymbol{\lambda}) \odot \big[\boldsymbol{V}^\top \boldsymbol{H}^{(l-1)}\boldsymbol{W}^{(l)}\big]\Big)\Big), \quad \hat{g}_{\text{spsp}} = \prod_{j=1}^m \Big(\hat{g}_{\vartheta}^{(j)} + \hat{g}_{\boldsymbol{\gamma}}^{(j)}\Big)$$

We see that $\hat{g}_{\text{spsp}}\big|_{[\lambda_{\max},2]} = \prod_{j=1}^m \hat{g}_{\boldsymbol{\gamma}}^{(j)}$ since $\hat{g}_{\vartheta}^{(j)}\big|_{[\lambda_{\max},2]} = 0$ for $1 \leq j \leq m$. This can express any polynomial up to order $mp$ on $[\lambda_{\max},2]$, since we assumed a layer-wise $p \geq 2$ and any polynomial with real coefficients factorizes into real-coefficient polynomials of degree less or equal to 2 by the fundamental theorem of algebra. On the interval $[0,\lambda_{\max}]$, on the other hand, the filter $\hat{g}_{\text{spsp}}\big|_{[0,\lambda_{\max}]}$ can express any IGNN filter $\hat{g}^{(l)}\big|_{[0,\lambda_{\max}]}$. For $m = 1$, this is immediately clear. Else, set $\hat{g}_{\vartheta}^{(j)}$ to constants $C_j \in \mathbb{R}_{\geq}$, $1 \leq j \leq m-1$ large enough that none of the polynomials $\Big(C_j + \hat{g}_{\boldsymbol{\gamma}}^{(j)}\Big)$, $1 \leq j \leq m-1$, has a zero in $[0,\lambda_{\max}]$. Defining $\hat{g}_{\vartheta}^{(m)} = \dfrac{\hat{g}^{(l)}\big|_{[0,\lambda_{\max}]}}{\prod_{j=1}^m \Big(C_j + \hat{g}_{\boldsymbol{\gamma}}^{(j)}\Big)} - \hat{g}_{\boldsymbol{\gamma}}^{(m)}\big|_{[0,\lambda_{\max}]}$ gives the desired function.

We proceed by making use of a result by D. Jackson (Natanson, 1964), which is essentially a converse to Theorem 9 which we used to prove Theorem 3:

**Theorem** (Jackson's theorem on an interval). *Let $a < b \in \mathbb{R}$, $k,r \in \mathbb{N}$ with $k \geq r - 1 \geq 0$, $f \in C^r[a,b]$. Then, a polynomial $p_k$ of degree less or equal to $k$ exists such that*

$$\|p_k - f\|_\infty \leq \frac{b-a}{2}\left(\frac{\pi}{2}\right)^r \frac{1}{(k+1)k\ldots(k-r+2)}\left\|\frac{d^r}{dx^r}f\right\|_\infty$$

Since $\hat{g}_{\text{spsp}}$ can express any polynomial up to order $mp$ on $[\lambda_{\max},2]$ and, for any such polynomial, find parameters for the spectral parts that match the ideal filter $\hat{g}^{(l)}\big|_{[0,\lambda_{\max}]}$ exactly (not contributing to the supremum error), we can directly transfer this theorem to our case. Define $\text{S}^2\text{GNN}_m^{(l)}$ from the lemma by setting the linear feature transforms and final-layer filters as above. For the filters in layers 1 through $m$, define $\gamma^{*(1)},\ldots,\gamma^{*(m)}$ such that $\prod_{j=1}^m \hat{g}_{\boldsymbol{\gamma}}^{*(j)}$ factorizes into into the polynomial from Jackson's theorem on $[\lambda_{\max},2]$, and $\vartheta^{*(1)},\ldots,\vartheta^{*(m)}$ to match $\hat{g}^{(l)}$ on $[0,\lambda_{\max}]$. This defines a filter $\hat{g}_{\text{spsp}}^{(l)}$. We then find, for $mp \geq r - 1 \geq 0$,

$$\|\hat{g}_{\text{spsp}}^{(l)} - \hat{g}^{(l)}\|_\infty \leq \frac{2-\lambda_{\max}}{2}\left(\frac{\pi}{2}\right)^r \frac{1}{(mp+1)\,mp\,\ldots\,(mp-r+2)}\left\|\frac{d^r}{d\lambda^r}\hat{g}^{(l)}\big|_{[0,\lambda_{\max}]}\right\|_\infty$$

Therefore, $\|\hat{g}_{\text{spsp}}^{(l)} - \hat{g}^{(l)}\|_\infty$ is of $\mathcal{O}\left(K_r(\hat{g},\lambda_{\text{cut}})(mp)^{-r}\right)$ and we can find a scaling constant that depends only on $r$. Since the Lipschitz constant of $\sigma$ is 1, we find for *any* graph $\mathcal{G}$ with eigenvalues $\boldsymbol{\lambda}$ and any graph signal $0 \neq \boldsymbol{X} \in \mathbb{R}^{|\mathcal{G}|}$,

$$\frac{\left\|\Big[(\text{S}^2\text{GNN}_m^{(l)})_{\mathcal{G}} - (\text{IGNN}^{(l)})_{\mathcal{G}}\Big](\boldsymbol{X})\right\|_F}{\|\boldsymbol{X}\|_F} \leq \frac{\left\|\boldsymbol{V}\Big(\hat{g}_{\text{spsp}}^{(l)} - \hat{g}^{(l)}\Big)(\boldsymbol{\lambda}) \odot \big[\boldsymbol{V}^\top \boldsymbol{X}\boldsymbol{W}^{(l)}\big]\right\|_F}{\|\boldsymbol{X}\|_F}$$

$$\leq \frac{\|\hat{g}_{\text{spsp}}^{(l)} - \hat{g}^{(l)}\|_\infty \left\|(\boldsymbol{V}\boldsymbol{V}^\top)\boldsymbol{X}\boldsymbol{W}^{(l)}\right\|_F}{\|\boldsymbol{X}\|_F} \leq \|\hat{g}_{\text{spsp}}^{(l)} - \hat{g}^{(l)}\|_\infty \|\boldsymbol{W}^{(l)}\|_2$$

$$= \mathcal{O}\left([\|\boldsymbol{W}^{(l)}\|_2 K_r(\hat{g},\lambda_{\text{cut}})](mp)^{-r}\right)$$

with a scaling constant that depends only on $r$. Exactly the same procedure and bounds hold if we instead keep $m$ fixed and increase $p$. This finishes the proof of Lemma 1. $\qquad\square$

We can now prove the main theorem by induction. Lemma 1 gives the initial step. Now, assume the theorem holds for $L$ IGNN layers. We can then choose $\big(\text{S}^2\text{GNN}_m\big)_{m\in\mathbb{N}} =$

$\left(\mathrm{S}^2\mathrm{GNN}_m^{(L+1)} \circ \mathrm{S}^2\mathrm{GNN}_m^{(L\circ\cdots\circ1)}\right)_{m\in\mathbb{N}}$, where $\mathrm{S}^2\mathrm{GNN}_m^{(L+1)}$ are the approximating models fulfilling Lemma 1, while $\mathrm{S}^2\mathrm{GNN}_m^{(L\circ\cdots\circ1)}$ fulfill the induction assumption. We assume fixed $p$ and increasing $m$, but the proof is fully analogous in the other case. Applying the same decomposition to $(\mathrm{IGNN}_m)_{m\in\mathbb{N}}$ lets us express the error on a graph sequence $(\mathcal{G}_m)_{m\in\mathbb{N}}$ as

$$\frac{\left\|\left[(\mathrm{S}^2\mathrm{GNN}_m)_{\mathcal{G}_m} - (\mathrm{IGNN})_{\mathcal{G}_m}\right](\boldsymbol{X})\right\|_F}{\|\boldsymbol{X}\|_F}$$

$$= \frac{\left\|\left[(\mathrm{S}^2\mathrm{GNN}_m^{(L+1)} \circ \mathrm{S}^2\mathrm{GNN}_m^{(L\circ\cdots\circ1)})_{\mathcal{G}_m} - (\mathrm{IGNN}_m^{(L+1)} \circ \mathrm{IGNN}_m^{(L\circ\cdots\circ1)})_{\mathcal{G}_m}\right](\boldsymbol{X})\right\|_F}{\|\boldsymbol{X}\|_F}$$

$$\leq (\|\boldsymbol{X}\|_F)^{-1} \left\|\left[(\mathrm{S}^2\mathrm{GNN}_m^{(L+1)} \circ \mathrm{S}^2\mathrm{GNN}_m^{(L\circ\cdots\circ1)})_{\mathcal{G}_m} - (\mathrm{S}^2\mathrm{GNN}_m^{(L+1)} \circ \mathrm{IGNN}_m^{(L\circ\cdots\circ1)})_{\mathcal{G}_m}\right](\boldsymbol{X})\right\|_F$$

$$+ (\|\boldsymbol{X}\|_F)^{-1} \left\|\left[(\mathrm{S}^2\mathrm{GNN}_m^{(L+1)} \circ \mathrm{IGNN}_m^{(L\circ\cdots\circ1)})_{\mathcal{G}_m} - (\mathrm{IGNN}_m^{(L+1)} \circ \mathrm{IGNN}_m^{(L\circ\cdots\circ1)})_{\mathcal{G}_m}\right](\boldsymbol{X})\right\|_F$$

$$\leq \left[\|\hat{g}\|_\infty^{\max}\|\boldsymbol{W}\|_2^{\max} + \mathcal{O}(K_r^{\max}(\lambda_{\mathrm{cut}})(pm)^{-r})\right]\mathcal{O}\left(C_L(\|\hat{g}\|_\infty^{\max},\|\boldsymbol{W}\|_2^{\max})\,K_r^{\max}(\lambda_{\mathrm{cut}})\,(pm)^{-r}\right)$$

$$+ \mathcal{O}(K_r^{\max}(\lambda_{\mathrm{cut}})(pm)^{-r})(\|\hat{g}\|_\infty^{\max}\|\boldsymbol{W}\|_2^{\max})^L$$

$$= \mathcal{O}\left(C_{L+1}(\|\hat{g}\|_\infty^{\max},\|\boldsymbol{W}\|_2^{\max})\,K_r^{\max}(\lambda_{\mathrm{cut}})\,(pm)^{-r}\right).$$

We first used the triangle inequality in line 3 to split the difference into two terms. Next, we bound the first term using the induction assumption, as well as the Lipschitz constant of $\mathrm{S}^2\mathrm{GNN}_m^{(L+1)}$, which in turn, by Lemma 1, is the Lipschitz constant of $\mathrm{IGNN}_m^{(L+1)}$ up to a term of $\mathcal{O}(K_r^{\max}(\lambda_{\mathrm{cut}})(pm)^{-r})$. We moreover bound the second term using the Lipschitz constant of $\mathrm{IGNN}_m^{(L\circ\cdots\circ1)}$, as well as Lemma 1 to arrive at the final result. $\qquad\square$

### H.5 Proof of Theorem 5

We next prove the stability of our positional encodings:

**Theorem 5.** *The Positional Encodings* PE *in Eq. 5 are stable according to Definition 1.*

Recall the definition of stability via Hölder continuity:

**Definition 1** (Stable PE)**.** *(Huang et al., 2024) A PE method* $\mathrm{PE}: \mathbb{R}^{n\times k} \times \mathbb{R}^k \to \mathbb{R}^{n\times k}$ *is called stable, if there exist constants* $c, C > 0$, *such that for any Laplacian* $\boldsymbol{L}, \boldsymbol{L}'$, *and* $\boldsymbol{P}_* = \arg\min_{\boldsymbol{P}} \|\boldsymbol{L} - \boldsymbol{P}\boldsymbol{L}'\boldsymbol{P}^\top\|_F$

$$\|\mathrm{PE}(\mathrm{EVD}(\boldsymbol{L})) - \boldsymbol{P}_* \mathrm{PE}(\mathrm{EVD}(\boldsymbol{L}'))\|_\mathrm{F} \leq C \cdot \|\boldsymbol{L} - \boldsymbol{P}_*\boldsymbol{L}'\boldsymbol{P}_*^\top\|_\mathrm{F}^c. \tag{6}$$

For this proof, we build on the work of Huang et al. (2024) where the authors show that under the assumptions of Definition 2, and some minor adjustments, a positional encoding of the following form Eq. 23 is stable (Theorem 10).

$$\mathrm{SPE}(\boldsymbol{V},\boldsymbol{\lambda}) = \rho\left(\boldsymbol{V}\,\mathrm{diag}\left(\phi_1(\boldsymbol{\lambda})\right)\boldsymbol{V}^\top, \boldsymbol{V}\,\mathrm{diag}\left(\phi_2(\boldsymbol{\lambda})\right)\boldsymbol{V}^\top, \ldots, \boldsymbol{V}\,\mathrm{diag}\left(\phi_k(\boldsymbol{\lambda})\right)\boldsymbol{V}^\top\right) \tag{23}$$

**Definition 2.** *The key assumptions for SPE are as follows:*

- $\phi_\ell$ *and* $\rho$ *are permutation equivariant.*

- $\phi_\ell$ *is* $K_\ell$-*Lipschitz continuous: for any* $\boldsymbol{\lambda}, \boldsymbol{\lambda}' \in \mathbb{R}^k, \|\phi_\ell(\boldsymbol{\lambda}) - \phi_\ell(\boldsymbol{\lambda}')\|_\mathrm{F} \leq K_\ell \|\boldsymbol{\lambda} - \boldsymbol{\lambda}'\|.$

- $\rho$ *is* $J$-*Lipschitz continuous: for any* $[\boldsymbol{B}_1, \boldsymbol{B}_2, \ldots, \boldsymbol{B}_k] \in \mathbb{R}^{n\times n\times k}$ *and* $[\boldsymbol{B}_1', \boldsymbol{B}_2', \ldots, \boldsymbol{B}_k'] \in \mathbb{R}^{n\times n\times k}, \|\rho(\boldsymbol{B}_1, \boldsymbol{B}_2, \ldots, \boldsymbol{B}_k) - \rho(\boldsymbol{B}_1', \boldsymbol{B}_2', \ldots, \boldsymbol{B}_k')\|_\mathrm{F} \leq J \sum_{l=1}^k \|\boldsymbol{B}_\ell - \boldsymbol{B}_\ell'\|_\mathrm{F}.$

**Theorem 10** (Stability of Eq. 23 by Huang et al. (2024))**.** *Under Definition 2,* SPE *(Eq. 23) is stable with respect to the input Laplacian: for Laplacians* $\boldsymbol{L}, \boldsymbol{L}'$,

$$\|\mathrm{SPE}(\mathrm{EVD}(\boldsymbol{L})) - \boldsymbol{P}_* \mathrm{SPE}(\mathrm{EVD}(\boldsymbol{L}'))\|_\mathrm{F} \leq (\alpha_1 + \alpha_2)\,k^{5/4}\sqrt{\|\boldsymbol{L} - \boldsymbol{P}_*\boldsymbol{L}\boldsymbol{P}_*^\top\|_\mathrm{F}}$$
$$+ \left(\alpha_2\frac{k}{\gamma} + \alpha_3\right)\|\boldsymbol{L} - \boldsymbol{P}_*\boldsymbol{L}\boldsymbol{P}_*^\top\|_\mathrm{F}, \tag{24}$$

*where the constants are $\alpha_1 = 2J\sum_{l=1}^k K_\ell, \alpha_2 = 4\sqrt{2}J\sum_{l=1}^k M_\ell$, and $\alpha_3 = J\sum_{l=1}^k K_\ell$. Here $M_\ell = \sup_{\boldsymbol{\lambda}\in[0,2]^k} \|\phi_\ell(\boldsymbol{\lambda})\|$ and again $\boldsymbol{P}_* = \arg\min_{\boldsymbol{P}\in\Pi(n)} \|\boldsymbol{L} - \boldsymbol{P}_*\boldsymbol{L}\boldsymbol{P}_*^\top\|_{\mathrm{F}}$. The eigengap $\gamma = \lambda_{k+1} - \lambda_k$ is the difference between the $(k+1)$-th and $k$-th smallest eigenvalues, and $\gamma = +\infty$ if $k = n$.*

We prove a similar bound for general weighted adjacency matrices $\boldsymbol{A} \in \mathbb{R}_{\geq 0}^{n\times n}$ (note that such a stability result would be trivial if we restrict $\boldsymbol{A} \in \{0,1\}^{n\times n}$, since any function on a finite set is Lipschitz continuous). To achieve this, we need a technical assumption in order to ensure that the function values do not blow up and degree normalization is indeed a Lipschitz continuous function: We assume that the domain of $\boldsymbol{A}$ is restricted to (symmetric) matrices whose degrees are uniformly bounded by some constants $0 < \tilde{D}_{\min} < \tilde{D}_{\max}$:

$$d_u := \sum_v A_{u,v} \in [\tilde{D}_{\min}, \tilde{D}_{\max}] \qquad \forall u \in \{1, \ldots, n\}. \tag{25}$$

To decompose the proof into smaller pieces we commonly use the well-known fact that the composition of Lipschitz continuous functions $f_1 \circ f_2$, with constants $C_1$ and $C_2$, is also Lipschitz continuous $\|f_1(f_2(y)) - f_1(f_2(x))\| \leq C_1 C_2 \|y - x\|$ with constant $C_1 C_2$.

*Proof.* Our proposed encoding (Eq. 5) matches roughly Eq. 23. Specifically, $\phi_\ell(\boldsymbol{\lambda}) = \mathrm{softmax}((\lambda_j - \boldsymbol{\lambda})\odot(\lambda_j - \boldsymbol{\lambda})/\sigma^2)$ with $\sigma \in \mathbb{R}_{>0}$. However, $\rho_\ell(\boldsymbol{B}_1, \boldsymbol{B}_2, \ldots, \boldsymbol{B}_k)$ does not directly match $\|_{j=1}^k[\boldsymbol{B}_j \odot \boldsymbol{A}] \cdot \vec{1}$, since it is also a function of the adjacency $\boldsymbol{A}$. Nevertheless, we show that $\phi_\ell$ is $K_\ell$-Lipschitz continuous and $\rho$ is $J$-Lipschitz continuous, where we also bound the change of $\boldsymbol{A}$.

We will start with $\phi_\ell(\boldsymbol{\lambda}) = \mathrm{softmax}((\lambda_j - \boldsymbol{\lambda})\odot(\lambda_j - \boldsymbol{\lambda})/\sigma^2)$. The softmax is well-known to be of Lipschitz constant 1 w.r.t. the $L^2$ vector norm/Frobenius norm. $-\boldsymbol{x}/\sigma$ has a Lipschitz constant of $1/\sigma$. This leaves us with the quadratic term $\psi_u(\boldsymbol{\lambda}) = (\lambda_u - \boldsymbol{\lambda}) \odot (\lambda_u - \boldsymbol{\lambda})$ where we bound the norm of the Jacobian

$$J_{\psi_u} = \begin{bmatrix} -2(\lambda_u - \lambda_1) & 0 & \ldots & 0 & \ldots & 0 \\ 0 & -2(\lambda_u - \lambda_2) & \ldots & 0 & \ldots & 0 \\ \vdots & \vdots & \ddots & \vdots & \vdots & \vdots \\ 0 & 0 & \ldots & 0 & \ldots & 0 \\ \vdots & \vdots & \ddots & \vdots & \vdots & \vdots \\ 0 & 0 & \ldots & 0 & \ldots & -2(\lambda_u - \lambda_k) \end{bmatrix} \tag{26}$$

that is zero everywhere except for the diagonal entries, excluding its $u$-th entry. Thus, $\|J_{\psi_u}\|_{\mathrm{F}} \leq 2k \max_{v\in\{1,2,\ldots,k\}}(\lambda_v - \lambda_u) \leq 2k(\lambda_k - \lambda_1) \leq 4k$, as $0 = \lambda_1 \leq \lambda_k \leq 2$. We can therefore use $K_\ell := 4k/\sigma$.

Now we continue with $\tilde{\rho}_\ell(\boldsymbol{A}, \boldsymbol{B}_1, \boldsymbol{B}_2, \ldots, \boldsymbol{B}_k)$. For $f(\boldsymbol{A}, \boldsymbol{B}) = (\boldsymbol{B}\odot\boldsymbol{A})\cdot\vec{1}$ with a general weighted adjacency $\boldsymbol{A} \in \mathbb{R}^{n\times n}$, we consider

$$\begin{aligned}
\|(\boldsymbol{B} \odot \boldsymbol{A}) \cdot \vec{1} - (\boldsymbol{B}' \odot \boldsymbol{A}') \cdot \vec{1}\|_{\mathrm{F}} &\underset{(A)}{\leq} \|\vec{1}\|_2 \|\boldsymbol{B} \odot \boldsymbol{A} - \boldsymbol{B}' \odot \boldsymbol{A}'\|_{\mathrm{F}} \\
&= \sqrt{n}\|\boldsymbol{B} \odot \boldsymbol{A} - \boldsymbol{B}' \odot \boldsymbol{A}'\|_{\mathrm{F}} \\
&= \sqrt{n}\|\boldsymbol{B} \odot \boldsymbol{A} - \boldsymbol{B}' \odot \boldsymbol{A} + \boldsymbol{B}' \odot \boldsymbol{A} - \boldsymbol{B}' \odot \boldsymbol{A}'\|_{\mathrm{F}} \\
&\underset{(B)}{\leq} \sqrt{n}\|\boldsymbol{B} \odot \boldsymbol{A} - \boldsymbol{B}' \odot \boldsymbol{A}\|_{\mathrm{F}} + \sqrt{n}\|\boldsymbol{B}' \odot \boldsymbol{A} - \boldsymbol{B}' \odot \boldsymbol{A}'\|_{\mathrm{F}} \\
&= \sqrt{n}\|(\boldsymbol{B} - \boldsymbol{B}') \odot \boldsymbol{A}\|_{\mathrm{F}} + \sqrt{n}\|\boldsymbol{B}' \odot (\boldsymbol{A} - \boldsymbol{A}')\|_{\mathrm{F}} \\
&\underset{(C)}{\leq} \sqrt{n}\underbrace{\max_{u,v} A_{u,v}}_{\underset{(D)}{\leq} \tilde{D}_{\max}} \|\boldsymbol{B} - \boldsymbol{B}'\|_{\mathrm{F}} + \underbrace{\max_{u,v} B'_{u,v}}_{\underset{(E)}{\leq} 1} \sqrt{n} \underbrace{\|\boldsymbol{A} - \boldsymbol{A}'\|_{\mathrm{F}}}_{(F)}.
\end{aligned}$$

$$\tag{27}$$

(A) holds by Cauchy-Schwarz, (B) by triangle inequality, (C) by Cauchy-Schwarz, (D) follows from the domain of $\boldsymbol{A}$, and (E) is true since the largest eigenvalue of $\boldsymbol{B} = \boldsymbol{V}\phi_\ell(\boldsymbol{\lambda})\boldsymbol{V}^\top$ is 1 because $\phi_\ell(\boldsymbol{\lambda})_j \leq 1, \forall 1 \leq j \leq k$.

To further bound (F), i.e. $\|\boldsymbol{A} - \boldsymbol{A}'\|_{\mathrm{F}}$, note that $\|\boldsymbol{L} - \boldsymbol{L}'\|_{\mathrm{F}} = \|\boldsymbol{D}^{-1/2}\boldsymbol{A}\boldsymbol{D}^{-1/2} - \boldsymbol{D}'^{-1/2}\boldsymbol{A}'\boldsymbol{D}'^{-1/2}\|_{\mathrm{F}}$. For $g(\boldsymbol{A}) := \boldsymbol{D}^{1/2}\boldsymbol{A}\boldsymbol{D}^{1/2}$, our initial assumption from Eq. 25 yields the existence of a Lipschitz constant $C_{\tilde{D}_{\min},\tilde{D}_{\max}}$ for $g$, which can be verified by computing the partial derivatives of $g$. Thus, we can bound

$$
\begin{aligned}
\|\boldsymbol{A} - \boldsymbol{A}'\|_{\mathrm{F}} &= \|g(\boldsymbol{D}^{-1/2}\boldsymbol{A}\boldsymbol{D}^{-1/2}) - g(\boldsymbol{D}'^{-1/2}\boldsymbol{A}'\boldsymbol{D}'^{-1/2})\|_{\mathrm{F}} \\
&\leq C_{\frac{\tilde{D}_{\min}}{\tilde{D}_{\max}}, \frac{\tilde{D}_{\max}}{\tilde{D}_{\min}}} \|\boldsymbol{D}^{-1/2}\boldsymbol{A}\boldsymbol{D}^{-1/2} - \boldsymbol{D}'^{-1/2}\boldsymbol{A}'\boldsymbol{D}'^{-1/2}\|_{\mathrm{F}} \\
&= C_{\frac{\tilde{D}_{\min}}{\tilde{D}_{\max}}, \frac{\tilde{D}_{\max}}{\tilde{D}_{\min}}} \|\boldsymbol{L} - \boldsymbol{L}'\|_{\mathrm{F}} =: \alpha_4 \|\boldsymbol{L} - \boldsymbol{L}'\|_{\mathrm{F}}.
\end{aligned}
\tag{28}
$$

As concatenation of $k$ vectors $\|_{j=1}^{k}\boldsymbol{x}$ has a Lipschitz constant of 1, we have $J = \sqrt{n}\tilde{D}_{\max}$. Moreover, we have an additional term for the RHS of Eq. 24 with constant $\alpha_4\sqrt{n}k$, coming from (F) and Eq. 28.

To finalize the proof, we restate the beginning of the proof of Huang et al. (2024) and incorporate the additional $\boldsymbol{A}$-dependency of $\tilde{\rho}_\ell(\boldsymbol{A}, \boldsymbol{B}_1, \boldsymbol{B}_2, \ldots, \boldsymbol{B}_k)$ with $\boldsymbol{B}_j = \boldsymbol{V}\operatorname{diag}(\phi_j(\boldsymbol{\lambda}))\boldsymbol{V}^\top$ for $1 \leq j \leq k$.

$$
\begin{aligned}
&\|\mathrm{SPE}(\mathrm{EVD}(\boldsymbol{L}), \boldsymbol{L}) - \boldsymbol{P}_* \mathrm{SPE}(\mathrm{EVD}(\boldsymbol{L}'), \boldsymbol{L})\|_{\mathrm{F}} \\
&= \|\tilde{\rho}_\ell(\boldsymbol{A}, \boldsymbol{B}_1, \boldsymbol{B}_2, \ldots, \boldsymbol{B}_k) - \boldsymbol{P}_*\tilde{\rho}_\ell(\boldsymbol{A}', \boldsymbol{B}_1', \boldsymbol{B}_2', \ldots, \boldsymbol{B}_k')\|_{\mathrm{F}} \\
&= \|\tilde{\rho}_\ell(\boldsymbol{A}, \boldsymbol{B}_1, \boldsymbol{B}_2, \ldots, \boldsymbol{B}_k) - \tilde{\rho}_\ell(\boldsymbol{P}_*\boldsymbol{A}'\boldsymbol{P}_*^\top, \boldsymbol{P}_*\boldsymbol{B}_1'\boldsymbol{P}_*^\top, \boldsymbol{P}_*\boldsymbol{B}_2'\boldsymbol{P}_*^\top, \ldots, \boldsymbol{P}_*\boldsymbol{B}_k'\boldsymbol{P}_*^\top)\|_{\mathrm{F}} \\
&\leq \underbrace{\left[ J \sum_{l=1}^{k} \|\boldsymbol{B}_l - \boldsymbol{P}_*\boldsymbol{B}_l'\boldsymbol{P}_*^\top\|_{\mathrm{F}} \right]}_{\text{subject of Huang et al. (2024)}} + \alpha_4\sqrt{n}k \|\boldsymbol{L} - \boldsymbol{P}_*\boldsymbol{L}_l'\boldsymbol{P}_*^\top\|_{\mathrm{F}}.
\end{aligned}
\tag{29}
$$

Including the extra term stemming from our $\boldsymbol{A}$-dependent $\tilde{\rho}_\ell(\boldsymbol{A}, \boldsymbol{B}_1, \boldsymbol{B}_2, \ldots, \boldsymbol{B}_k)$, the stability guarantee reads

$$
\begin{aligned}
\|\mathrm{SPE}(\mathrm{EVD}(\boldsymbol{L}), \boldsymbol{L}) - \boldsymbol{P}_* \mathrm{SPE}(\mathrm{EVD}(\boldsymbol{L}'), \boldsymbol{L})\|_{\mathrm{F}} &\leq (\alpha_1 + \alpha_2)k^{5/4}\sqrt{\|\boldsymbol{L} - \boldsymbol{P}_*\boldsymbol{L}\boldsymbol{P}_*^\top\|_{\mathrm{F}}} \\
&+ \left(\alpha_2\frac{k}{\gamma} + \alpha_3 + \alpha_4\sqrt{n}k\right)\|\boldsymbol{L} - \boldsymbol{P}_*\boldsymbol{L}\boldsymbol{P}_*^\top\|_{\mathrm{F}}
\end{aligned}
\tag{30}
$$

with the newly introduced $\alpha_4$ arising as Lipschitz constant of (inverse) degree normalization. The proof is complete. $\qquad\square$

**Windowing for "eigengap" independent bounds.** Note that $C$ depends on the eigengap between $1/\lambda_{k+1} - \lambda_k$ at the frequency cutoff. One should be able to improve upon this bound with windowing (see Fig. 7)), effectively lowering the Lipschitz constant of $\hat{h}_j(\boldsymbol{\lambda})$ around $\lambda_k$. We leave a formal treatment of this insight to future work.

### H.6 Proof of Theorem 6

We next prove the expressivity of a GNN/S²GNN in combination with our positional encodings:

**Theorem 6.** *S²GNNs are strictly more expressive than 1-WL with the PE of Eq. 5.*

For this, we assume that the positional encodings are the only node attributes, subsuming a constant feature or that there is a linear transformation on the raw features. We require that the choice of spatial MPGNN / spectral filter is at least as expressive as the 1-WL test, which is the case, e.g., for GIN. Moreover, we assume that the node-level embeddings are aggregated to the graph level using summation.

*Proof.* To show that $\mathrm{GNN}(\mathrm{PE}(\boldsymbol{V}, \lambda))$ is strictly more expressive as 1-WL. For all graphs that 1-WL can distinguish, the GNN may learn to ignore the PE. Thus, we only need to prove that the positional encodings/node features of $\mathrm{PE}(\boldsymbol{V}, \lambda)$ suffice to distinguish some graphs that 1-WL could

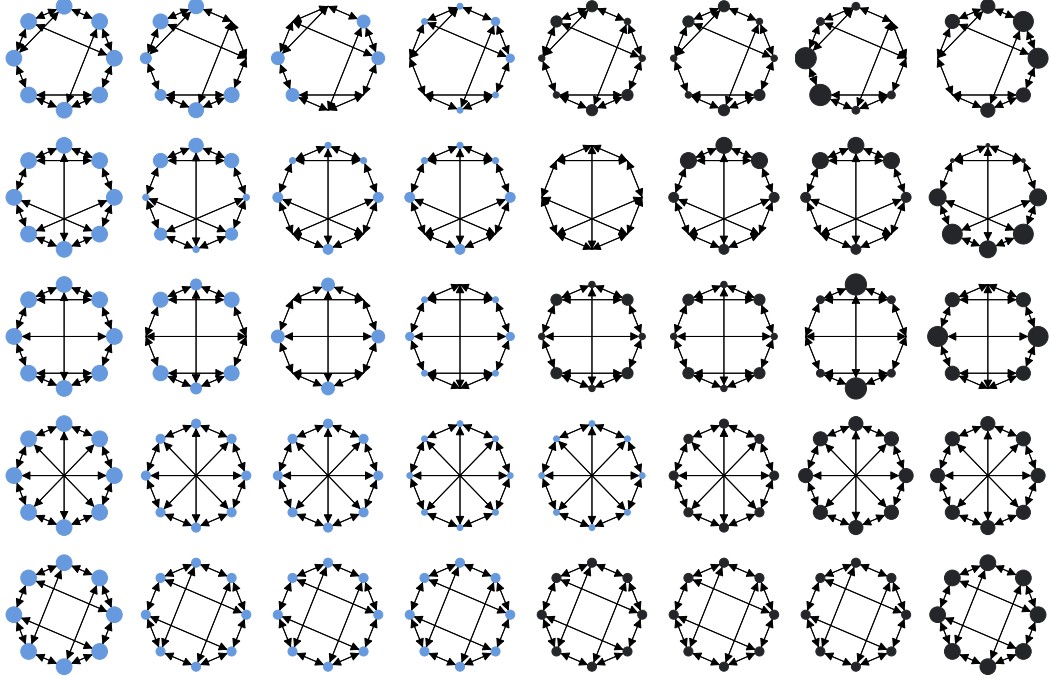

Figure 15: Our positional encodings PE Eq. 5 illustrated in the node colors and sizes. We plot all 5 (rows) 3-regular graphs with 8 nodes and all possible dimensions of the encoding (columns). We use $\sigma = 0.001$. Color denotes the sign, and size encodes the absolute value. We hypothesize that the visual "smoothness" between graphs and dimensions is due to our PE's stability (Theorem 5).

not distinguish. For all graphs that 1-WL can distinguish we know, by assumption, that the GNN can distinguish the graphs.

As Li et al. (2020) point out, 1-WL (and MPGNN that are as capable as 1-WL) cannot distinguish degree-regular graphs with the same number of nodes and degrees. A degree regular graph is a graph where each node has the same degree. This is closely related to Theorem 11.

We next show that our PE alone distinguishes certain degree-regular graphs. In this construction, we consider all 3-regular graphs with $n = 8$ nodes for this (see Fig. 15). The encodings $\vec{1}$ PE result in the following values with $\sigma = 0.001$ and rounded to max 2 decimal places:

$$\vec{1}^\top \text{PE}(\text{EVD}(\boldsymbol{L}_1)) = [3 \quad 1.73 \quad 1 \quad 0.41 \quad -1 \quad -1 \quad -1.73 \quad -2.41]$$

$$\vec{1}^\top \text{PE}(\text{EVD}(\boldsymbol{L}_2)) = [3 \quad 1.56 \quad 0.62 \quad 0.62 \quad 0 \quad -1.62 \quad -1.62 \quad -2.41]$$

$$\vec{1}^\top \text{PE}(\text{EVD}(\boldsymbol{L}_3)) = [3 \quad 1.73 \quad 1 \quad 0.41 \quad -1 \quad -1 \quad -1.73 \quad -2.41] \tag{31}$$

$$\vec{1}^\top \text{PE}(\text{EVD}(\boldsymbol{L}_4)) = [3 \quad 1 \quad 1 \quad 0.41 \quad 0.41 \quad -1 \quad -2.41 \quad -2.41]$$

$$\vec{1}^\top \text{PE}(\text{EVD}(\boldsymbol{L}_5)) = [3 \quad 1 \quad 1 \quad 1 \quad -1 \quad -1 \quad -1 \quad -3]$$

By constructing examples, this shows that our PE can distinguish 4 out of the 5 3-regular graphs with 8 nodes. Thus, our PE may distinguish at least some graphs that 1-WL cannot. This concludes the proof.

$\square$

# I    Expressivity of Spectral Filters and Spectrally Designed Spatial Filters

While it is well-known that common spatial MPGNNs are at most as expressive as 1-WL and that spectrally designed GNNs can be more expressive than 1-WL (Theorem 2 of Balcilar et al. (2021a)), we show that spectral GNNs are not able to distinguish degree-regular graphs. This upper bound was

not known/formalized prior to our work (Bo et al., 2023b). Fortunately, our PE largely mitigates the limitation. The improved expressivity of our positional encodings, along with their efficiency, stems from the element-wise product with $A$ (see also Geerts (2021)).

**Theorem 11.** *Spectral filters $V \operatorname{diag}(\hat{g}(\boldsymbol{\lambda})) V^\top \vec{1}$ are strictly less expressive than 3-WL with Laplacian $L = D - A$, $L = I - D^{-1}A$, or $L = I - D^{-1/2}AD^{-1/2}$.*

*Proof.* The proof relies on properties of the eigenvectors for the different choices $L_\mathrm{u} = D - A$, $L_\mathrm{rw} = I - D^{-1}A$, or $L_\mathrm{s} = I - D^{-1/2}AD^{-1/2}$. For $L_\mathrm{u}\vec{1} = \lambda_0\vec{1} = 0$ and $L_\mathrm{rw}\vec{1} = \lambda_0\vec{1} = 0$ the first eigenvector is constant. The first eigenvector of $L_\mathrm{s}$ is $D^{1/2}\vec{1}$ (ignoring normalization). Thus, for degree-regular graphs, the first eigenvector of $L_\mathrm{s}$ is also constant.

By the orthogonality of eigenvectors, $\boldsymbol{v}_u \perp \boldsymbol{v}_v$ if $u \neq v$, we know that all other eigenvectors are orthogonal to constant node features. Consequently, the "Fourier transformed" node features are $V^\top \vec{1} = \begin{bmatrix} \sqrt{n} & 0 & \dots & 0 \end{bmatrix}$ for all three choices $L_\mathrm{u}$, $L_\mathrm{rw}$, and $L_\mathrm{s}$. Since this is true for all degree-regular graphs, spectral GNNs cannot distinguish degree-regular graphs with the same number of nodes.

Since the 3-WL test can distinguish some degree-regular graphs, 3-WL is strictly more expressive than a spectral GNN. $\qquad\square$

**Corollary 1.** *"Spectrally designed" MPGNNs that use a polynomial parametrization of filter $\operatorname{diag}(\hat{g}(\boldsymbol{\lambda}))$ are strictly less expressive than 3-WL with the same choices for $L$.*

*Proof.* With a polynomial parametrization of the spectral filter $\hat{g}(\boldsymbol{\lambda})$, we know $V(\hat{g}(\boldsymbol{\lambda}) \odot [V^\top \boldsymbol{x}]) = V \operatorname{diag}(\hat{g}(\boldsymbol{\lambda})) V^\top \boldsymbol{x} = \hat{g}(L)\boldsymbol{x} = \sum_{j=0}^{p} \gamma_j L^j \boldsymbol{x}$ (see § 2). Due to this equivalence between a spectral and spatial filter and the constant node features $\boldsymbol{x} = \vec{1}$, any polynomial filter $\sum_{j=0}^{p} \gamma_j L^j \vec{1}$ cannot distinguish degree-regular graphs. This argument also holds if the polynomial filter is normalized by the maximum eigenvalue as done by ChebNet (Defferrard et al., 2017). $\qquad\square$

# J  Further Remarks on S²GNNs

We next provide insights, details, and remarks on the details and variants of S²GNNs, accompanying the main section § 3. The structure roughly follows the main body.

Next to the overview in Fig. 2 and the method description of the main part, we provide pseudo-code in Algo. 1 for a Spatio-Spectral Graph Neural Network on a graph with node attributes, and in Algo. 2 for a spectral filter.

---

**Algorithm 1** Spatio-Spectral Graph Neural Network (S²GNN), implementing Eq. 1

1: **Input:** Adjacency $A \in \mathbb{R}_{\geq 0}^{n \times n}$, , node attributes $X \in \mathbb{R}^{n \times d^{(0)}}$, number of eigenvectors $k$
2: $V, \boldsymbol{\lambda} \leftarrow \mathrm{EVD}(L(A), k)$
3: $H^{(0)} \leftarrow X + \mathrm{PE}(V, \boldsymbol{\lambda})$
4: **for** $l \in \{1, 2, \dots, \ell\}$ **do**
5: $\quad H^{(l)} \leftarrow \mathrm{Spectral}^{(l)}(H^{(l-1)}; V, \boldsymbol{\lambda}) + \mathrm{Spatial}^{(l)}(H^{(l-1)}; A)$
6: **Return** $H^{(\ell)}$

---

**Algorithm 2** Real-valued spectral filter of Eq. 4

1: **Input:** Node embeddings $H^{(l-1)} \in \mathbb{R}^{n \times d^{(l-1)}}$, eigenvalues $\boldsymbol{\lambda} \in [0, 2]^k$, eigenvectors $V \in \mathbb{R}^{n \times k}$
2: $\hat{g}_\vartheta^{(l)}(\boldsymbol{\lambda}) \leftarrow \mathrm{Smearing}(\boldsymbol{\lambda})W \odot \mathrm{Window}(\boldsymbol{\lambda})$
3: $\hat{H}^{(l-1)} \leftarrow V^\top H^{(l-1)}$
4: $\hat{H}^{(l)} \leftarrow s_\zeta^{(l)}(\hat{g}_\vartheta^{(l)}(\boldsymbol{\lambda}) \odot \hat{H}^{(l-1)})$
5: $H^{(l)} \leftarrow V \hat{H}^{(l)}$
6: **Return** $H^{(l)}$

---

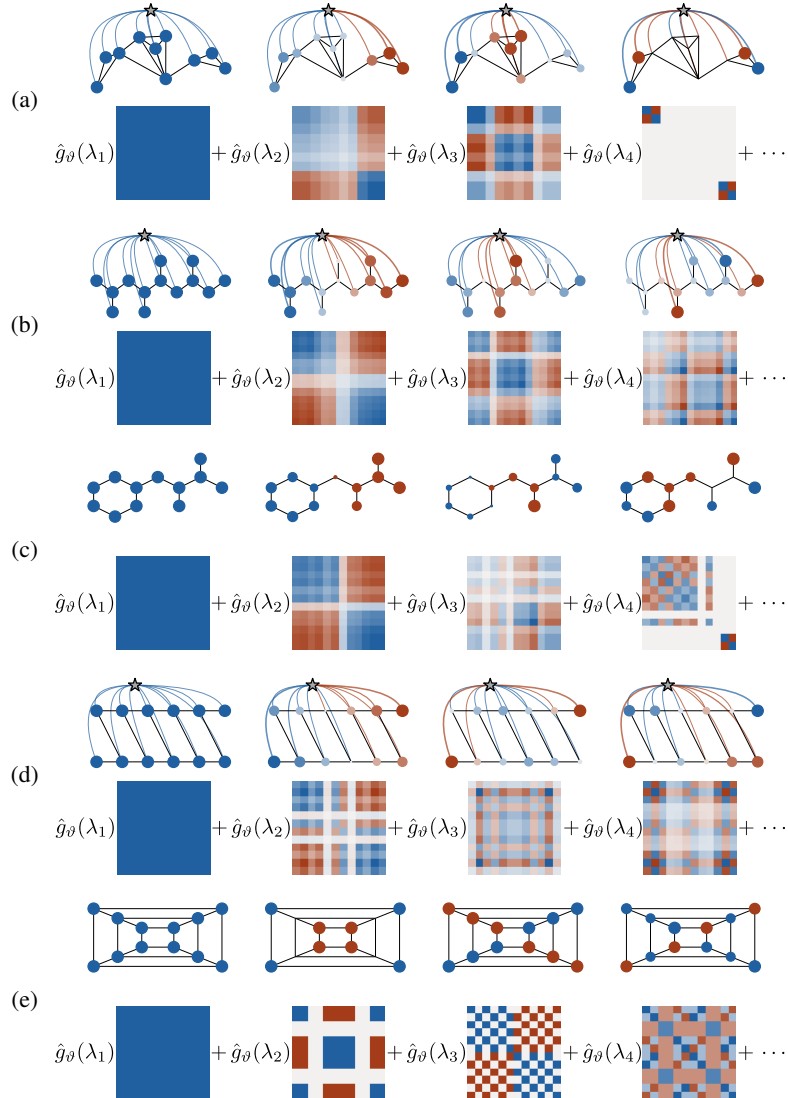

Figure 16: Sketch of intra- and inter-cluster message passing capabilities $\boldsymbol{V}(\hat{g}_\vartheta(\boldsymbol{\lambda}) \odot [\boldsymbol{V}^\top \boldsymbol{X}] = [\sum_{j=1}^k \hat{g}_\vartheta(\lambda_j)\boldsymbol{v}_j\boldsymbol{v}_j^\top]\boldsymbol{X}$. The "star" node reflects the *global* Fourier coefficient and colors/widths illustrate its signed and weighted message passing. We show the first four eigenvectors, order nodes left to right in $\boldsymbol{v}_j$, and sum repeated eigenvalues.

## J.1 Visualization of Spectral Filters

In Fig. 16, we provide further examples of hierarchies/eigenspaces spectral filters have access to, complementing Fig. 3 & 4. Here and in the main part, we use the main diagonal of $\sum_{j \text{ s.t. } \lambda_j=\lambda_u} \boldsymbol{v}_j\boldsymbol{v}_j^\top$ for deciding on the edge weights of the graph structures, potentially summing over multiple eigenvectors with identical values $\lambda_j = \lambda_u$. We take the product $\prod_{j \text{ s.t. } \lambda_j=\lambda_u} \text{sign}(\boldsymbol{v}_j)$ for visualizing the sign of the $n$ edges for the global aggregation.

For all graphs, the first eigenvector denotes the constant signal (for $\boldsymbol{L} = \boldsymbol{D} - \boldsymbol{A}$). For (a-d), we observe that the second eigenspace roughly describes a half oscillation, i.e., the left vs. right part of the graph. The third eigenspace separates the middle parts. For (a), the fourth eigenspace models the interactions between the extremal nodes. For (b-d), the frequency increments again, effectively clustering the graph in four roughly equal pieces. For (e), the eigenspaces model the interplay between (automorphic) inner and outer structures, as well as the vertical and horizontal symmetry.

### J.2 Composition of Filters

Composing a *residual connection* with a graph filter $G = \operatorname{diag}(\hat{g}(\boldsymbol{\lambda})) \in \mathbb{R}^{n \times n}$ yields $Y = VGV^\top H + H = V(G+I)V^\top H$, *chaining multiple filters* (without nonlinearities) results in $VG_2V^\top VG_1V^\top H = VG_2G_1V^\top H$. Chaining and residual connections resolve to $V(G_2G_1 + G_2 + G_1 + I)V^\top H$. Hence, an arbitrary sequence of graph filters (Eq. 2) can be more flexible due to the interactions between filters. Note that this composition is only true in the absence of nonlinearities. Nevertheless, the main intuition about how filters interact remains approximately the same also in the light of nonlinearities.

### J.3 Exhaustive Reasons Why Low Frequencies Are Sensible

A sensible default is to focus on the low frequencies. We specifically identify the following six reasons: (1) Low frequencies model the smoothest global signals w.r.t. the high-level graph structure (see Fig. 3 & 4). (2) Gama et al. (2020) find that, under a relative perturbation model (perturbation budget proportional to connectivity), stability implies $C$-integral-Lipschitzness ($\exists C > 0\colon |\lambda d\hat{g}/d\lambda| \leq C$), i.e., the filter can vary arbitrarily around zero but must level out towards larger $\lambda$. Stability to graph perturbations is a strong domain-agnostic prior. (3) Many physical long-range interactions are power laws with a flattening frequency response. For example, we construct an explicit graph filter modeling the electric potential of charges in a 1D "ion crystal" (§ G) and find that a low-pass window is optimal. (4) Sequence models like Hyena (Poli et al., 2023) apply global low-pass filters through their exponential windows. (5) Cai et al. (2023) prove that an MPGNN plus virtual node (see § E) can emulate DeepSets (Zaheer et al., 2017) and, thus, approximate self-attention to any precision. Nonetheless, we find that a virtual node alone does not necessarily yield good generalization (§ 3.1.1 & 4.1). (6) Nonlinearities "spill" features between frequency bands (Gama et al., 2020). This includes spillage from higher frequencies to the band of the spectral filter. Gama et al. (2020) argue that this spillage makes it possible to learn stable yet expressive graph filters and is also a feature of stable message passing models.

### J.4 Scaling to Graphs of Different Magnitude

For scaling a single to graphs of different orders of magnitude, it can be beneficial to rescale the eigenvalues before learning the filter $\hat{g}_\vartheta(\boldsymbol{\lambda})$. That is, we use $\hat{g}_\vartheta^{(l)}(\tilde{\boldsymbol{\lambda}})$ with rescaled $\tilde{\boldsymbol{\lambda}}$.

For example, the eigenvalues for a path/sequence are $\lambda_j \approx (1 - \cos(\pi j/n))$. Thus, the resolution is poor, especially for the eigenvalues close to zero since $\cos$ approaches slope 0. For this reason, we consider rescaling the eigenvalues with

$$\tilde{\lambda}_j = 1/\pi \cos^{-1}(1 - \lambda_j) \tag{32}$$

or

$$\tilde{\lambda}_j = n/\pi \cos^{-1}(1 - \lambda_j) \tag{33}$$

The latter is, e.g., convenient for identifying the second lowest eigenvalue regardless of $n$. Due to the poor numerical properties of these relations, we evaluate $\cos^{-1}(1 - \lambda_j) = \tan^{-1}(\sqrt{2\lambda_j - \lambda_j^2}/1 - \lambda_j)$ instead.

### J.5 Spectral Normalization

While the GFT and its inverse preserve the norm of the input (e.g., $\|\hat{x}\|_2 = \|V^\top x\|_2 = \|x\|_2$), this is not true if operating on a truncated frequency spectrum or if the filter $\hat{g}_\vartheta(\boldsymbol{\lambda})$ suppresses certain frequencies. For example, in the example of a virtual node (for simplicity here with $L = D - A$), a signal $x$ that is zero at every node but one at a single node, then the signal will be equally scattered to every frequency. Then, suppressing all frequencies but $\lambda = 0$, yields $\|V\mathbb{1}_{\{0\}}V^\top x\|_2 = 1/\sqrt{n}$.

Motivated by this unfortunate scaling, we also consider normalization in the spectral domain. Specifically, we normalize $\hat{H} = \hat{g}_\vartheta(\boldsymbol{\lambda}) \odot [V^\top f_\theta(H)] \in \mathbb{R}^{k \times d}$ s.t. $\hat{H}_j \leftarrow (1-a_j)\hat{H}_j + a_j \hat{H}_j/\|\hat{H}_j\|_2$ with learnable $a \in [0,1]^d$. This allows, e.g., broadcasting a signal from one node without impacting its scale. However, we empirically find that this normalization only helps only marginally in the over-smoothing experiment (Di Giovanni et al., 2023a) and otherwise can destabilize training. We also consider variants where the norm in the spectral domain is scaled with the norm of the signal

in the spatial domain with more or less identical results. We hypothesize that such normalization is counter-productive for, e.g., a bandpass filter if the signal does not contain the corresponding frequencies.

## J.6 Adjusting S²GNNs to Directed Graphs

For the spectral filter of Eq. 3, we use $f_\theta^{(l)}(\hat{\boldsymbol{H}}^{(l)}) = \boldsymbol{H}^{(l)} \odot [\sigma(\boldsymbol{H}^{(l)} \boldsymbol{W}_{G,\Re}^{(l)}) + i \cdot \sigma(\boldsymbol{H}^{(l)} \boldsymbol{W}_{G,\Im}^{(l)})]$ and subsequently map the result of $\mathrm{Spectral}$ back the real domain, e.g., using $\boldsymbol{w}_\Re^{(l)} \Re(\mathrm{Spectral}^{(l)}(\boldsymbol{H}^{(l-1)})) + \boldsymbol{w}_\Im^{(l)} \Im(\mathrm{Spectral}^{(l)}(\boldsymbol{H}^{(l-1)}))$, with learnable weights $\boldsymbol{w}_\Re^{(l)}, \boldsymbol{w}_\Im^{(l)} \in \mathbb{R}^d$ and real $\Re(\cdot)$ as well as imaginary component $\Im(\cdot)$. For the positional encodings $\mathrm{PE}(\boldsymbol{V}, \boldsymbol{\lambda})$ of § 3.2.4, we use $\boldsymbol{A}_s$ in Eq. 5 and concatenate real as well as imaginary components. The neural network for the spectral domain $s_\zeta$ of § 3.2.2 generalizes without adjustment. Similar to Koke & Cremers (2024), one could also employ complex weights; however, we do not.

## J.7 Computational Remarks

We use readily available eigensolvers (`scipy`) and, thus, use a fixed number of eigenvectors (typically $k \ll n$) instead of determining $k$ based on $\lambda_{\mathrm{cut}}$. The partial eigendecomposition is of complexity $\mathcal{O}(km)$ for $m$ edges, while the spectral filter has complexity $\mathcal{O}(kdn)$. On a different remark, we batch multiple graphs using block diagonal matrices (Fig. 17).

**Spectral graph-level readouts.** The key insight is that frequencies are a global concept, and hence, the GFT can be used for global readouts in graph-level tasks. With $k \ll n$, such a readout is practically free in the presence of intermediate spectral layers and of $\mathcal{O}(kn)$ otherwise. Thus, there is the opportunity for a computationally convenient aggregation of global information, including a sort of graph-level "jumping knowledge" (Xu et al., 2018). The only caveat is that the Fourier coefficients are not unique due to the ambiguity in the eigendecomposition. To maintain permutation equivariance, we take the absolute value and aggregate over dimension $k$ in Eq. 4 instead of the multiplication with $\boldsymbol{V}$.

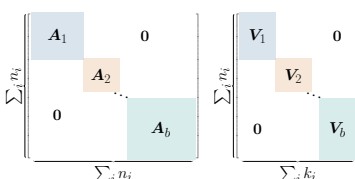

Figure 17: Block diagonal batching for spatial and spectral filters.

We observe that such intermediate readout can improve performance slightly, e.g., on TPUGraphs. However, we leave a systematic evaluation of its benefits for future work.

**Linear bottle necks.** To circumvent overfitting, we commonly replace the linear transformations $\boldsymbol{W}\boldsymbol{X}$ in $f_\theta^{(l)}(\hat{\boldsymbol{H}}^{(l)})$ and $\hat{g}_\vartheta(\boldsymbol{\lambda})$ with low-rank bottlenecks $\boldsymbol{W}_2\boldsymbol{W}_1\boldsymbol{X}$, s.t. $\boldsymbol{W} \in \mathbb{R}^{d \times d}$, $\boldsymbol{W}_2 \in \mathbb{R}^{d \times d'}$, $\boldsymbol{W}_1 \in \mathbb{R}^{d' \times d}$, and $d' < d$.

# K Limitations

We expect that many common graph benchmarks do not have or only insignificant long-range interactions. We observe that MPGNNs are less likely to overfit, perhaps since locality is a good inductive bias in many circumstances (Bronstein et al., 2021). Moreover, we observe that the spectral filter (§ 3.2.1) may converge slowly and get stuck in local optima. We find that a sufficient amount of randomly initialized filters mitigates this issue to a large extent. Further, one can introduce inductive biases via windowing functions (Fig. 7), like the exponential window used by Hyena (Poli et al., 2023).

Even if the true causal model generating the target consists of long-range interactions, it might be sufficient to model the training data solely using (potentially spurious) local interactions. This might be especially true if the training nodes are samples from a "small" vicinity of the graph (e.g., OGB Products (Hu et al., 2020)).

Closely related to the previous point is the amount of available training data. We hypothesize that S²GNNs are more *data-hungry* than their purely spatial counterpart. That is, to reliably detect (non-spurious) long-range interactions in the training data, a sufficient amount of data is required. Similar findings have been made, e.g., in the image domain (Dosovitskiy et al., 2021).

Except for heterophilic graphs, direction plays a small role in graph machine learning even though many benchmark tasks actually consist of directed graphs (Rossi et al., 2023). Moreover, there is a lack of benchmarks involving directed graphs, which require long-range interactions. Note that most of the theoretical findings generalize to directed graphs under appropriate modeling decisions/assumptions. However, we do not make this discussion explicit since MPGNNs for directed graphs are still actively researched.

Since a lot of the previous points hover around the insufficiency of the available benchmarks, we propose two new tasks § 4.1 and derive further datasets, e.g., for associative recall.

While we demonstrate the practicality of $S^2$GNNs in § 4.3 on large-scale benchmarks, the partial eigendecomposition EVD starts to become costly on the largest graphs we use for evaluation. Even though we did not experiment with lowering the requested precision, etc., we expect that for scaling further, naïve approaches might not be sufficient. One direction could be to utilize GPUs instead of CPUs or to adapt concepts, e.g., from spectral clustering (von Luxburg, 2007).

Even though there are many important reasons why we should utilize a spectral filter on the low end of the spectrum, there might be tasks for which this choice is suboptimal. One way to estimate the frequency band to which one should apply a spectral filter is via a polynomial regression and then determine where the derivative is maximal. Note that it is efficient to calculate the eigenvectors around an arbitrary location of the spectrum, e.g., with the "shift-invert mode" of `scipy/ARPACK` (Lehoucq et al., 1998).

Due to the many possible design decisions of spectrally parametrized filters, the neglect of spectral filters in prior work, and the lack of appropriate benchmarks, it was not possible to ablate all the details. We expect that future work will discuss the specific building blocks in greater detail.

## L  Broader Impact

We expect that $S^2$GNNs will have similar societal implications as other model developments like Convolutional Neural Networks (CNNs) (LeCun et al., 1989), LSTMs (Hochreiter & Urgen Schmidhuber, 1997), transformers (Vaswani et al., 2017), or modern Graph Neural Networks (Gilmer et al., 2017). Since such models may be used as building blocks in architectures for predictive tasks, generative modeling, etc., they have a wide range of positive and negative implications. Nevertheless, we expect that $S^2$GNNs will not have more negative implications than other machine learning model innovations.

## M  Experimental Results

This section provides further details on the experimental setup (§ M.1), the computational cost (§ M.3), and graph constructions with additional experimental results for the clustering tasks (§ M.6); likewise we provide details for the distance regression (§ M.7), arXiv-year (§ M.8), and provide nodes on the graph construction in TPUGraphs (§ M.10). Note that the sections on clustering (§ M.6) and distance regression (§ M.7) also contain ablations and further insights.

### M.1  Experimental Details

**Implementation.** The code base is derived from Cao et al. (2023), which on the other hand derive the code of Rampášek et al. (2022). The implementation heavily relies on PyTorch geometric (Fey & Lenssen, 2019).

**Datasets.** We collect the main statistics, including licenses, for the datasets in Table 6. The provided code will download all datasets along with the experiment execution, except for TPUGraphs, where one should follow the official instructions. Due to the high variation in results, we merge all "layout" datasets and present the results on this joint dataset. We use the fixed public splits for all experiments and proceed accordingly for our datasets (see § M.6 and § M.7).

**Hyperparameters.** While we provide full parameters for all experiments and models in our code, we gather an overview of the used $S^2$GNNs variants here. The parameters were determined through cascades of random search throughout the development of the method. We list the most important parameters in Table 7.

Table 6: Dataset statistics and licenses.

| Name | # of graphs | Average # of nodes | Average # of edges | Task | License |
|------|-------------|--------------------|--------------------|------|---------|
| Peptides func (Dwivedi et al., 2022) | 15,535 | 150.9 | 307.3 | graph multi-label classification | CC BY-NC 4.0 |
| Peptides struct (Dwivedi et al., 2022) | 15,535 | 150.9 | 307.3 | graph regression | CC BY-NC 4.0 |
| CLUSTER (Dwivedi et al., 2023) | 12,000 | 117.2 | 4,301.7 | node classification | CC-BY 4.0 |
| LR-CLUSTER (**ours**) | 12,000 | 896.9 | 6,195.1 | node classification | CC-BY 4.0 |
| Tree Distance regression (**ours**) | 55,000 | 749.2 | 748.2 | node regression | CC-BY 4.0 |
| DAG Distance regression (**ours**) | 55,000 | 748.6 | 821.8 | node regression | CC-BY 4.0 |
| Oversquashing extended (derived from (Di Giovanni et al., 2023a)) | 730 | 43.8 | 231.9 | node classification | CC-BY 4.0 |
| Associative recall small (derived from (Poli et al., 2023)) | 26,000 | 524.7 | 523.7 | node classification | CC-BY 4.0 |
| Associative recall 30k (derived from (Poli et al., 2023)) | 11,000 | 30,003.8 | 30,002.8 | node classification | CC-BY 4.0 |
| OGB arXiv (Hu et al., 2020) | 1 | 169,343 | 1,166,243 | node classification | MIT |
| OGB Products (Hu et al., 2020) | 1 | 2,449,029 | 61,859,140 | node classification | MIT |
| TPUGraphs (Phothilimthana et al., 2023) | $\approx$31,000,000 | $\approx$6,100 | NA | graph ranking | Apache License |

**Usage of external results.** The performance of baselines is commonly taken from leaderboards and the respective accompanying papers. This specifically includes the results in Table 1, Table 2, and Table 11.

**Setup.** For clustering (§ M.6), distance regression (§ M.7), and arXiv-year (§ M.8) we report the detailed setup in the respective sections. For the other tasks, the relevant details are:

- **Peptides:** We follow the setup and implementation of Rampášek et al. (2022). That is, we train for 250 epochs with a batch size of 200. We rerun experiments on 10 random seeds.

- **Over-squashing:** We derive the setup from Di Giovanni et al. (2023a). In the main part (Fig. 5), for the GCN, we report the numbers of their Figure 3 for a GCN on "Clique Path" graphs. For the spectral filter, we actually consider the more challenging setting where we do not train one model per graph size. Instead, we train one model for all sequence lengths. The task is to retrieve the correct of five possible classes on the other end of the graph. In the extended experiment of Fig. 12, we compose the dataset of "Clique Path" and "Ring" graphs (see Di Giovanni et al. (2023a)). To avoid $m = \mathcal{O}(n^2)$, we limit the fully connected clique to 15 nodes. For training and validation, we enumerate all graphs with even $n \in \{4, 6, \ldots, 50\}$ and train for 500 epochs. For test, we enumerate the graphs with even $n \in \{52, 54, \ldots, 100\}$. We rerun experiments on 10 random seeds.

- **Associative recall:** We construct one dataset consisting of key-value sequences of length 20 to 999. As Poli et al. (2023), we use a vocabulary of 30. We sample 25,000/500 random graphs for train/validation. For the test set, we randomly generate 500 graphs for the sequence lengths of 1,000 to 1,199. We train for 200 epochs. In the experiment with validation/test sequence length 30k (Table 2), we generate 10,000 training graphs of length 29,500 to 30,499 and finetune $S^2$GNNsGCN from the smaller setup. We rerun experiments on 10 random seeds.

Table 7: Important $S^2$GNNs specific hyperparameters and runtimes. The times for the EVD cover the respective dataset entirely.

| Dataset | # MP layers | # spec. layers | Dim. $d$ | # spec. filters per layer | # eigenvectors $k$ / frequency cutoff $\lambda_{\text{cut}}$ | Spectral NN | Train time | EVD time | GPU | Notes |
|---------|-------------|----------------|----------|---------------------------|--------------------------------------------------------------|-------------|-----------|----------|-----|-------|
| Peptides-Func | 3 | 3 | 224 | 128 | $\lambda_{\text{cut}} = 0.7$ | ✗ | 1 h | 2 min | 1080Ti | |
| Peptides-Struct | 3 | 1 | 260 | 260 | $\lambda_{\text{cut}} = 0.7$ | ✗ | 1 h | 2 min | 1080Ti | |
| CLUSTER | 18 | 17 | 64 | 32 | $\lambda_{\text{cut}} = 1.3$ | ✗ | 1.2 h | 4 min | 1080Ti | |
| LR-CLUSTER (**ours**) | 4 | 1 | 128 | 128 | $k = 10, \lambda_{\text{cut}} = 0.05$ | ✗ | 20 min | 4 min | 1080Ti | |
| Distance regression (**ours**) | 5 | 4 | 236 | 236 | $k = 50, \lambda_{\text{cut}} = 0.1$ | ✗ | 3 h | 1.5 h | A100 | 1080Ti possible with smaller batch size |
| Oversquashing extended | 0 | 1 | 16 | 16 | $k = 20, \lambda_{\text{cut}} = 0.05$ | ✗ | 3 min | 3 s | 1080Ti | |
| Associative recall | 3 | 3 | 224 | 224 | $k = 10, \tilde{\lambda}_{\text{cut}} = 10$ | ✓ | 3 h | closed form | 1080Ti | eigenvalue transform (Eq. 33) & exponential window |
| arXiv-year | 4 | 2 | 256 | 256 | $k = 100, \lambda_{\text{cut}} = 0.05$ | ✓ | 1 h | 5 min | 1080Ti | |
| Open Graph Benchmark Products | 6 | 2 | 256 | 164 | $k = 100 \to \lambda_{\text{cut}} \approx 0.056$ | ✓ | 11 h | 26 min | A100 | eigenvalue transform (Eq. 32) |
| TPU Graphs | 3 | 1 | 128 | 64 | $k = 100, \lambda_{\text{cut}} = 0.05$ | ✓ | 40 h | 4 h | A100 | transformer-based $\hat{g}$ |

- **OGB Products:** Even though full-graph training with 3 layers GCN plus one spectral layer fits into a 40 GB A100 GPU, we find that batched training works better. We randomly divide the graph during training into 16 parts and train a 6-layer S²GAT with spectral layers after the second and last message passing step. Inference is performed on the entire graph at once. We rerun experiments on 5 random seeds.

- **TPUGraphs:** This is the only dataset where we use a transformer to model $\hat{g}$ instead of the procedure detailed in § 3.2.1. We fix the number of eigenvectors to $k = 100$ and do not apply any windowing. Due to the large variation of results, we merge all "layout" tasks into a single dataset. Since the default graph construction is not able to express all relevant information, we adapt it as detailed in § M.10, however, the empirical impact was small. TPUGraphs "layout" consists of a few hundred distinct graph structures with a large variation on the node-level configuration/features. We sample 10,000 configurations for each graph structure of each "layout" sub-split. Here, we introduce two batch dimensions: (1) batching over multiple graphs and (2) batching over the configurations. In each training step of the 1,000 epochs, we sample a small subset of configurations per graph structure and apply a pairwise hinge loss to rank the configurations. We do not perform random reruns due to the computational cost.

## M.2 Qualitative Experiments

In Fig. 6 and Fig. 8, we provide qualitative insights about the approximation of filters and ringing.

In Fig. 6, we construct a true filter by adding a discontinuous filter at $\lambda = 0$ and a polynomial filter of order 3. For the spectral part, we use the true filter values and fit a Chebyshev polynomial on the remaining part. We then plot the response of the true filter and its approximations on a path graph with 21 nodes and $L = I - D^{-1}A$.

Similarly, in Fig. 8, we use a path graph with 100 nodes and $L = I - D^{-1}A$. We then construct a perfect low pass ($k = 25$) and approximate a rectangular wave.

## M.3 Computational Cost

We report the computational cost for the experiments in Table 7 for a single random seed. On top of the pure cost of reproducing our numbers, we conducted hyperparameter searches using random search. Partially, we required 100s of runs to determine good parameter ranges. A generally well-working approach was first to reproduce the results of the best available MPGNN in prior work. Thereafter, we needed to assess how likely additional capacity would lead to overfitting. Usually, we reduced the number of message-passing steps, added the spectral filter, and determined appropriate values for the number of eigenvectors $k$. In the light of overfitting, it is a good idea to lower the number of Gaussians in the smearing of the filter parametrization (§ 3.2.1), introduce bottle-neck layers (§ J), and use fewer spectral filters than hidden dimensions.

**Runtime with precalculated eigenvectors.** In Fig. 18, we contrast the runtime cost of a spectral convolution with spatial messages passing on ogb-arXiv (170k nodes) of Hu et al. (2020), using an Nvidia GTX 1080Ti. This essentially compares a sparse matrix multiplication (adjacency matrix)

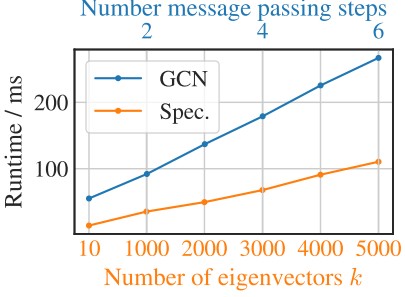

Figure 18: Runtime comparison on arXiv (w/o EVD) using $k = 2,500$ for the spectral filter.

with matrix multiplications on dense "tall and skinny" matrices (GFT). we find that one GCN-layer here is as costly as a spectral filter with approx. $k = 2,500$ eigenvectors.

**Large-scale benchmarks.** On the large-scale datasets OGB Products and TPUGraphs, we perform full-graph training (without, e.g., segment training (Cao et al., 2023)) using 3 DirGCN layers inter-layered with spectral filters targeting a pair-wise hinge loss. The spectral GNN uses the Magnetic Laplacian to incorporate direction. The spatial MPGNN closely resembles the model of Rossi et al. (2023), except that we half the dimension for the forward and backward message passing and concatenate the result. We shrink the dimensions to model the direction at a very low cost. We conclude that S²GNNs can be very practical even if applied at scale and can effectively model long-range interactions also on large graphs.

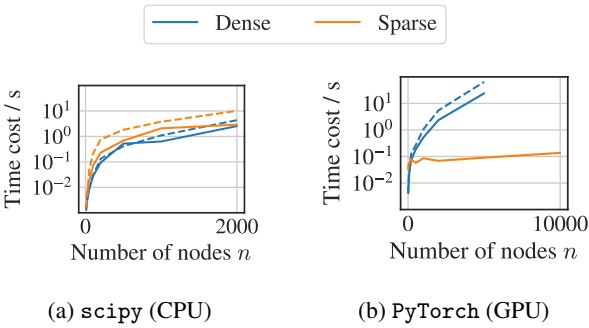

(a) `scipy` (CPU)           (b) `PyTorch` (GPU)

Figure 19: Runtime of partial eigendecomposition $k = 25$ of Erdős Rényi graph with average degree 5. Dashed mark directed/Hermitian Laplacian.

**Eigendecompositon.** We show the computational cost for the eigendecomposition of a random Erdős Rényi graph (every edge has equal likelihood to be drawn). We use `scipy` (CPU) and `PyTorch` (GPU) with default arguments. For the sparse decomposition with `PyTorch`, we use the `svd_lowrank` method. Note that the default parameters for `PyTorch` are usually leading to large numerical errors. Fig. 19 demonstrates that the cost of the eigendecomposition is manageable. For large graphs like ogbn-products (2.5 mio. nodes), the EVD takes around 30 minutes with $k = 100$ on 6 CPU cores of an AMD EPYC 7542. Note that the default parameters of the eigensolver allow for 1000s of iterations or until the error in the 32-bit float representation achieves machine precision.

### M.4 S²GNN Aggregation Ablation

In the main body we present two ways to combine a spatial and spectral filter: An *additive combination* (Eq. 1) and an arbitrary sequence of filters (Eq. 2). In this section, we perform an ablation analysis on the peptides-func benchmark and report the results in Table 8.

Instead of summation of the spatial and spectral parts, **concatenation** is another possible option. Getting input features $\boldsymbol{H}^{(l-1)} \in \mathbb{R}^{n \times d^{(l-1)}}$, we design $\mathrm{Spectral}^{(l)}$ and $\mathrm{Spatial}^{(l)}$ to map to $\mathbb{R}^{n \times d^{(l)}/2}$ and update the embeddings as

$$\boldsymbol{H}^{(l)} = \mathrm{Spectral}^{(l)}(\boldsymbol{H}^{(l-1)}; \boldsymbol{V}, \boldsymbol{\lambda}) \,\|\, \mathrm{Spatial}^{(l)}(\boldsymbol{H}^{(l-1)}; \boldsymbol{A}). \tag{34}$$

Additionally, we consider **normalization** of the addends at the end of each embedding update, dividing by $1/\sqrt{2}$ (concatenation w/ residual) and $1/\sqrt{3}$ (summation w/ residual) as an attempt to keep the variance constant.

Inspired by recent advancements in state space models like Mamba (Gu & Dao, 2023), we also consider modeling an update step in a similar way, identifying the convolutional part with $\mathrm{Spatial}^{(l)}$ and the SSM part with $\mathrm{Spectral}^{(l)}$.

The following table shows results for the different design choices to combine the spatial and spectral parts, with all hyperparameters being precisely the ones reported in Table 7 for peptides-func.

Table 8: Ablation of different aggregation functions on the peptides-func benchmark, with our PE.

| Aggregation | Normalization | # params | Test AP ($\uparrow$) |
|---|---|---|---|
| Concat | ✗ | 322k | $0.6827 \pm 0.0055$ |
| | ✓ | 322k | $0.6783 \pm 0.0023$ |
| Sum | ✗ | 323k | $0.7235 \pm 0.0059$ |
| | ✓ | 323k | $0.7171 \pm 0.0070$ |
| Mamba-like | N/A | 474k | $0.7073 \pm 0.0081$ |
| Sequential | N/A | 323k | $\mathbf{0.7311 \pm 0.0066}$ |

## M.5 Number of Eigenvectors Ablation on Peptides-Func

In Fig. 20, we ablate the number of eigenvectors on the real-world dataset peptides-func. Since we here limit the number of eigenvalues by cut-off frequency $\lambda_{\text{cut}}$, we report the average number of eigenvectors.

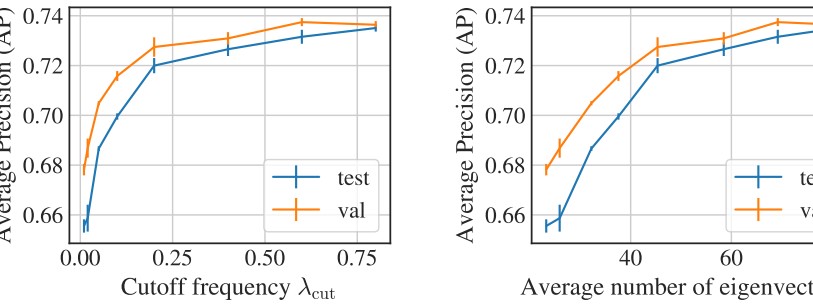

Figure 20: Average precision vs. number of used eigenvalues on the peptides-func long-range benchmark task via frequency cutoff $\lambda_{\text{cut}}$.

## M.6 Clustering Tasks

We use a clustering task `LR-CLUSTER` based on Gaussian Mixture Models (GMMs), which requires long-range interactions to measure the ability of S²GNN to spread information within clusters and consider the original `CLUSTER` task from Dwivedi et al. (2023) based on Stochastic Block Models (SBMs) in order to measure the ability to discriminate between the clusters. The differences are apparent from an illustration of some exemplary graphs in Fig. 21 & 22. While `LR-CLUSTER` has long-range interactions, the challenge of `CLUSTER` is to discriminate between the clusters. Without the arrangement of nodes, colors, and different edge weights, for `CLUSTER`, it is virtually impossible to discriminate the clusters by visual inspection.

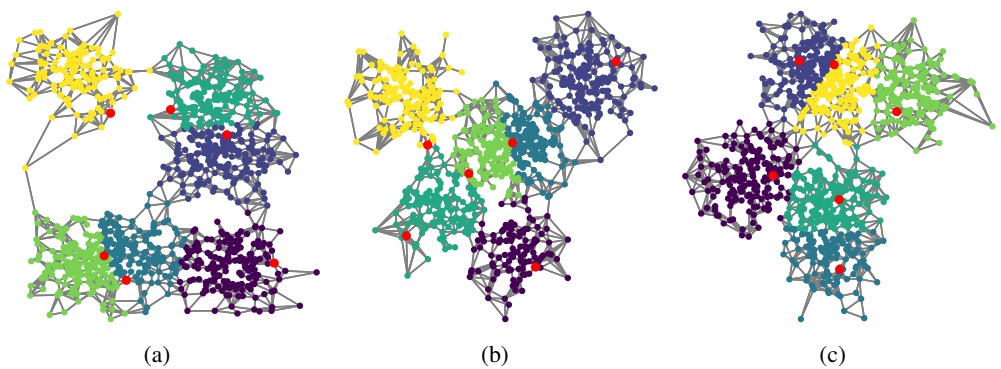

(a)  (b)  (c)

Figure 21: Examples of generated graphs for the `LR-CLUSTER` task (GMM). Labeled nodes are marked red.

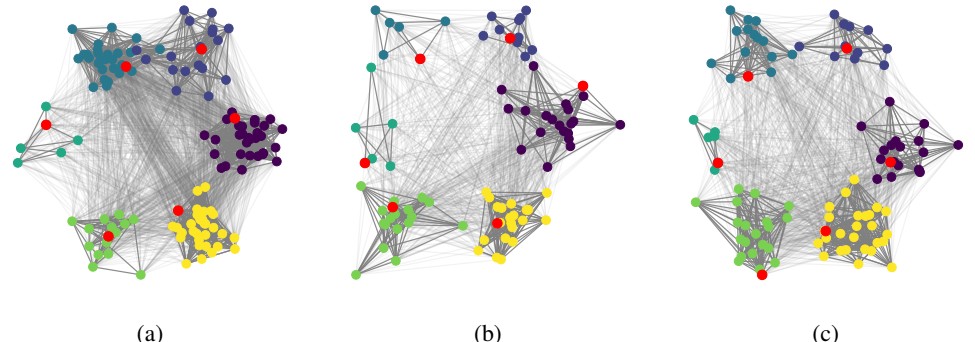

Figure 22: Examples of generated graphs for the CLUSTER task (SBM). Labeled nodes are marked red. Edges within clusters are highlighted.

Nevertheless, we find that the spectral filter is well aligned with the cluster structure in these tasks. We plot this some exemplary filter in Fig. 23. The findings match the explanations of § 4.1 also for CLUSTER.

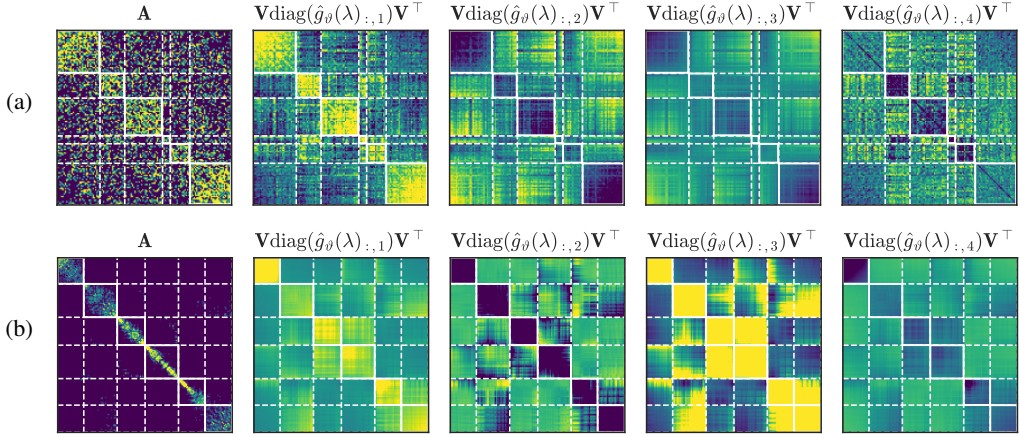

Figure 23: SBM-based (a), visualized in Fig. 21a, and *our* GMM-based (b), visualized in Fig. 22a, graphs along with four learned filters. Large entries are yellow, small are blue, and white lines denote clusters.

In the remainder of the section, we provide full details of the experiment setups. Moreover, we provide additional results not presented in the main text, including ablations.

### M.6.1 GMM Clustering LR-CLUSTER

**Setup.** To sample an input graph, we start by generating $C = 6$ $p$-dimensional cluster centers $\mu_c \sim U[0, 10]^p$ for $c \in \{0, \ldots, C - 1\}$ (we use $p = 2$). Next, we draw $n_c \in \{100, \ldots, 199\}$ points $x_{ic} \sim \mathcal{N}(\mu_c, 4I_p)$ which will represent the nodes of the graph. Subsequently, we update the class memberships such that every point is in its most likely class according to the underlying probabilistic model. Finally, we connect each node $v$ to its $e_v \sim U(\{1, \ldots, 10\})$ closest neighbors by Euclidean distance $\| \cdot \|_2$. This whole procedure is repeated until the generated graph is connected. We then discard the location information and only keep the graph structure. In this way, we generate graphs of an average diameter of $\approx 33$. See Fig. 21 for depictions of example graphs.

Apart from the graph generation procedure, we adhere closely to Dwivedi et al. (2023): We introduce input features in $\{0, 1, 2, \ldots, C\}$, where a feature value of $c = 1, \ldots, C$ corresponds to the node

Table 9: Accuracy on the GMM clustering task for varying number of eigenvectors $k$, using 4 GCN layers and one spectral layer in the end.

| $k$ | 0 (MPGNN) | 1 (Virtual Node) | 2 | 3 | 4 |
|---|---|---|---|---|---|
| S$^2$GCN | $0.4546 \pm 0.0002$ | $0.4646 \pm 0.0001$ | $0.6786 \pm 0.0010$ | $0.7429 \pm 0.0026$ | $0.7971 \pm 0.0008$ |
| S$^2$GCN (+ PE) | $0.4546 \pm 0.0002$ | $0.4642 \pm 0.0007$ | $0.7221 \pm 0.0008$ | $0.7860 \pm 0.0005$ | $0.8202 \pm 0.0011$ |

| 5 | 6 | 7 | 8 | 9 | 10 |
|---|---|---|---|---|---|
| $0.8322 \pm 0.0004$ | $0.8511 \pm 0.0008$ | $0.8510 \pm 0.0008$ | $0.8519 \pm 0.0005$ | $0.8517 \pm 0.0006$ | $0.8513 \pm 0.0018$ |
| $0.8440 \pm 0.0006$ | $0.8538 \pm 0.0012$ | $0.8548 \pm 0.0011$ | $0.8546 \pm 0.0002$ | $0.8545 \pm 0.0005$ | $0.8554 \pm 0.0005$ |

Table 10: Accuracy on the GMM clustering task for varying number of MP layers, while comparing a purely spatial GCN model to S$^2$GCN with one spectral layer added in the end.

| | 2 | 3 | 4 | 5 |
|---|---|---|---|---|
| GCN | $0.2700 \pm 0.0002$ | $0.3557 \pm 0.0000$ | $0.4544 \pm 0.0003$ | $0.5521 \pm 0.0001$ |
| GCN (+ PE) | $0.2684 \pm 0.0005$ | $0.3552 \pm 0.0015$ | $0.4550 \pm 0.0004$ | $0.5526 \pm 0.0006$ |
| S$^2$GCN | $0.8517 \pm 0.0003$ | $0.8520 \pm 0.0008$ | $0.8518 \pm 0.0005$ | $0.8512 \pm 0.0002$ |
| S$^2$GCN (+ PE) | $0.8547 \pm 0.0007$ | $0.8550 \pm 0.0010$ | $0.8552 \pm 0.0015$ | $0.8539 \pm 0.0010$ |

| 6 | 7 | 8 | 9 | 10 |
|---|---|---|---|---|
| $0.6367 \pm 0.0001$ | $0.7013 \pm 0.0001$ | $0.7448 \pm 0.0003$ | $0.7708 \pm 0.0007$ | $0.7860 \pm 0.0004$ |
| $0.6387 \pm 0.0012$ | $0.7104 \pm 0.0011$ | $0.7609 \pm 0.0009$ | $0.7931 \pm 0.0005$ | $0.8135 \pm 0.0007$ |
| $0.8512 \pm 0.0008$ | $0.8509 \pm 0.0008$ | $0.8511 \pm 0.0003$ | $0.8504 \pm 0.0009$ | $0.8509 \pm 0.0006$ |
| $0.8552 \pm 0.0008$ | $0.8542 \pm 0.0004$ | $0.8545 \pm 0.0008$ | $0.8536 \pm 0.0013$ | $0.8542 \pm 0.0008$ |

being in class $c - 1$ and a feature value of 0 means that the class is unknown and has to be inferred by the model. Only one node $v_c$ per class is randomly chosen to be labeled and all remaining node features are set to 0. The output labels are defined as the class labels. We use weighted cross entropy loss for training and class-size-weighted accuracy as a target metric. We generate 10,000 training and 1,000 val/test graphs each and report the average $\pm$ standard deviation over 3 random reruns.

**Models.** As an underlying spatial model baseline, we use a vanilla GCN (Kipf & Welling, 2017). We compare this to S$^2$GCN, only applying one spectral convolution immediately before the last spatial layer. We investigate the influence of the number $k \in \{0, 1, \ldots, 10\}$ of eigenvectors to be taken into account with 4 spatial layers, with $k = 0$ indicating the absence of a spectral layer (see Fig. 10a, and Table 9 for the underlying data). We also vary the number of spatial MP layers from 2 to 10 and compare the performance of a purely spatial GCN to the corresponding S$^2$GCN with one spectral convolution (see Fig. 10b, and Table 10 for the underlying data).

Throughout all evaluations, we maintain a consistent hyperparameter configuration: Specifically, we use an inner dimension of 128, GELU (Hendrycks & Gimpel, 2016) as an activation function, no dropout, and residual connections for all spatial and spectral layers. For the spectral layer, we implement the gating mechanism $f_\theta^{(l)}$, but abstain from a neural network in the spectral domain (§ 3.2.2), bottlenecks, or parameter sharing. We train for 50 epochs with a batch size of 50, using the AdamW optimizer (Loshchilov & Hutter, 2019) with a base learning rate of 0.003, a weight decay of 0.0001, a cosine scheduler and 5 warmup epochs.

**Further discussion.** The clustering task comes naturally to S$^2$GCN, as a spectral layer can simulate certain variations of spectral clustering (von Luxburg, 2007): Suppose $\boldsymbol{H}^{(l-1)} \in \mathbb{R}^{n \times C}$ is a one-hot encoding of the cluster labels, i.e. $\boldsymbol{H}_{v,c}^{(l-1)} = \delta_{v,v_c}$, with $c \in \{1, \ldots, C\}$ and $v_c$ being the unique labeled node per class. In its simplest form, taking $\hat{g}_\vartheta^{(l)}(\boldsymbol{\lambda}) \equiv 1$ and $f_\theta^{(l)} \equiv \mathrm{id}$, the spectral layer Spectral$^{(l)}$ from Eq. 3 turns into $\boldsymbol{H}^{(l)} = \boldsymbol{V}\boldsymbol{V}^\top \boldsymbol{H}^{(l-1)}$. Hence, $\boldsymbol{H}_{v,c}^{(l)} = \boldsymbol{V}_{v,:}^\top \boldsymbol{V}_{v_c,:}$ encodes a notion of similarity between a node $v$ and each labeled node $v_c$. This relates to the Euclidean distance $\|\boldsymbol{V}_{v,:} - \boldsymbol{V}_{v_c,:}\|_2 = \sqrt{\|\boldsymbol{V}_{v,:}\|_2^2 + \|\boldsymbol{V}_{v_c,:}\|_2^2 - 2\boldsymbol{V}_{v,:}^\top \boldsymbol{V}_{v_c,:}}$ which is more typically used for spectral clustering.

### M.6.2 SBM Clustering CLUSTER **(Dwivedi et al., 2023)**

**Setup.** We conduct an ablation study on the original CLUSTER task (Dwivedi et al., 2023), which uses a similar setup to our GMM clustering task, however drawing from a SBM instead: For each cluster, $n_c \in \{5, \ldots, 35\}$ nodes are sampled. Nodes in the same community are connected with a probability of $p = 0.55$, while nodes in different communities are connected with a probability of $q = 0.25$. While there is no need for long-range interactions in this task, considering that the average diameter of the graphs is just $\approx 2.17$, separating the clusters is much harder than in the GMM clustering task (see Fig. 23 for example adjacency matrices from the SBM and GMM models). We use weighted cross entropy loss for training and class-size-weighted accuracy as a target metric. We report the average $\pm$ standard deviation over 3 random reruns.

**Models.** In our ablation study, we consider GCN (Kipf & Welling, 2017), GAT (Veličković et al., 2018), and GatedGCN (Bresson & Laurent, 2018) as MPGNN baselines, following a setup similar to Dwivedi et al. (2023). We consider models with 4 layers (roughly 100k parameters) and 16 layers (roughly 500k parameters), while keeping most hyperparameters the same as in the benchmark, including inner dimension, dropout, and the number of heads for GAT. However, our reported baseline results and parameter counts differ slightly as we are using a different post-MP head, where we maintain a constant dimension until the last layer, in contrast to Dwivedi et al. (2023) who progressively shrink the inner dimension. We construct the corresponding $S^2$GNNs by modifying each baseline model, replacing the $3^{\text{rd}}$ and the $5^{\text{th}}/15^{\text{th}}$ layers with spectral layers, ensuring a roughly equivalent parameter count. Additionally, each model is optionally supplemented by our positional encodings PE (§ 3.2.4).

We further conduct a hyperparameter search on the most promising base MPGNN candidate, GatedGCN, which leads to an optimized version of $S^2$GNN. This optimized model has 18 spatial MPGNN layers, spectral layers between all spatial layers, and additional RWSE encodings. The inner dimension is adjusted to keep the model well below a parameter budget of 500k. Finally, we also evaluate $S^2$GCN and $S^2$GAT using these hyperparameter settings.

Throughout all evaluations, we use GELU (Hendrycks & Gimpel, 2016) as an activation function, residual connections for all spatial and spectral layers, and implement the gating mechanism $f_\theta^{(l)}$ without employing a neural network in the spectral domain. We use a batch size of 128 for training the 4-layer models and 64 for all other models. For

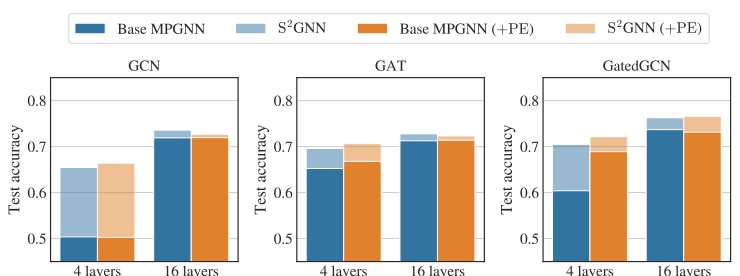

Figure 24: Effects of the spectral part on SBM clustering performance for different base architectures.

the spectral layer, we use the partial eigendecomposition corresponding to the lowest $k = 50$ eigenvalues ($k = 100$ for the optimized $S^2$GNN versions), spectral normalization, and $\lambda_{\text{cut}} = 1.3$. For the optimized models, we employ parameter sharing with 128 heads, and a bottleneck of 0.25 in feature gating. We use 8 attention heads for all GAT versions in accordance with Dwivedi et al. (2023) (in-

Table 11: Results on the CLUSTER task (Dwivedi et al., 2023). Transformer models that outperform our $S^2$GatedGCN are underlined.

| | Model | Accuracy ($\uparrow$) |
|---|---|---|
| **Transformer** | ARGNP Cai et al. (2022) | $0.7735 \pm 0.0005$ |
| | GPS Rampášek et al. (2022) | $0.7802 \pm 0.0018$ |
| | TIGT Choi et al. (2024) | $0.7803 \pm 0.0022$ |
| | GPTrans-Nano Chen et al. (2023) | $0.7807 \pm 0.0015$ |
| | Exphormer (Shirzad et al., 2023) | $0.7807 \pm 0.0004$ |
| | EGT (Hussain et al., 2022) | $\underline{0.7923 \pm 0.0035}$ |
| | GRIT (Ma et al., 2023) | $\underline{0.8003 \pm 0.0028}$ |
| **GNN** | GatedGCN | $0.7608 \pm 0.0020$ |
| | $S^2$GatedGCN (**ours**) | $0.7808 \pm 0.0005$ |

Table 12: Ablation results on the SBM clustering task (Dwivedi et al., 2023). The best mean test accuracy is bold, second is underlined.

| MPGNN base | # total layers | Inner dim. | Spec. filters | Pos. enc. | Dropout | # params | Train accuracy ($\uparrow$) | Test accuracy ($\uparrow$) |
|---|---|---|---|---|---|---|---|---|
| GCN | 4 | 146 | ✗ | ✗ | 0.0 | 109k | $0.5059 \pm 0.0018$ | $0.5037 \pm 0.0023$ |
| | | | ✗ | ✓ | 0.0 | 117k | $0.5053 \pm 0.0010$ | $0.5026 \pm 0.0006$ |
| | | | 1 | ✗ | 0.1 | 117k | $0.6492 \pm 0.0009$ | $0.6545 \pm 0.0013$ |
| | | | 1 | ✓ | 0.1 | 125k | $0.6663 \pm 0.0020$ | $0.6640 \pm 0.0021$ |
| | 16 | 172 | ✗ | ✗ | 0.0 | 508k | $0.7354 \pm 0.0009$ | $0.7190 \pm 0.0010$ |
| | | | ✗ | ✓ | 0.0 | 517k | $0.7378 \pm 0.0017$ | $0.7194 \pm 0.0010$ |
| | | | 2 | ✗ | 0.1 | 527k | $0.7535 \pm 0.0011$ | $0.7359 \pm 0.0017$ |
| | | | 2 | ✓ | 0.1 | 536k | $0.7526 \pm 0.0021$ | $0.7269 \pm 0.0011$ |
| | 18 | 124 | 17 | PE+RWSE | 0.2 | 491k | $0.8022 \pm 0.0147$ | $\underline{0.7711 \pm 0.0020}$ |
| GAT | 4 | 152 | ✗ | ✗ | 0.0 | 120k | $0.6705 \pm 0.0008$ | $0.6525 \pm 0.0010$ |
| | | | ✗ | ✓ | 0.0 | 128k | $0.7167 \pm 0.0001$ | $0.6680 \pm 0.0020$ |
| | | | 1 | ✗ | 0.1 | 128k | $0.7093 \pm 0.0007$ | $0.6960 \pm 0.0010$ |
| | | | 1 | ✓ | 0.1 | 136k | $0.7398 \pm 0.0006$ | $0.7065 \pm 0.0007$ |
| | 16 | 176 | ✗ | ✗ | 0.0 | 541k | $0.8537 \pm 0.0025$ | $0.7126 \pm 0.0014$ |
| | | | ✗ | ✓ | 0.0 | 549k | $0.8740 \pm 0.0014$ | $0.7139 \pm 0.0022$ |
| | | | 2 | ✗ | 0.1 | 558k | $0.8723 \pm 0.0013$ | $0.7277 \pm 0.0005$ |
| | | | 2 | ✓ | 0.1 | 567k | $0.8836 \pm 0.0005$ | $0.7232 \pm 0.0010$ |
| | 18 | 120 | 17 | PE+RWSE | 0.1 | 469k | $0.8071 \pm 0.0262$ | $0.7681 \pm 0.0003$ |
| GatedGCN | 4 | 70 | ✗ | ✗ | 0.0 | 106k | $0.6181 \pm 0.0020$ | $0.6039 \pm 0.0019$ |
| | | | ✗ | ✓ | 0.0 | 110k | $0.7292 \pm 0.0031$ | $0.6889 \pm 0.0027$ |
| | | | 1 | ✗ | 0.1 | 90k | $0.6933 \pm 0.0003$ | $0.7050 \pm 0.0001$ |
| | | | 1 | ✓ | 0.1 | 94k | $0.7245 \pm 0.0002$ | $0.7217 \pm 0.0018$ |
| | 16 | 78 | ✗ | ✗ | 0.0 | 505k | $0.8667 \pm 0.0019$ | $0.7369 \pm 0.0011$ |
| | | | ✗ | ✓ | 0.0 | 509k | $0.8753 \pm 0.0257$ | $0.7314 \pm 0.0058$ |
| | | | 2 | ✗ | 0.1 | 464k | $0.8086 \pm 0.0016$ | $0.7627 \pm 0.0010$ |
| | | | 2 | ✓ | 0.1 | 468k | $0.8302 \pm 0.0011$ | $0.7659 \pm 0.0003$ |
| | 18 | 64 | 17 | PE+RWSE | 0.2 | 460k | $0.8202 \pm 0.0024$ | $\mathbf{0.7808 \pm 0.0005}$ |

ner dimension is not expanded but split up), except for the optimized version, which uses 4 heads. For the purely spatial models, we use $p = 0.0$ as dropout (similar to Dwivedi et al. (2023)). We observe this to lead to overfitting for models with spectral layers, for which we set $p \in \{0.1, 0.2\}$. Hyperparameters differing between the compared models are listed in Table 12. We train for 100 epochs using the AdamW optimizer (Loshchilov & Hutter, 2019) with a base learning rate of 0.001, no weight decay, and a cosine scheduler with 5 warmup epochs.

**Results.** Results for the CLUSTER task are presented in Table 12, Table 11 and Fig. 24. Introducing a spectral layer significantly enhances performance on the 4-layer architectures, both with and without positional encodings. The effect is most pronounced on GCN, where replacing just a single GCN layer by a spectral layer boosts accuracy from 0.504 to 0.655. Notably, introducing two spectral layers still has a consistent positive effect on all 16-layer architectures.

## M.7 Distance Regression

**Setup.** We generate directed random trees with one source by sampling trees with $n \in \{500, \ldots, 999\}$ nodes, picking one node at random to declare as a source and introducing edge directions accordingly. To construct random DAGs with long distances, we start from such directed random trees and proceed by adding $\lfloor n/10 \rfloor$ edges at random, choosing each edge direction such that the resulting graph is still a DAG. Additionally, we mark the source node with a node feature. Besides evaluating all models in an in-distribution regime, we also assess the generalization power of the methods by drawing out-of-distribution val/test splits from slightly larger graphs of $n \in \{1000, \ldots, 1099\}$ and $n \in \{1100, \ldots, 1199\}$ nodes each. We use $L^2$ loss for training and $R^2$ as a target metric. We sample 50,000 training and 2,500 val/test graphs each and report the average $\pm$ standard deviation over 3 random reruns.

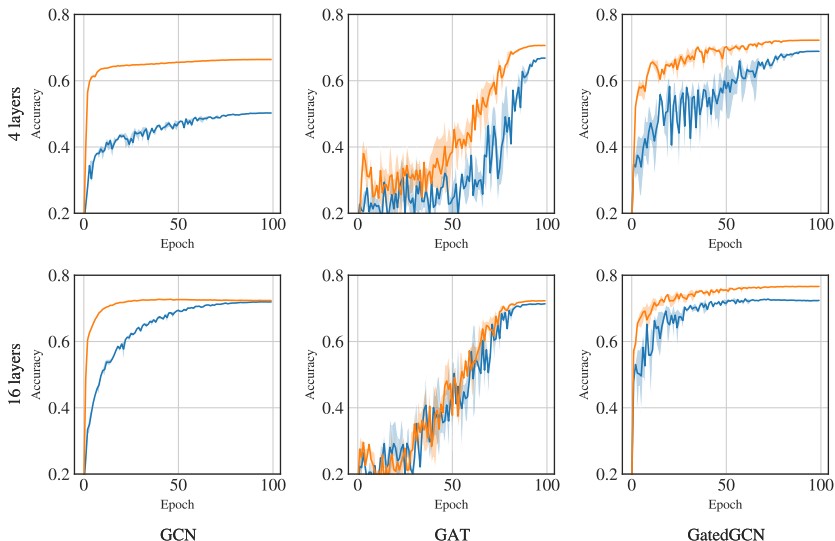

Figure 25: Test accuracy curves for the SBM clustering task. Curves are shown for the models from Table 12 with PE, with the MPGNN baseline and the respective S²GNN.

Table 13: Results on the distance task, with DirGCN as base. The best mean score is bold, second is underlined.

|  | +Spec. filter | PE | in-distribution | | | out-of-distribution | | |
|---|---|---|---|---|---|---|---|---|
|  |  |  | $MAE\,(\downarrow)$ | $RMSE\,(\downarrow)$ | $R^2\,(\uparrow)$ | $MAE\,(\downarrow)$ | $RMSE\,(\downarrow)$ | $R^2\,(\uparrow)$ |
| **DAGs** | ✗ | ✗ | $7.0263 \pm 0.0033$ | $9.0950 \pm 0.0005$ | $0.1915 \pm 0.0001$ | $8.1381 \pm 0.0368$ | $10.7735 \pm 0.0402$ | $0.1214 \pm 0.0066$ |
|  | ✗ | ✓ | $6.8252 \pm 0.0008$ | $8.8636 \pm 0.0024$ | $0.2322 \pm 0.0004$ | $8.0018 \pm 0.0018$ | $10.4432 \pm 0.0017$ | $0.1745 \pm 0.0003$ |
|  | undir. | ✗ | $1.9248 \pm 0.0116$ | $3.2687 \pm 0.0100$ | $0.8956 \pm 0.0006$ | $3.0471 \pm 0.0192$ | $4.9467 \pm 0.0263$ | $0.8148 \pm 0.0020$ |
|  | undir. | ✓ | $1.7384 \pm 0.0039$ | $2.9934 \pm 0.0046$ | $0.9124 \pm 0.0003$ | $2.7950 \pm 0.0041$ | $4.5834 \pm 0.0117$ | $0.8410 \pm 0.0008$ |
|  | direc. | ✗ | $\underline{1.2401 \pm 0.0173}$ | $\underline{2.1600 \pm 0.0340}$ | $\underline{0.9544 \pm 0.0014}$ | $\underline{2.1824 \pm 0.0787}$ | $\underline{3.7694 \pm 0.0710}$ | $\underline{0.8924 \pm 0.0040}$ |
|  | direc. | ✓ | $\mathbf{1.1676 \pm 0.0032}$ | $\mathbf{2.0428 \pm 0.0066}$ | $\mathbf{0.9592 \pm 0.0003}$ | $\mathbf{2.0565 \pm 0.0326}$ | $\mathbf{3.5887 \pm 0.0434}$ | $\mathbf{0.9025 \pm 0.0024}$ |
| **Trees** | ✗ | ✗ | $13.7472 \pm 0.0478$ | $17.3902 \pm 0.0277$ | $0.0958 \pm 0.0029$ | $16.8554 \pm 0.0559$ | $21.6454 \pm 0.1394$ | $0.0144 \pm 0.0127$ |
|  | ✗ | ✓ | $11.6316 \pm 0.0370$ | $15.0123 \pm 0.0249$ | $0.3262 \pm 0.0022$ | $14.9837 \pm 0.0501$ | $19.3659 \pm 0.0610$ | $0.2110 \pm 0.0050$ |
|  | undir. | ✗ | $1.0236 \pm 0.0408$ | $1.7991 \pm 0.1956$ | $0.9902 \pm 0.0020$ | $1.5981 \pm 0.2221$ | $2.7377 \pm 0.4786$ | $0.9839 \pm 0.0053$ |
|  | undir. | ✓ | $1.2887 \pm 0.1195$ | $2.0095 \pm 0.2638$ | $0.9878 \pm 0.0031$ | $1.7184 \pm 0.3288$ | $2.5791 \pm 0.5372$ | $0.9856 \pm 0.0055$ |
|  | direc. | ✗ | $\underline{0.8166 \pm 0.5012}$ | $\underline{1.2224 \pm 0.7600}$ | $\underline{0.9944 \pm 0.0060}$ | $\underline{1.5280 \pm 0.4539}$ | $\underline{2.2942 \pm 0.7592}$ | $\underline{0.9881 \pm 0.0069}$ |
|  | direc. | ✓ | $\mathbf{0.7767 \pm 0.3306}$ | $\mathbf{1.1512 \pm 0.5839}$ | $\mathbf{0.9954 \pm 0.0041}$ | $\mathbf{0.9911 \pm 0.6911}$ | $\mathbf{1.5064 \pm 1.0206}$ | $\mathbf{0.9938 \pm 0.0077}$ |

**Models.** As a MPGNN baseline, we use a five-layer directed version of GCN, DirGCN (Rossi et al., 2023), with three post-message-passing layers, and concatenating instead of averaging over the source-to-target and target-to-source parts. We compare these baselines to S²DirGCN of the form Eq. 2 with four spectral layers, alternating spatial and spectral convolutions and employing residual connections. We benchmark versions of S²DirGCN that ignore edge direction in the spectral convolution against directed versions in which we set $q = 0.001$. In all cases, we use the partial eigendecomposition corresponding to the $k = 50$ lowest eigenvalues. All models are optionally enriched by the positional encodings from § 3.2.4. Throughout all evaluations, we use an inner dimension of 236, GELU (Hendrycks & Gimpel, 2016) as an activation function, and dropout $p = 0.05$. For the spectral layers, we utilize the gating mechanism $f_\theta^{(l)}$, not employing a neural network in the spectral domain, we use spectral normalization, $\lambda_{\text{cut}} = 0.1$, and a bottleneck of $0.03$ in the spectral layer. We train for 50 epochs, using a batch size of 36 and the AdamW optimizer (Loshchilov & Hutter, 2019) with a base learning rate of 0.001, a weight decay of 0.008, and a cosine scheduler with 5 warmup epochs.

**Results.** In Table 13, we show the performance of the different models on DAGs and trees. We observe that the simple MPGNNs are notably surpassed by all versions of S²DirGCN. While S²DirGCN achieves nearly perfect predictions on the tree tasks in both the directed and undirected case, the undirected version is outperformed by the directed version on the DAG tasks. Here, performance also reduces slightly in the out-of-distribution regime. The great performance on the tree

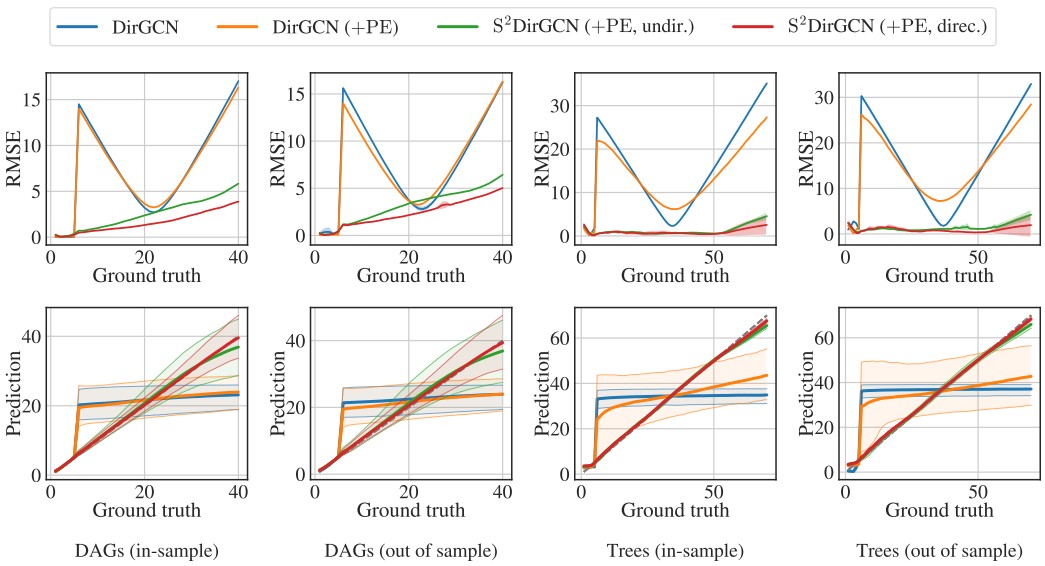

Figure 26: RMSE and 90% prediction intervals for distance predictions by ground truth.

task is due to the fact that trees are *collision-free* graphs (Geisler et al., 2023), where the phase of each eigenvector is $\exp(i2\pi q(d_v + c))$ for each node $v$, with $d_v$ representing the distance to the source node and $c \in \mathbb{R}$ being an arbitrary constant (due to phase invariance of the eigenvector). It is noteworthy that a simple MPGNN with positional encodings, despite having the distances (shifted by $c$) readily available, fails the task, as the information about the phase of the source node cannot be effectively shared among all nodes. In Fig. 26, we compare the distance predictions by the different models. While the prediction of all models is close to perfect below a distance of 5, the spatial MPGNNs are almost unable to distinguish higher distances. By contrast, $S^2$DirGCN predicts reasonable distances regardless of the ground truth, with the absolute error only increasing slowly.

### M.8    Heterophilic arXiv-year (Lim et al., 2021)

**Setup.** We evaluate $S^2$GNN on a large-scale heterophilic dataset, namely *arXiv-year*. *arXiv-year* is based on OGB arXiv (Hu et al., 2020), but instead of paper subject areas, the year of publication (divided into 5 classes) is the prediction target. While there are no long-range interactions in this dataset, preliminary experiments indicated that the phase of the Magnetic Laplacian eigenvectors on its own can also be predictive of the class label. We report average $\pm$ standard deviation over 5 reruns with the splits from Lim et al. (2021), using a different random seed for each run.

**Models.** We use DirGCN (Rossi et al., 2023) as a baseline and largely follow the original setup. However, we observe that using 4 layers (instead of 6) and introducing a dropout of $p = 0.5$ improves baseline performance. Furthermore, we drop the jumping knowledge used by Rossi et al. (2023). We compare this baseline to $S^2$DirGCN with two spectral layers (after the second and third spatial layers) and apply residual connections only for the spectral layers. For the spectral layers, we set $q = 0.0001$ and use the partial eigendecomposition with $k = 100$, a NN in the spectral domain § 3.2.2, no feature gating, and a bottleneck of $0.05$. All other parameters are kept similar to the DirGCN base of Rossi et al. (2023). We train for 2000 epochs using the AdamW optimizer (Loshchilov & Hutter, 2019) with a base learing rate of 0.005, no weight decay, and a cosine scheduler with 50 warmup epochs.

**Results.** We report the results in Table 14. Notably, our $S^2$DirGCN outperforms both our baseline DirGCN as well as the recent FaberNet (Koke & Cremers, 2024), albeit by a very tight margin. However, we found hyperparameter optimizations to be quite noisy, and as such, the resulting performance metrics should be interpreted cautiously. A more comprehensive evaluation of $S^2$GNN's power on heterophilic datasets, potentially with long-range interactions, is left for future work.

Table 14: Results on arXiv-year. Best mean test accuracy is bold, second is underlined.

| Model | Accuracy ($\uparrow$) |
|---|---|
| DirGCN (Rossi et al., 2023) | $0.6408 \pm 0.0026$ |
| FaberNet (Koke & Cremers, 2024) | $\underline{0.6462 \pm 0.0101}$ |
| *DirGCN (**tuned**)* | $0.6450 \pm 0.0025$ |
| *$S^2$DirGCN (**ours**)* | $\mathbf{0.6472 \pm 0.0024}$ |

## M.9 Large-Scale PCQM4Mv2 (Hu et al., 2021)

We show that $S^2$GNNs are very parameter efficient. Even though we only conduct a very rudimentary hyperparameter search, $S^2$GNNs keep up with state of the art approaches. Specifically, we adapt the hyperparameters from peptides-func and achieve comparable performance to the state of the art (excluding external data) with about 3-20% of the number of parameters.

Table 15: Results on PCQM4Mv2 (Hu et al., 2021) (validation).

| Method | MAE ($\downarrow$) | # Parameters | Notes |
|---|---|---|---|
| EGT (Hussain et al., 2022) | 0.0857 | 89.3 mio. | 16 layers |
| GRIT (Ma et al., 2023) | 0.0859 | 16.6 mio. | 16 layers |
| GPS (Rampášek et al., 2022) | 0.0852 | 13.8 mio. | 16 layers |
| TGT-At Hussain et al. (2024) | 0.0671 | 203.9 mio. | 32 layers, pretraining on 3D coordinates (RDKit) |
| **our** $S^2$GNN | 0.0870 | **2.8 mio.** | 5 layers, hyperparameters adapted from peptides-func |

## M.10 TPUGraphs Graph Construction

The "XLA" collections of TPUGraphs contain many constructs that are most certainly suboptimal for a machine-learning-based runtime prediction. However, in our preliminary experiments, we could not show that our graph construction yielded better results in a statistically significant manner. Nevertheless, we include this discussion since it might be insightful.

To understand the challenges with the default graph construction, note that in the TPUGraphs dataset each node represents an operation in the compuational graph of Accelerated Linear Algebra (XLA). Its incoming edges are the respective operands, and the outgoing edges signal where the operation's result is used. Thus, the graph describes how the tensors are being transformed. An (perhaps unnecessary) challenge for machine learning models arises from using `tuple`, which represents a sequence of tensors of arbitrary shapes. In this case, the model needs to reason how the tuple is constructed, converted, and unpacked again. Moreover, directly adjacent tensors/operations can be very far away in the graphs of TPUGraphs.

We identified and manually "fixed" three cases to eliminate this problem largely in the TPUGraphs dataset: Tuple-GetTupleElement, While, and Conditional. Since we could not access the configurations in the HLO protobuf files and C++ XLA extraction code, we decided to perform these optimizations ourselves. However, it might be a better strategy to utilize the existing XLA compiler etc.

Additionally, to the subsequently described graph structure changes, we extract the order of operands from the HLO protobud files. Outgoing edges are assumed to be unordered except for the GetTupleElement operation, where the tuple index is used as order. Moreover, we extracted all features masked in the C++ code and then excluded constant features.

### M.10.1 Tuple-GetTupleElement

The dataset contains aggregations via the XLA Tuple operation that are often directly followed by a GetTupleElement operation. To a large extent, these constructs are required for the subsequently discussed While and Conditional operations. Importantly, the model could not extract the relationships through a tuple aggregation since the `tuple_index` was not included in the default features. Moreover, the resulting tuple hub nodes severely impact the Fourier basis of the graphs (see § 2). We illustrate the graph simplification in Fig. 27 and denote the edge order of incoming edges from top to bottom. The edge order represents the order of operands.

We propose dropping immediate Tuple-GetTupleElement constructs and directly connecting predecessors and successors. For this, we generate a graph solely consisting of direct connections and then resolve multiple consecutive Tuple-GetTupleElement constructs via a graph traversal (depth-first search).

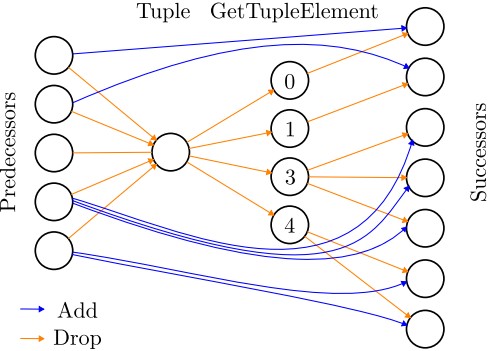

Figure 27: Tuple-GetTupleElement simplification: the Tuple aggregates the output of multiple predecessors/operations and then the GetTupleElement extracts the tensor according to its index (number in respective nodes). We propose dropping immediate Tuple-GetTupleElement constructs and connecting predecessors and successors.

We perform the Tuple-GetTupleElement simplification after dealing with While and Conditionals. However, for the sake of simplicity, we will avoid using tuples in the subsequent explanations for While and Conditional. In other words, the subsequent explanations extend to functions with multiple arguments via the use of tuples.

### M.10.2 While Operation

The While operation has the signature `While(condition, body, init)` where `condition` is a function given the `init` or output of the `body` function. Note that in the TPUGraph construction, `body` as well as `condition` only represent the outputs of the respective function and their operands need to be extracted from HLO.

To avoid hub nodes and to retain the dataflow between operations (important for decisions about the layout), operands and outputs are connected directly. Technically, we am modeling a do-while construct because the condition is not connected to the inputs. Since the successors of the while are of type GetTupleElement, they relabeled to a new node type, signaling the end of a while loop. To support nested while loops, each node in the body is assigned a new node feature signaling the number of while body statements it is part of.

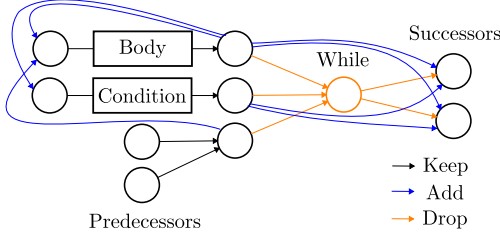

Figure 28: Instead of aggregating everything into a hub node, we propose to connect respective inputs and outputs.

### M.10.3  Conditional Operation

`Conditional(branch_index, branch_computations, branch_operands)` is the most common signature of the Conditional operation, where the integer-valued `branch_index` selects which `branch_computations` is executed with the respective input in `branch_operands`. Similarly to the While operation, we introduce new node types for the inputs of computations and the successors (they are GetTupleElement operations).

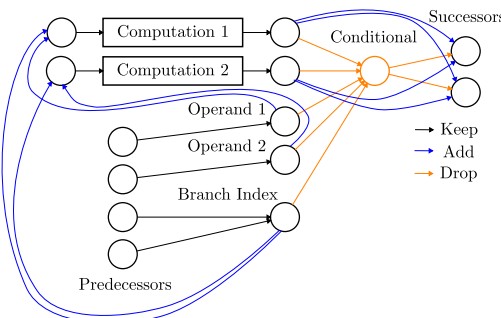

Figure 29: Instead of aggregating everything into a hub node, we propose to connect respective inputs and outputs. Here as an example with two conditional computations.

