# OpenReview forum: "Spatio-Spectral Graph Neural Networks"
_NeurIPS.cc/2024/Conference — NeurIPS 2024 poster_

### Official Review · Reviewer_rAa9 · 2024-06-25

**Soundness:** 4
**Presentation:** 3
**Contribution:** 4
**Rating:** 8
**Confidence:** 4

**Summary:**

The authors are proposing Spatio-Spectral Graph Neural Networks ($S^2GNNs$), a hybrid model that combines (Spatial) Message Passing Neural Networks (MPNNs) and Spectral Graph Neural Networks. The authors argue that, by combining message-passing with spectral filters, the model can better model both local and global information, alleviating phenomena such as over-squashing and under-reaching. The authors discuss the design space of their proposed method and showcase a technique for obtaining position encodings that can make their model more expressive than the 1-WL test. The authors perform multiple experiments on both synthetic and real-world datasets.

**Strengths:**

- The paper is well-written, and the figures and tables are aesthetically pleasing.
- The tackled problem is well-motivated.
- The method is relatively easy to comprehend. The idea is simple and elegant, yet it has very strong empirical results.
- The proposed $S^2 GNN$ is analyzed very well. Including mechanistic in-context learning experiments is very interesting in the context of long-range modeling on graphs. Moreover, the connections between $S^2 GNNs$ and SSMs that the authors make are insightful and could be very interesting to some readers.
- The empirical results on the Peptides datasets are very good. $S^2 GNNs$ obtain state-of-the-art results on peptides-func *without* using Position Encodings, which is very hard to accomplish even with models that consider the graph as being complete (such as Graph Transformers). Moreover, the proposed method uses significantly fewer parameters than other competing models.
- The theoretical analysis is well thought out. The authors also make some remarks about the advantages of having spectral filters, and they provide some very useful intuitions that connect them to using virtual nodes (Fig 3, Fig 4).
- The authors introduce two new datasets - a long-range clustering dataset (\textsc{LR-CLUSTER}) and a long-range regression dataset that is then used to make several observations about their model - such as the ability to solve long-range tasks and the alignment between the clusters and the learned spectral filters.
- The authors also propose a way of obtaining some Laplacian Position Encodings that improve the overall performance and make their model more expressive than the 1-WL test.
- The paper contains an extensive amount of supplementary materials detailing other experiments and proofs for the theoretical results.

**Weaknesses:**

- The paper's main weaknesses are that the main text is somewhat unfocused, and some related work is not mentioned. I will detail in the following:
	- While the authors generally do a very good job of providing valuable theoretical insights regarding their design choices, I would argue that some of them are unnecessary and distract the reader from the main points of the paper. For instance, I believe the discussion about "high-resolution filters for low frequencies" from lines 149-167 to be somewhat hand-wavy and distracting, especially the remark that "as graph problems discretize continuous phenomena, stability to the graph perturbations is a strong domain-agnostic prior". While I don't think that the discussion is bad, I feel that the supplementary materials would be a much better place in which to include it.
	- I believe that Subsection 3.3 (Parametrizing Spectral Filters) is somewhat confusing, especially for a reader who is not very familiar with spectral methods. I did not initially understand the discussion about the window (Lines 247-253), and I believe it should contain more context. The authors should either expand the section or move it to the supplementary materials.
	- The proposed positional encodings are a nice addition, but similar laplacian positional encodings, such as [[1]], have not been discussed as related work. Other works, such as [[2]], are cited in the paper but not in the context of the laplacian positional encodings that they propose.
	- Some other recent methods deal with long-range relationships on graphs and have not been discussed in the related work. Notably, GRED [[3]] uses Linear Recurrent Units to model distant relationships with a mechanism similar to k-hop MPNNs and also draws inspiration from SSMs; AMP [[4]] has a message filtering mechanism and is adaptively increasing the number of layers depending on the dataset; NBA [[5]] is passing messages in a non-backtracking manner, obtaining a slightly better score than $S^2 GCN$ on *peptides-struct* ($0.2424$ v.s. $0.2447$), and PR-MPNN [[6]] rewires graphs adaptively by learning a probability distribution for the adjacency matrices.
- The authors mention on line 69 that the runtime of their method is equivalent to that of an MPNN, but I could not find a table containing a direct comparison. It would be great if this were included somewhere. There are training times on page 35, Table 6, but no comparison with a simple MPNN (such as a GCN or a GIN model).


Overall, I believe that the paper is very strong. The model is conceptually simple but contains novel ideas. The empirical results are very good, and the authors test their method on many different scenarios. I found the mechanistic in-context learning experiments to be particularly interesting. I believe the work could have a considerable impact on how we design GNN architectures going forward.

As stated previously, the main weakness is that the paper feels somewhat unfocused, and some important related work is omitted. This is a somewhat strange complaint to have, but I feel that the scope of the paper is very broad. In my opinion, the overall style of the paper would be better suited for a journal submission if some details from the appendix were moved into the main text. For a conference with more limited space, I would advise the authors to move some of the more in-depth details to the appendix. This is in no way an argument for rejection, I just believe that the paper would be a lot nicer to read if it didn't contain these details in the main text.

I recommend a 7 (Accept), but I will update my score to an 8 (Strong Accept) if the authors expand their related work discussion and provide more details for Subsection 3.3. I would also recommend to the authors that they move some of the details from Section 3 to the Appendix, but not doing so will not affect my final score.



[1]: https://arxiv.org/abs/2110.07875
[2]: https://arxiv.org/pdf/2012.09699
[3]: https://arxiv.org/pdf/2312.01538
[4]: https://arxiv.org/pdf/2312.16560
[5]: https://arxiv.org/pdf/2310.07430
[6]: https://arxiv.org/abs/2310.02156

**Questions:**

- In the graph rewiring literature, the models are often benchmarked on other graph-level datasets (such as QM9, Zinc, or the TUDataset collection). Did the authors try their method on these datasets? I am not requesting these experiments, but they would be a nice addition.
- The Long-Range Graph Benchmark also contains COCO-SP, PascalVOC-SP, and PCQM-Contact. Did the authors try to benchmark their models on these datasets? I'm personally not a fan of COCO-SP and PascalVOC-SP because they're obtained from Computer Vision datasets, so the GNN inductive bias might not be appropriate for them. Still, PCQM-Contact is a molecular dataset, and it would be interesting to see some results on it.

**Limitations:**

- The limitations are addressed in the appendix.

---

> ### Author Rebuttal · Authors · 2024-08-06
>
> We thank the reviewer for the valuable points and positive feedback. We will use the extra space in the camera-ready version to address the reviewer's points.
>
> ---
>
> ## High-resolution filters for low frequencies
>
> We truncate the spectral filter for efficiency reasons mainly, whereas the special choice of a low-pass window is required by neither our theoretical analysis nor by any specifics of its implementation (we state this in ll. 192-193, now moved out of §3.2 directly into §3 for clarity). While we intended ll. 149-167 to provide supplementary intuition as for why low-pass windows might be a sensible default, we do agree that this might distract the reader from initially understanding the essentials of our method. We intend to keep points (1) and (2) as main arguments (except for the point on "as graph problems discretize continuous phenomena [...]") and move the remainder (ll. 158-167) to the appendix.
>
> ---
>
> ## Parametrizing spectral filters
>
> To improve readability, we have rewritten the second sentence (ll. 240-241), explaining our use of the term "Gaussian smearing":
> > As depicted in Fig. 7, we learn a channel-wise linear combination of translated Gaussian basis functions (similar to the "Gaussian smearing" used by Schütt et al., 2017).
>
> Moreover, we have overhauled the window discussion (ll. 247-253) to explain, first, how the Gibbs phenomenon adversely affects learning, and, second, how we alleviate it by overlaying the filters with a window function:
> > We multiply the learned combinations of Gaussians by an envelope function (we choose a Tukey window) that decays smoothly to zero outside the cutoff $\lambda_{\text{cut}}$. This counteracts the so-called "Gibbs phenomenon" (also known as "ringing"): as visualized for a path graph/sequence of 100 nodes in Fig. 8, trying to approximate a spatially-discontinuous target signal using a low-pass range of frequency components results in an overshooting oscillatory behavior near the spatial discontinuity. Dampening the frequencies near $\lambda_{\text{cut}}$ by a smooth envelope function alleviates this behavior. We note that the learned filter may, in principle, overrule the windowing at the cost of an increased weight decay penalty.
>
> ---
>
> ## Additional structure to enhance presentation
>
> To streamline the presentation and reading flow, we will use the extra space to introduce additional structure. Specifically, we have revised the paper to make sure that the overall "story arc" implied by Fig. 2 (core theory in §3.1-3.2, design space considerations in §3.3-3.6) is more closely aligned with the section structure: §3 of the revised manuscript will instead feature two subsections, §3.1 (Theoretical Analysis) and §3.2 (Design Space) that form the umbrella for the previous §3.1-3.2 and §3.3-3.7. This extra structure will allow for better guidance through the considerations around S$^2$GNNs. Naturally, we greatly welcome any additional suggestions on how to improve the presentation.
>
> ---
>
> ## Positional encodings
>
> We thank the reviewer for the helpful suggestions and we have extended the related work section accordingly. We would like to note that LapPE [2] (and accordingly LSPE [1]) breaks permutation equivariance (sign invariance is approximately enforced through augmentations; repeated eigenvalues are ignored). We will use the extra page to explicitly discuss this aspect in the revised manuscript instead of only referring to background literature. Experimentally, we have so far (implicitly) addressed LapPE, e.g., through our evaluations of GPS.
>
> ---
>
> ## Further related work
>
> We thank the reviewer for pointing out additional related work, and we have included them in our manuscript accordingly. Thus, we will implement the requested changes in the camera-ready version.
>
> ---
>
> ## Runtime comparison of message passing and spectral filters
>
> Great suggestion! Of course, the runtime comparison depends severely on the used hardware since it essentially compares a sparse matrix multiplication (adjacency matrix) with matrix multiplications on dense "tall and skinny" matrices (GFT). With OGB-arXiv (170k nodes) as the graph for comparison, we find that one GCN-layer here is as costly as a spectral filter with approx. 2,500 eigenvectors. We provide the detailed plot in the global PDF.
>
> ---
>
> ## Additional dataset: PCQM
>
> Since the primary download of PCQM-contact is currently not available, we report results on PCQM4Mv2 instead (also 3.4 million graphs etc.). Due to the time and resource constraints during the rebuttal phase, we were not able to tune the hyperparameters. Instead, here we manually merged the configuration from S$^2$GNN with the configuration from the Long Range Graph Benchmark. This yields a much smaller yet very effective model for PCQM4Mv2. For the camera-ready version, we intend to scale the model up and also follow the pretraining procedure of TGT-At (without using RDKit coordinates at test time).
>
> | Method | Validation MAE (↓) | # Parameters | Comment |
> |---|---|---|---|
> | EGT (Hussain et al., 2022) | 0.0857 | 89.3 mio. | 16 layers |
> | GRIT (Ma et al., 2023) | 0.0859 | 16.6 mio. | 16 layers |
> | GPS (Rampášek et al., 2022) | 0.0852 | 13.8 mio. | 16 layers |
> | TGT-At (Hussain et al., 2024) | 0.0671 | 203.9 mio. | 32 layers, uses RDKit coordinates, Pretraining on 3D coordinates |
> | S$^2$GNN (*ours*) | 0.0870 | **2.8 mio.** | 5 layers, no hyperparameter tuning |

---

> > ### Comment · Reviewer_rAa9 · 2024-08-08
> >
> > I want to thank the authors for their rebuttal. I believe they have addressed my concerns and other reviewers' concerns well.
> >
> > As per the initial review, I have updated my score to an 8 (strong accept). I am keeping my confidence to 4 since there might be some details I missed or some parts of the paper that I might have misunderstood.
> >
> > Thank you for the paper. I enjoyed reading it, and I believe that the work is very solid.

---

> > > ### Author Response · Authors · 2024-08-11
> > >
> > > We thank the reviewer for raising the score to 8 (strong accept) and the strong support on the concerns of the other reviewers! We value the suggestions very much and will take meticulous care on addressing them in the camera ready version.

---

### Official Review · Reviewer_H69b · 2024-07-08

**Soundness:** 3
**Presentation:** 2
**Contribution:** 2
**Rating:** 6
**Confidence:** 4

**Summary:**

This paper proposes Spatio-Spectral Graph Neural Networks (S²GNNs) to address the limitations of ℓ-step Spatial Message Passing Graph Neural Networks (MPGNNs), such as limited receptive fields and over-squashing. S²GNNs combine spatial and spectral graph filters for efficient global information propagation, offering tighter error bounds and greater expressiveness than MPGNNs. They outperform traditional MPGNNs, graph transformers, and graph rewirings on benchmarks and scale efficiently to millions of nodes.

**Strengths:**

1. Combining spectral and spatial methods is interesting. The proposed method can overcome many problems associated with MPNNs.
2. The theoretical proofs in this paper are sufficient and, in my opinion, sound.
3. This paper offers a lot of intuitive analysis, with examples that help readers understand the concepts.
4. S²GNN performs excellently in multiple tasks and can be extended to various applications. It also demonstrates scalability, running efficiently on graphs with millions of nodes.

**Weaknesses:**

1. This paper's writing and structure can be further improved. Adding descriptions to the notation and including pseudocode would enhance readability.
2. The experiment includes too few baselines. For instance, only GAT and GCN are used as baselines in Tables 3 and 4, while some of the latest methods, such as SpecFormer, which also uses Eigenvector, are not considered.

**Questions:**

1. Please refer to the weaknesses mentioned above.
2. What is the upper bound of the graph size that the EVD used in S²GNN can support? For example, can it handle graphs with billions of nodes, such as papers100m?
3. In real tasks, are the eigenvectors $k$ sensitive to performance? For instance, on products and TPU graphs, would using a larger $k$ improve performance?

**Limitations:**

The authors discussed limitations in the Appendix.

---

> ### Author Rebuttal · Authors · 2024-08-06
>
> We thank the reviewer for their feedback and for acknowledging the theoretical justification along with the ubiquitous possibilities of lifting GNNs to S$^2$GNNs. Furthermore, we thank the reviewer for highlighting that the paper contains a lot of intuitive analysis with examples.
>
> ---
>
> ## W1: Pseudocode and notation table
>
> We provide pseudo-code and a table covering the notations in the global response. If this does not fully address the reviewer's comment, we ask for clarification.
>
> ---
>
> ## W2: Baselines
>
> We include *task-specific state-of-the-art baselines* for each task/experiment (see Table 1 for peptides tasks, Table 2 for associative recall, in Table 10 for CLUSTER, and Table 12 for arXiv-year). In total, we compare to the following *21 baselines* (see paper for full references): (1) GAT  (Velickovic et al., 2018), (2) GCN (Kipf & Welling 2017; Tönshoff et al., 2023), (3) GatedGCN (Bresson & Laurent, 2018), (4) DirGCN (Rossi et al., 2023),  (5) FaberNet (Koke & Cremers, 2024), (6) TIGT (Choi et al., 2024), (7) MGT+WPE (Ngo et al., 2023), (8) GraphMLPMixer (He et al., 2023), Graph ViT (He et al., 2023), (9) GRIT (Ma et al., 2023), (10) DRew-GCN (Gutteridge et al., 2023), (11) PathNN (Michel et al., 2023), (12) CIN++ (Giusti et al., 2023), (13) ARGNP (Cai et al., 2022), (14) GPS (Rampášek et al., 2022), (15) GPTrans-Nano (Chen et al., 2023), (16) Exphormer (Shirzad et al., 2023), (17) EGT (Hussain et al., 2022), (18) Transformer (Vaswani et al., 2017), (19) Transformer w/ FlashAttention (Dao et al., 2022), (20) H3 (Fu et al., 2023), (21) Hyena (Poli et al., 2023).
>
> Following the suggestion, in the revised version of the manuscript, we will also include a comparison to further baselines on OGB Products (Table 3). It should be noted that the largest dataset considered by the suggested SpecFormer is arXiv (170k nodes). Scaling by >10x (Products, 2.5M nodes) would be prohibitively expensive due to SpecFormer's large required fraction of eigenvalues. Furthermore, SpecFormer does not discuss directed graphs and, thus, is not directly applicable to TPUGraphs (Table 4).
>
> ---
>
> ## Q2: Scaling the eigendecomposition
>
> There is no clear scalability limit for the eigendecomposition as calculating $k$ eigenvectors scales linearly in the number of edges $m$. However, it should be noted that the EVD of such large graphs will come at considerable computational cost and one should consider approximate methods. For example, there are open-source implementations for an approximate partial SVD (equivalent to EVD for PSD matrices) that support a hundred million rows and columns: [https://github.com/criteo/Spark-RSVD](https://github.com/criteo/Spark-RSVD).
>
> ---
>
> ## Q3: Performance vs. number of eigenvectors on real tasks
>
> We plot the influence of the number of eigenvectors in the global response (Figure 2) on the real-world dataset peptides-func. Since we here limit the number of eigenvalues by cut-off frequency $\lambda_{\text{cut}}$ (Figure 2a), we report the average number of eigenvectors in Figure 2b.

---

> > ### Comment · Reviewer_H69b · 2024-08-13
> >
> > Thank you for the author's response. Considering the comments from other reviewers, I will raise the score to 6. However, the writing of the paper still needs improvement.

---

> > > ### Author Response · Authors · 2024-08-13
> > >
> > > We appreciate the reviewer's feedback and thank them for updating their score! We would appreciate if the reviewer has any further specific suggestions for improving the paper's content, methodology, or impact, s.t. we can strengthen our contribution and score.

---

### Official Review · Reviewer_EZD5 · 2024-07-12

**Soundness:** 3
**Presentation:** 3
**Contribution:** 3
**Rating:** 6
**Confidence:** 5

**Summary:**

The paper presents Spatio-Spectral Graph Neural Networks (S$^2$GNNs), a novel paradigm that combines spatial and spectral parameters to overcome the limitations of $\ell$-step MPGNNs, notably their restricted receptive fields and over-squashing issues. S$^2$GNNs achieve global information propagation efficiently, surpassing MPGNNs in approximation accuracy, expressivity, and scalability, demonstrated through superior performance on various graph tasks and ability to handle large-scale datasets.

**Strengths:**

1. The paper presents a rigorous theoretical basis and comprehensive analysis, reinforcing the validity of the proposed approach.

2. The proposed method is straightforward and practical, with the added advantage of being easily integrable into a wide range of existing GNN models.

3. The method has been extensively tested across various backbones and datasets of differing types and sizes, demonstrating its robustness and versatility.

**Weaknesses:**

1. **Deficiency in Ablation Studies for Embedding Updates**: Eq.~(1) within the paper that updates to the latent embeddings are achieved through the summation of spectral and spatial layers. This method warrants a broader range of ablation studies, such as exploring concatenation, normalization, or adopting a randomized approach to select between spectral or spatial optimizations like [1], to validate the robustness and efficiency of these embeddings.

2. **Inadequate Comparative Analysis for Over-Squashing and Long-Range Interactions**: The paper's treatment of over-squashing and long-range interactions could be enhanced by incorporating comparative analyses with established methods. Notably, the GCNII[2], which utilizes skip connections and PageRank to facilitate up to 64 layers in a GNN, and methods based on random walks that intensify depth-first search capabilities for extended influence, should be examined. This inclusion would enhance the analysis and provide a more rigorous evaluation of the proposed model’s capabilities in handling long-range interactions.

[1] Chang, Heng, et al. "Not all low-pass filters are robust in graph convolutional networks." Advances in Neural Information Processing Systems 34 (2021): 25058-25071.
[2] Chen, Ming, et al. "Simple and deep graph convolutional networks." International conference on machine learning. PMLR, 2020.

**Questions:**

See above

---

> ### Author Rebuttal · Authors · 2024-08-06
>
> We thank the reviewer for their feedback and for acknowledging the theoretical justification along with the ubiquitous possibilities of lifting GNNs to S$^2$GNNs.
>
> ---
>
> ## W1: Ablation on the combination of spatial and spectral filter
>
> We agree with the reviewer that different combinations may yield beneficiary properties. In the camera-ready version, we will highlight that we focus on the simplest options, showing that the combination is very effective (Eq.  1 & 2). From an approximation-theoretic standpoint, choosing addition moreover recovers the analysis in §3.2 most naturally. Nevertheless, we value the suggestion and present the following comparison of concatenation, addition (Eq. 1), a Mamba-like composition, and arbitrary sequence of filters (Eq. 2), on peptides-func.
>
> | Aggregation | Normalization | # params | Test AP (↑)       |
> |-------------|---------------|----------|-------------------|
> | Concat      | ✗             | 322k     | 0.6827 ± 0.0055   |
> | Concat      | ✓             | 322k     | 0.6783 ± 0.0023   |
> | Sum (Eq. 1)         | ✗             | 323k     | *0.7235 ± 0.0059* |
> | Sum         | ✓             | 323k     | 0.7171 ± 0.0070   |
> | Mamba-like  | -             | 474k     | 0.7073 ± 0.0081   |
> | Sequential (Eq. 2)  | -             | 323k     | **0.7311 ± 0.0066** |
>
> For the normalization, we add the factor $\sqrt{1/2}$ if aggregating two values and $\sqrt{1/3}$ for three values. We use the same hyperparameters as before and solely alter the aggregations. In a different set of experiments, we also tried BatchNorm and GraphNorm, which did not yield significant improvements.
>
> We agree that there are many important aspects, including robustness like in [1]. However, it is not clear how to transfer their approach to our setting since we do not have the option to tie the weights of the spectral and spatial filters. We may try this option if the reviewer can provide more details on the specifics.
>
> ---
>
> ## W2: Comparative Analysis
>
> We do not state that S$^2$GNNs provide the *only* solution to over-squashing, but we would like to note that we did compare our method to a wide range of popular methods that overcome over-squashing, such as various graph transformer models and rewiring approaches. Our theoretical analysis can be extended to the mentioned GCNII [2] with similar conclusions.
> However, we want to highlight that although GCNII vanquishes over-smoothing, the information propagation through the graph is similarly constrained by the graph structure as in many other standard MPGNN architectures like GCN. This implies that GCNII does not effectively alleviate all potential over-squashing issues (i.e., information still needs to pass bottlenecks in the graph). We conducted additional experiments with GCNII on the peptides-func benchmark, using similar hyperparameters as GCN-tuned (Tönshoff et al., 2023) and varying the number of layers from 6 up to 64. While results in the standard 6-layer setting approach performance of GCN-tuned (AP of 0.6860±0.0050 with GCN, 0.6656±0.0004 with GCNII), performance starts to deteriorate after ~16 layers, which we conjecture to be caused by over-squashing.
>
> We will include GCNII in the camera-ready version and appreciate further pointers if the point is not fully addressed.

---

> > ### Comment · Reviewer_EZD5 · 2024-08-12
> >
> > Thank you for your response. All of my concerns have been addressed. I will arise my score accordingly.

---

> > > ### Author Response · Authors · 2024-08-12
> > >
> > > We appreciate the reviewer's feedback and thank them for updating their score! Since we resolved all weaknesses/concerns, we would appreciate if the reviewer has any further suggestions for improving the paper's content, methodology, or impact. We would greatly appreciate their input to strengthen our contribution and score.

---

### Official Review · Reviewer_ohZY · 2024-07-14

**Soundness:** 2
**Presentation:** 2
**Contribution:** 2
**Rating:** 5
**Confidence:** 3

**Summary:**

The proposed method Spatio-Spectral Graph Neural Networks (S2GNNs) combine the spectral GCN and spatial GCNs embeddings linearly. The paper considers a deep dive into spectral filter properties and attempt to motivate the combination with spatial filtering. There are well known results repeated in the method section, which could have been part of background. Also, the contribution towards directed graphs is not clear.

**Strengths:**

1.	The paper considers a deep dive into spectral filter properties and attempt to motivate the combination of spectral filters with spatial filtering.
2.	The linear combination of spectral and spatial embeddings is new.

**Weaknesses:**

1.	There are well known results repeated in the method section. For example, it is established that the spectral filters are permutation equivariant, Locality relates to spectral smoothness.
2.	The paper sometimes fails to connect the dots. For example, MPNNs (spatial) suffer from oversquashing, but at the same time spectral GNNs can handle oversquashing issue. So why do we need to combine them both? Its not clear in Introduction.
3.	Contribution towards the directed graphs is unclear, any spectral filters can be adapted to directed graphs via magnetic Laplacian. The main point of the paper is to demonstrate the superiority against existing methods for undirected settings first.

**Questions:**

GNNs aim to learn data driven filters. “Ideal” filters may NOT be a “good” choice always. Then what is the need of implementing an ideal discontinuous spectral filter discussed in Section 3.2?

In the experiments, did you take the directionality into account in S2GNN? If, yes what is the performance of S2GNN without directionality?

---

> ### Author Rebuttal · Authors · 2024-08-06
>
> We thank the reviewer for their critical thoughts about our work! We are convinced that we conclusively resolved all points brought up by the reviewer. We would highly appreciate a major reevaluation of our submission.
>
> Before going into detail, we want to highlight that our empirical results yield conclusive evidence for the validity of the advocated GNN construction. For example, S$^2$GNNs outperform all prior message passing schemes, transformers, and graph rewirings on the long-range benchmark task peptides-func by a comfortable margin using substantially fewer parameters.
>
> ---
>
> ## W1: Well known results repeated
>
> Chief among our contributions is the exploration of novel GNN design spaces, given that a part of the model is *parametrized explicitly on the spectrum*. **There indeed are reasonable parametrizations of such filters that would break permutation equivariance** – e.g., truncating after a fixed number of eigenvectors, specifically if the last considered eigenspace is degenerate. As we show, each eigenspace must be either left out or included entirely to preserve equivariance. As our treatment substantially extends the traditional parametrization of spectral filters as Laplacian polynomials, proving that equivariance still holds is essential to the formal completeness of our work. Additionally, we capture a **more general case than usual**, considering complex-valued eigenvectors. We detail this in §F.1 and will highlight these differences in the method section as well.
>
> We decided to include the fact that "locality relates to spectral smoothness" in the main part *for readability reasons* and because it is a vital insight for our method.
>
> In case we missed important references etc., we would be glad to include them upon the reviewer's request.
>
> ---
>
> ## W2: Connecting the dots
>
> Since the reviewer focuses in their critique on the introduction, we want to briefly point out that the technical arguments can be found in the method section (§3, §3.1, and §3.2). From this technical viewpoint, combining both filters is a fundamental prerequisite for the approximation-theoretic guarantees derived in §3.2.
>
> In the introduction, starting with Figure 1, we illustrate the complimentary limitations of each parametrization on its own. We state in lines 38-39 that a spectrally parametrized convolution should operate on a truncated frequency spectrum for efficiency reasons. This implies a limitation of a purely spectral parametrization. Finally, in lines 61-62, we state explicitly that the combination of both approaches alleviates over-squashing, providing a forward pointer to the technical argument.
>
> In the camera-ready version, we plan to change
> > Conversely, spectral filters act globally ($p_\max$), even with truncation of the frequency spectrum ($\lambda_\text{cut}$) **that is required for efficiency.**
>
> to
> > Conversely, spectral filters act globally ($p_\max$), even with truncation of the frequency spectrum ($\lambda_\text{cut}$). The truncation of the frequency spectrum is required for efficiency. Yet, the combination of spatial and spectral filters provably leverages the strengths of each parametrization.
>
> We appreciate further suggestions and more details on potentially missing connections.
>
> ---
>
> ## Q1: GNNs aim to learn data driven filters. Ideal filters may NOT be a “good” choice.
>
> **We empirically demonstrate that our parametrization can substantially improve over the prior state of the art**, providing strong evidence that the capability of approximating ideal filters is useful! Perhaps we misunderstand a part of the reviewer's concern. However, **it is common practice in deep learning to strive for a maximally general hypothesis class s.t. training can decide for a good parametrization in a data-driven manner**. Popular examples include Graph Isomorphism Networks (GIN) [1] or even the well-known works on universal approximation. These works also show how to overcome certain limitations or how to obtain full generality. While we make a connection to **universal approximation of idealized GNNs** [2] explicit in lines 195-197, in the camera-ready version, we will elaborate more on the motivation in Section 3.2. We should also mention that the discontinuity is solely a worst-case example (lines 204-206) and that our approximation-theoretic discussion implies that S$^2$GNNs are strictly more powerful than Message-Passing GNNs (MPGNNs).
>
> If this answer does not fully address the reviewer's questions, we kindly ask for clarification.
>
> [1] Xu et al. How Powerful are Graph Neural Networks? ICLR 2019.
>
> [2] Wang and Zhang. How Powerful are Spectral Graph Neural Networks. ICML 2022.
>
> ---
>
> ## Q2 & W3: Directionality
>
> The ability to support directed graphs shows the general applicability of S$^2$GNNs. For graph machine learning to generalize, e.g., sequence models, directedness is required. This is why we show that S$^2$GNNs are also exceptional sequence models in the associative recall task.
>
> We are the first to use the Magnetic Laplacian for a spectrally parametrized filter. We provide the important and unique (e.g., for potential $q$) design decisions in lines 298-303 and §H.5. It should be noted that only very few MPGNNs, including MagNet (Zhang et al., 2021), also use the Magnetic Laplacian, out of the thousands of papers that work with graph neural networks.
>
> We present results on directed graphs in Figure 12 (distance regression) and Figure 15 (associative recall), Table 2 (associative recall), Table 4 (TPU Graphs), Table 11 (distance regression), Figure 24 (distance regression), Table 12 (arXiv-year). We ablate the importance of directed graphs, e.g., in Figure 15 and Table 11. In both cases, directionality improves performance.
>
> We will use the extra space in the camera-ready version to make these points more explicit and appreciate further suggestions.

---

> > ### Comment · Area_Chair_8zcd · 2024-08-12
> >
> > Dear Reviewer ohZY
> >
> > Could you please read the rebuttal and respond to the authors?
> > We are approaching the end of the discussion period, and your feedback will be critical.
> >
> > Thank you!
> > AC

---

> > > ### Comment · Area_Chair_8zcd · 2024-08-12
> > >
> > > Dear Reviewer and Authors,
> > >
> > > Since this paper has low and high scores, we should discuss the paper in more detail.
> > >
> > > Specifically, can the reviewers please discuss the review of ohZY with the lowest score and the response from the authors? Do you agree about the weaknesses 1-3? Are you satisfied with the response?
> > > Please feel free to share any additional thoughts you might have.
> > >
> > > Thank you
> > > AC

---

> > > > ### Comment · Reviewer_rAa9 · 2024-08-12
> > > >
> > > > I thank the AC for their involvement in the rebuttal discussion period.
> > > >
> > > > Regarding the review:
> > > >
> > > > - I believe weaknesses 1 and 3 are fair concerns. However, in my opinion, the Authors have properly addressed them in their rebuttal.
> > > >
> > > > - Regarding weakness 2 - I believe that the Authors motivate their approach very well throughout the paper. Nevertheless, including stronger arguments in the Introduction section might make the paper easier to follow. The Authors seem to acknowledge this in their rebuttal.
> > > >
> > > > Overall, I believe that the concerns raised by the Reviewer are legitimate but have been addressed by the Authors. Moreover, in my opinion, the raised concerns are minor in the context of the proposed work. The overall work is on combining spectral convolutions and message-passing, analyzing why the combination works well in practice, especially in modeling long-range relationships. Therefore, I would argue that the initial concerns are insufficient to justify a rejection (3).
> > > >
> > > > Finally, while I can't make any assumptions regarding whether the Authors' responses satisfied Reviewer ohZY, I believe that the authors did a good job answering the questions and clarifying the concerns raised by the Reviewer.
> > > >
> > > > Nevertheless, I am aware that my review is the most positive one, and I believe that input from the other Reviewers would be valuable for this discussion.

---

> > > > > ### Comment · Reviewer_ohZY · 2024-08-13
> > > > > **Official Comment by Reviewer ohZY**
> > > > >
> > > > > I thank the authors for their well put response. I have increased my score.
> > > > >
> > > > > Also, thank you AC and reviewers for engaging in the discussion.

---

> > > > > > ### Author Response · Authors · 2024-08-14
> > > > > >
> > > > > > We thank reviewer ohZY for acknowledging that we resolved his concerns in our rebuttal and reviewer rAa9 for the additional suggestion on including "stronger arguments in the Introduction". We propose to extend the suggested change in our rebuttal to reviewer ohZY:
> > > > > > > Conversely, spectral filters act globally ($p_{\max}$), even with truncation of the frequency spectrum ($\lambda_{\text{cut}}$). The truncation of the frequency spectrum is required for efficiency. Yet, the combination of spatial and spectral filters provably leverages the strengths of each parametrization. **On an intuitive level, the spectral filter extends the entirely global aggregation of a so-called virtual node to a graph-adaptive intra- and inter-cluster message passing (see Fig. 3 & 4). In contrast, message passing recursively aggregates over local neighborhoods.** Utilizing the strengths of both parametrizations ...
> > > > > >
> > > > > > We would appreciate clarification if this does not fully address reviewer rAa9's suggestion. Moreover, we would be grateful for further suggestions from reviewer ohZY for improving the paper's content, methodology, or impact s.t. we can strengthen our contribution and score.

---

### Author Rebuttal · Authors · 2024-08-06

We thank all the reviewers for their time and valuable feedback! Notably, we thank reviewers EZD5, H69b, and rAa9 for uniformly acknowledging our theoretical foundations/analysis of our Spatio-Spectral Graph Neural Networks (S$^2$GNNs), along with our method's general applicability and strong empirical results. We also want to highlight the assessment of reviewer rAa9:
> [...] the work could have a considerable impact on how we design GNN architectures going forward.

At the request of the reviewers, we have conducted several additional experiments and now provide pseudocode as well as a table summarizing the notation. We divert results to the reviewer-specific rebuttal where beneficial for presentation. In the attached pdf of the global response:
1. We compare the runtimes of a GCN with a spectral filter (w/o EVD) on OGB-arXiv (170k nodes), showing that one GCN-layer here is as costly as a spectral filter with 2,500 eigenvectors. This is due to the better parallelizability of dense matrix multiplications over sparse matrix multiplications. Requested by reviewer rAa9.
1. We study the influence of the number of eigenvectors on the real-world task peptides-func. Requested by reviewer H69b.
1. We provide pseudocode for S$^2$GNNs and a real-valued spectral filter. Requested by reviewer H69b.
1. We provide a table summarizing our notation. Requested by reviewer H69b.

Additionally, addressing reviewer EZD5, we now include further ablations on combining spatial with spectral filters, and we study the GCNII architecture. Following the suggestion of reviewer rAa9, we also report preliminary yet competitive results on the large-scale dataset PCQM4Mv2.

---

> ### Author Response · Authors · 2024-08-14
> **Conclusion of Discussion Period**
>
> Dear AC, dear Reviewers,
>
> We greatly appreciate the discussion of our work and we cherish the feedback! In our view, the presented paper makes profound technical contributions and accordingly contains an above-average amount of content (see next paragraph). We made the conscious decision not to withhold any significant contribution from the main body and thus make sure that our work is reviewed in its entirety. While this enforces a more compact writing style, we would like to highlight that, indeed, none of the presented aspects was deemed unnecessary by the reviewers. Instead, reviewers EZD5, H69b, and rAa9 uniformly acknowledged our theoretical foundations/analysis of our Spatio-Spectral Graph Neural Networks (S$^2$GNNs), along with our method's general applicability and strong empirical results. We are moreover certain that the 11% extra space in the camera-ready version will suffice for the discussed adjustments to increase explicitness and, hence, address all the changes suggested by the reviewers.
>
> Arguably, in comparison to impactful works like ChebNet [1] or GCN [2], our technical contributions are of much greater scope. In essence, ChebNet [1] made the Chebyshev-polynomial coefficients of [4] learnable, GCN [2] uses a fixed parametrization again but stacks multiple layers interleaved with non-linearities. Instead, we provide a new modeling paradigm rooted in approximation theory and signal processing that goes beyond pure recursive neighborhood aggregations and unlocks new design spaces.
>
> We appreciate further feedback that would benefit our work.
>
> [1] Defferrard et al. "Convolutional Neural Networks on Graphs with Fast Localized Spectral Filtering" (2016)
>
> [2] Kipf & Welling "Semi-Supervised Classification with Graph Convolutional Networks" (2017)
>
> [3] Veličković et al. "Graph Attention Networks" (2017)
>
> [4] Hammond et al. "Wavelets on Graphs via Spectral Graph Theory" (2009)

---

### Decision · Program_Chairs · 2024-09-25

**Decision:**

Accept (poster)

**Comment:**

This paper studies an essential and timely topic. It proposes a novel hybrid method of spatio-spectral Graph Neural Networks to combine spatial and spectral graph filters and ensure better efficiency and bounds. After a fruitful discussion between the authors and the reviewers, this paper has an overall positive feeling.

The authors should carefully read and address all remaining concerns in the final version of the paper.